# Simultaneous estimation of wintertime sea ice thickness and snow depth from space-borne freeboard measurements

Hoyeon Shi[1], Byung-Ju Sohn[1,2], Gorm Dybkjær[3], Rasmus Tage Tonboe[3], Sang-Moo Lee[1,4,5]

[1]School of Earth and Environmental Sciences, Seoul National University, Seoul, Republic of Korea
[2]Key Laboratory for Aerosol-Cloud-Precipitation of China Meteorological Administration, School of Atmospheric Physics, Nanjing University of Information Science & Technology, Nanjing, China
[3]Danish Meteorological Institute, Copenhagen, Denmark
[4]Center for Environmental Technology, ECEE, University of Colorado-Boulder, Boulder, Colorado, USA
[5]National Snow and Ice Data Center, CIRES, University of Colorado-Boulder, Boulder, Colorado, USA

*Correspondence to*: Byung-Ju Sohn (sohn@snu.ac.kr)

**Abstract.** A method of simultaneously estimating snow depth and sea ice thickness using satellite-based freeboard measurements over the Arctic Ocean during winter was proposed. The ratio of snow depth to ice thickness (referred to as $\alpha$) was defined and used in constraining the conversion from the freeboard to ice thickness in satellite altimetry without prior knowledge of snow depth. Then, $\alpha$ was empirically determined using the ratio of temperature difference of the snow layer to the difference of the ice layer, to allow the determination of $\alpha$ from satellite-derived snow surface temperature and snow–ice interface temperature. The proposed method was evaluated against NASA's Operation IceBridge measurements, and results indicated that the algorithm adequately retrieves snow depth and ice thickness simultaneously: retrieved ice thickness was found to be better than the methods relying on the use of snow depth climatology as input, in terms of mean bias. The application of the proposed method to CryoSat-2 radar freeboard measurements yields similar results. In conclusion, the developed $\alpha$-based method has the capacity to derive ice thickness and snow depth, without relying on the snow depth information as input to the buoyancy equation or the radar penetration correction for converting freeboard to ice thickness.

## 1 Introduction

Satellite altimeters have been used to estimate sea ice thickness for nearly two decades (Laxon et al., 2003; Kwok et al., 2009; Laxon et al., 2013). The altimeters do not measure sea ice thickness directly but measure the sea ice freeboard which is then converted to sea ice thickness with assumptions, for example, regarding the snow depth, snow/ice densities, and radar penetration (Ricker et al., 2014). We hereafter refer to this procedure as 'freeboard to thickness conversion'.

Generally, there are two types of satellite altimeters measuring different sea ice freeboards: 1) Lidar altimeters such as NASA's ICESat (Zwally et al., 2002) and ICESat-2 (Markus et al., 2017) missions measure the total freeboard ($F_t$): the height from the sea surface in leads, to the snow surface. 2) Radar altimeters such as ESA's CryoSat-2 (CS2) (Wingham et al., 2006) measure the radar freeboard ($F_r$): difference in the radar ranging between the sea surface and the radar scattering horizon. By applying

two corrections terms regarding the wave propagation speed change in the snow layer ($F_c$) and displacement of the scattering horizon from the ice surface ($F_p$), the radar freeboard is converted to the ice freeboard ($F_i$): the height from the sea surface to the snow–ice interface ($F_i$). Several studies indicate that the radar scattering horizon is at or above the snow–ice interface depending on ice type and snow/ice conditions (Nandan et al., 2017; Armitage and Ridout, 2015; Willatt et al., 2011; Tonboe

et al. 2010). However, the radar scattering horizon is often treated as the snow–ice interface (Kurtz et al., 2014; Kwok and Cunningham, 2015; Hendricks et al., 2016; Guerreiro et al., 2017, Tilling et al., 2018). The three different freeboards are indicated in Fig. 1.

For both lidar and radar altimeters, snow depth ($h_s$) is required as an input to constrain the freeboard to thickness conversion; thus, the conversion results are highly dependent on snow depth (Ricker et al., 2014; Zygmuntowska et al., 2014; Kern et al.

2015). The buoyancy equation used in the freeboard to thickness conversion describes the balance between buoyancy and the weight of snow and ice. For a given freeboard, snow/ice densities, and assumptions on radar penetration of the snow layer, sea ice thickness ($H_i$) is a function of $h_s$. According to Zygmuntowska et al. (2014), up to 70% of uncertainty in the freeboard to thickness conversion stems from the poorly constrained snow depth. However, mapping the Arctic scale snow depth distribution is challenging. The most commonly used snow depth information necessary for the freeboard to thickness

conversion is the modified version of the snow depth climatology by Warren et al. (1999) (hereafter W99). W99 is based on in-situ measurements at Soviet drifting stations (1954–1991) mostly on multi-year ice (MYI). Kurtz and Farrell (2011) compared W99 with Operation IceBridge (OIB) snow depth measurements in 2009 and claimed that W99 was still valid in the MYI region and significantly differed from OIB snow depth on first-year ice (FYI). Based on that study, Modified W99 (hereafter MW99) was developed, which halves W99 snow depth in regions covered by FYI. MW99 is often used in CS2 ice

thickness products available at CPOM-UCL (Laxon et al., 2013), AWI (Ricker et al., 2014), and NSIDC (Kurtz and Harbeck, 2017).

However, the use of MW99 for the freeboard to thickness conversion understandably yields a substantial error, considering that W99 is climatology and not actual snow depth. This is because the actual snow depth distribution is subject to the year-to-year variation of the snow–ice system, thus the climatology based on the 37-year measurements of snow depth would deviate

significantly from the actual distribution (Webster et al., 2014). Accordingly, such deviation causes errors in the estimation of ice thickness. Thus, additional snow observations covering both MYI and FYI on the Arctic basin-scale would be ideal as a replacement of MW99.

There have been various approaches aimed at obtaining the snow depth distribution over the Arctic scale using satellite observations. Markus and Cavalieri (1998) developed an algorithm based on the Brightness Temperatures (TBs) of Special

Sensor Microwave/Imager (SSM/I) based on the negative correlation of the snow depth with the spectral gradient ratio between 18 and 37 GHz of vertically polarized TBs on the Antarctic FYI. Comiso et al. (2003) have updated the coefficients of this algorithm for the Advanced Microwave Scanning Radiometer for EOS (AMSR-E). However, snow depth retrieval using this algorithm is relatively less accurate when the MYI fraction within the grid cell is significant (Brucker and Markus, 2013).

Recently, Rostosky et al. (2018) suggested a new method: using the lower frequency pair of 7 and 19 GHz to overcome this limitation. Nonetheless, estimating the basin-scale snow depth distribution seems to be a difficult task.

There are other approaches involving the use of the lower frequency measurements at L-band. Using Soil Moisture Ocean Salinity (SMOS) measurements, Maaß et al. (2013) found that 1.4 GHz TB depends on the snow depth through the insulation effect of snow layer, and they determined snow depth by matching Radiative Transfer Model (RTM) simulated TBs with SMOS-measured TBs. Zhou et al. (2018) simultaneously estimated the sea ice thickness and snow depth by adding additional laser altimeter freeboard information, improving the Maaß et al. (2013) approach. However, both of these RTM-based approaches require a priori information on ice properties (e.g. temperature and salinity profiles).

Other satellite remote sensing approaches include the snow depth retrieval using dual-frequency altimetry (Guerreiro et al., 2016; Lawrence et al., 2018, Kwok and Markus, 2018), multilinear regression (Kilic et al., 2019), and a neural network approach (Braakmann-Folgmann and Donlon, 2019). In spite of promising results, the dual frequency altimetry method is available only for regions where two altimeters overlap with each other, reducing the great deal of spatial coverage. On the other hands, the regression/neural network methods based on AMSR-2 TBs are prone to the overfitting problem, limiting their applications to other microwave sensors.

Here, let us switch our point of view to solving the buoyancy equation instead of retrieving snow depth directly. Remember that there are two unknowns (snow depth and ice thickness) in the buoyancy equation for given snow/ice densities, freeboard, and assumptions on radar penetration of the snow layer. The attempt so far has been to add one constraint (snow depth information) to the buoyancy equation for solving ice thickness. However, if a particular relationship between two unknowns is available, it can be used to constrain the equation yielding both ice thickness and snow depth simultaneously.

To identify such a relationship, this study examines the vertical thermal structure within the snow/ice layers observed by drifting buoys. The vertical thermal structure of a snow–ice system in winter is rather simple; the temperature profile of the snow–ice system can be assumed to be piecewise linear, as illustrated in Fig. 1. Therefore, the temperatures at three interfaces can represent the thermal state of the snow–ice system fairly well; they are (1) air–snow interface temperature ($T_{as}$), (2) snow–ice interface temperature ($T_{si}$), and (3) ice–water interface temperature ($T_{iw}$). $T_{iw}$ is assumed to be nearly constant at the freezing temperature of seawater (Maaß et al., 2013), implying that two other interface temperatures ($T_{as}$ and $T_{si}$) are sufficient to describe the thermal structure of the system.

Based on this thermal structure, there is a constraint relating the snow depth and ice thickness. In identifying this constraint, conductive heat flux is assumed to be continuous through the snow–ice interface (Maykut and Untersteiner, 1971), implying that conductive heat fluxes within the snow and ice layers are same under the steady-state assumed in the given thermal structure. As the conductive heat flux is proportional to the bulk temperature difference of the layer divided by its thickness, it is possible to deduce the relationship between snow depth and ice thickness from the given thermal structure.

Once the relationship is obtained, then it is possible to apply it to the Arctic Ocean basin-scale because the thermal structure can be resolved from satellites, as shown in the recently available basin-scale and long-term satellite-derived interface temperatures (Dybkjær et al., 2020; Lee et al., 2018). In determining the snow depth along with the ice thickness, instead of

using the snow depth as an input to solve for the ice thickness, we intend to (1) examine the relationship between the vertical thermal structure of a snow–ice system ($T_{as}$ and $T_{si}$) and the thicknesses of the snow and ice layer ($h_s$ and $H_i$) using buoy measurements, (2) retrieve the sea ice thickness and the snow depth simultaneously by applying their relationship to the freeboard to thickness conversion as a constraint, thus replacing the snow depth information. The result may reduce uncertainty in the freeboard to ice thickness conversion by replacing the currently used snow depth climatology.

## 2 Method

Here, we provide the theoretical background of how the snow–ice thickness ratio ($\alpha = h_s / H_i$) can be related to $T_{as}$ and $T_{si}$. Then, after empirically determining the relationship of $\alpha$ to $T_{as}$ and $T_{si}$ from buoy measured temperature profiles, $\alpha$ obtained from satellite observed $T_{as}$ and $T_{si}$ is then used to constrain the conversion from freeboard to ice thickness over the Arctic Ocean during winter.

### 2.1 Theoretical background

We intend to find a relationship between snow depth and ice thickness in terms of the vertical thermal structure of the snow–ice system. Because the temperature gradients within the snow and ice layers are linked to both temperature and thickness, we focus on the temperature gradient. Owing to the physical reasoning that the conductive heat flux is continuous across the snow–ice interface (Maykut and Untersteiner, 1971), the following relationship is valid at the snow–ice interface:

$$k_{snow} \left. \frac{\partial T_{snow}}{\partial z} \right|_{z=0} = k_{ice} \left. \frac{\partial T_{ice}}{\partial z} \right|_{z=0} \tag{1}$$

In Eq. (1), the subscripts *snow* and *ice* denote their respective layers while $T$, $k$, and $z$ denote temperature, thermal conductivity, and depth, respectively. The snow–ice interface is defined as $z = 0$. Assuming a piecewise linear temperature profile within the snow–ice layer, Eq. (1) can be rewritten as follows:

$$k_{snow} \frac{T_{as}-T_{si}}{h_s} = k_{ice} \frac{T_{si}-T_{iw}}{H_i} \tag{2}$$

where subscripts *as*, *si*, and *iw* denote the air–snow, snow–ice, and ice–water interface, respectively, and $H_i$ and $h_s$ denote the sea ice thickness and snow depth as in Fig. 1. Introducing a variable $\alpha$, which is the snow–ice thickness ratio, Eq. (2) becomes:

$$\alpha = \frac{h_s}{H_i} = \frac{k_{snow}}{k_{ice}} \frac{\Delta T_{snow}}{\Delta T_{ice}} \tag{3}$$

Here, $\Delta T$ denotes the temperature difference between the top and bottom of each of the snow and ice layers (i.e. $\Delta T_{snow} = T_{as} - T_{si}$, $\Delta T_{ice} = T_{si} - T_{iw}$). As explained in detail in Sect. 2.3, $\alpha$ can be used to constrain the freeboard to thickness conversion. Thus, once $\alpha$ is known, both snow depth and ice thickness can be simultaneously estimated from altimeter-measured freeboard, instead of using snow depth data for ice thickness retrieval.

## 2.2 Empirical determination of '$\alpha$-prediction equation' from buoy measurements

To obtain $\alpha$, the conductivity ratio ($k_{snow}/k_{ice}$) should be known even if the temperature difference ratio ($\Delta T_{snow}/\Delta T_{ice}$) is given. In this study, instead of using the conventional conductivity ratio found in literature, it is empirically determined using buoy-measured $\alpha$ and $\Delta T_{snow}/\Delta T_{ice}$. Thus, the interface should be defined and determined from buoy-measured temperature profiles, which show a piecewise linear temperature profile as shown in Fig. 1.

The buoy-measured temperature profiles in the vertical resolution of 10 cm are used in this study (Sect. 3.1). Although the instrument initially sets the zero-depth reference position to be approximately at the snow–ice interface, the reference position can deviate from the initial location if the ice deforms, or if the snow refreezes after the temporary melt into snow-ice. In addition, the interfaces (air–snow, snow–ice, and ice–water) may be located in between measurement levels in a 10 cm spacing. Therefore, an interface searching algorithm is developed to determine three interfaces ($y_{as}$, $y_{si}$, $y_{iw}$) and their respective temperatures ($T_{as}$, $T_{si}$, $T_{iw}$) by extrapolating each piecewise linear temperature profile iteratively.

The interface searching algorithm iterates three processes to find the location and temperature of each interface: it (1) divides temperature profile into four layers using the most recently available locations of the three interfaces, (2) finds a linear regression line of the temperature profile at each layer, and (3) updates the location and temperature of each interface by finding an intersection between two adjacent regression lines. The algorithm fails if the temperature profile is far from linear, or the thickness of a certain layer is too thin to have less than two data points. More detailed procedures for determining the interface are provided in Fig. 2, as a flow chart. The outputs are $T_{as}$, $T_{si}$, $T_{iw}$, $H_i$ (= $y_{as}$ - $y_{si}$), and $h_s$ (= $y_{si}$ - $y_{iw}$). Examples of the interface searching results for 15-day averaged temperature profiles are shown in Fig. 3. The algorithm works adequately for both CRREL-IMB (Fig. 3a–c) and SHEBA buoy data (Fig. 3d–f).

Since $T_{as}$, $T_{si}$, $T_{iw}$, $H_i$, and $h_s$ can be obtained from the previous interface determination with buoy data, the calculation of $\Delta T_{snow}/\Delta T_{ice}$ and $\alpha$ is straightforward. Then, an empirical relationship can be obtained by relating $\alpha$ to $\Delta T_{snow}/\Delta T_{ice}$ by running a regression model, and details are given in Sect. 4. However, for the time being, we assume that the regression equation (referred to as an '$\alpha$-prediction equation' that will be discussed in Sect. 4) is used to predict $\alpha$ from $\Delta T_{snow}/\Delta T_{ice}$.

## 2.3 Simultaneous estimation of ice thickness and snow depth from satellite-based freeboard using $\alpha$

In this section, we describe how $\alpha$ can be used to constrain the freeboard to thickness conversion. Based on the assumed hydrostatic balance, ice thickness can be obtained from satellite-borne total freeboard or ice freeboard as follows:

$$H_i = \frac{\rho_w}{\rho_w - \rho_i} F_t - \frac{\rho_w - \rho_s}{\rho_w - \rho_i} h_s \tag{4}$$

$$H_i = \frac{\rho_w}{\rho_w - \rho_i} F_i + \frac{\rho_s}{\rho_w - \rho_i} h_s \tag{5}$$

Here, $\rho_w$, $\rho_i$, and $\rho_s$ denote the bulk densities of water, ice, and snow layer, respectively. Ice freeboard is obtained from radar freeboard by applying two correction terms regarding the change of the wave propagation speed in snow layer ($F_c$) and the displacement of the scattering horizon from the ice surface ($F_p$) (Kwok and Cunningham, 2015).

$$F_i = F_r + (F_c - F_p) \tag{6}$$

The correction terms are expressed in the following equations (Armitage and Ridout, 2015; Kwok and Markus, 2018).

$$F_c = (\eta_s - 1) f h_s \tag{7}$$

$$F_p = (1 - f) h_s \tag{8}$$

Here, $\eta_s$ denotes the refractive index of the snow layer and $f$ denotes the radar penetration factor (Armitage and Ridout, 2015), which is the depth of the radar scattering horizon relative to the snow depth (e.g. $f = 1$ if the radar scattering horizon is at snow–ice interface and $f = 0$ if the radar scattering horizon is at air-snow interface). Combination of Eqs. (6) – (8) yields the following relationship.

$$F_i = F_r + (f\eta_s - 1) h_s \tag{9}$$

Ice freeboard in Eq. (5) can be substituted by radar freeboard and snow depth using Eq. (9), i.e.:

$$H_i = \frac{\rho_w}{\rho_w - \rho_i} F_r + \frac{(f\eta_s - 1)\rho_w + \rho_s}{\rho_w - \rho_i} h_s \tag{10}$$

According to Eq. (10), the ice thickness can be estimated from the radar freeboard and the snow depth. Note that Eq. (10) becomes equivalent to the equation for the total freeboard (Eq. (4)) if $f = 0$ (i.e. if there is no radar penetration into snow layer). With the use of $\alpha$, defined in Eq. (3), Eqs. (4) and (10) become

$$H_i = \frac{\rho_w}{\rho_w - \rho_i + \alpha(\rho_w - \rho_s)} F_t \tag{11}$$

$$H_i = \frac{\rho_w}{\rho_w - \rho_i - \alpha\{(f\eta_s - 1)\rho_w + \rho_s\}} F_r \tag{12}$$

From Eqs. (3), (11) and (12), it is evident that the snow depth and ice thickness can be simultaneously estimated from the freeboards once $\alpha$, $\rho$, $f$ and $\eta_s$ are known.

In order to obtain $\alpha$ from satellite measurements of $T_{as}$ and $T_{si}$, we need to calculate the temperature difference ratio ($\Delta T_{snow}/\Delta T_{ice}$). For the calculation, $T_{iw}$ is set to be -1.5 °C. The freezing temperature of seawater is often assumed to be -1.8 °C; however, the value of -1.5 °C is chosen, based on the buoy observations. A sensitivity test indicated that the influence of a 0.3 °C difference in the freezing temperature on $\alpha$ was negligible (e.g. approximately 1.2% difference for typical interface temperatures of $T_{as} = -30$ °C and $T_{si} = -20$ °C). $\alpha$ values are calculated only at the pixel whose monthly sea ice concentration (SIC) is greater than 95% and rejected if $T_{as}$ is warmer than $T_{si}$. The densities are prescribed with those used for OIB data processing: $\rho_s$, $\rho_i$, and $\rho_w$ are 0.320 g cm$^{-3}$, 0.915 g cm$^{-3}$, and 1.024 g cm$^{-3}$, respectively (Kurtz et al., 2013). Although $\rho_s$ varies

seasonally (Warren et al., 1999) and $\rho_i$ is greater for MYI than FYI (Alexandrov et al., 2010), we use the same densities as those of OIB data because we intend to compare outputs against OIB data. In solving Eq. (12), cases showing negative ice thickness ($\alpha \geq \alpha_{crit} = 0.291$ for the given densities and radar penetration factor) are rejected. Radar penetration factor $f$ is set to be 0.84 for CS2 (Armitage and Ridout, 2015) and $\eta_s$ is parameterized as a function of the snow density, i.e., $\eta_s = (1+0.51\rho_s)^{1.5}$

(Ulaby et al., 1986).

Before the Arctic basin-scale retrieval, ice thickness is estimated from OIB total freeboard measurement using Eq. (11), and from OIB-derived radar freeboards (Sect. 3.3) using Eq. (12), using satellite-derived $\alpha$ as a constraint. At the same time, the corresponding snow depth is derived by multiplying the obtained sea ice thickness and $\alpha$ (Eq. (3)). Sea ice thicknesses are also calculated from Eqs. (4) and (10), using MW99 as snow depth, to examine how simultaneous retrievals compare with ice

thickness estimation using MW99. To differentiate various outputs, obtained snow depth and ice thickness are expressed with nomenclature such as '(constraint, freeboard source)'. For example, the snow depth estimated from satellite-derived $\alpha$ and OIB total freeboard is referred to as '$h_s (\alpha^{sat}, F_t^{OIB})$', and sea ice thickness from the MW99 and OIB radar freeboard is referred to as '$H_i (h_s^{MW99}, F_r^{OIB})$'. Finally, ice thickness and snow depth are estimated from CS2 radar freeboard (Sect. 3.4) over the Arctic Ocean.

**3 Data**

Here, we provide detailed information on the data sets used for the development of the retrieval algorithm, evaluation, and application to the Arctic ocean basin scale.

**3.1 CRREL and SHEBA buoy data**

To determine the empirical relationship between $\alpha$ and $\Delta T_{snow}/\Delta T_{ice}$ using Eq. (3), we need information regarding $h_s$, $H_i$, $T_{as}$,

$T_{si}$, and $T_{iw}$ (as depicted in Fig. 1). These are sourced from temperature profiles observed by buoys deployed over the Arctic, as part of the Surface Heat Energy Budget of the Arctic (SHEBA) campaign (Perovich et al., 2007) and the Cold Regions Research and Engineering Laboratory Ice Mass Balance (CRREL-IMB) buoy program (Perovich et al., 2019). Those buoy observations are stored for further analysis if there are no missing records over the entire period ranging from November to March of the following year. Detailed information regarding ice type and initial snow/ice thickness at deployment locations

are given in Table 1.

Time averages of temperature profiles are used as input to the interface searching algorithm (described in Sect. 2.2) to meet the required near-equilibrium states (e.g. linear temperature profile). However, because of the possibility that the results are dependent on the averaging period, we examine the results using various averaging periods from one to 30 days.

## 3.2 Satellite-derived skin and interface temperatures

For applying the buoy-based $\alpha$-prediction equation in retrieving the snow/ice thicknesses over the Arctic Ocean, satellite-derived $T_{as}$ and $T_{si}$ data are necessary. In this study, $T_{as}$ is obtained from Arctic and Antarctic ice Surface Temperatures from thermal Infrared satellites sensors – version 2 (AASTI-v2) data (Dybkjær et al., 2020), and the monthly mean for the 1982–2015 period is obtained from daily products. AASTI $T_{as}$ is derived from CM SAF cLoud, Albedo and surface Radiation dataset from AVHRR data - Edition 2 (CLARA-A2) dataset (Karlsson et al., 2017), based on the algorithm described in Dybkjær et

al. (2018). Information on the validation of this product is found in Dybkjær and Eastwood (2016). It is available in a 0.25° grid format, however, because other satellite data sets such as SIC are available in a 25 km Polar Stereographic SSM/I Grid, AASTI-v2 data are re-gridded in the same 25 km grid format. This reformatted AASTI-v2 dataset is called 'satellite skin temperature'.

$T_{si}$ is obtained from Snow/Ice Interface Temperature (SIIT) produced by Lee et al. (2018) over 30 years (1988–2017) of

wintertime (December to February) using SSM/I and Special Sensor Microwave Imager/Sounder (SSMIS) homogenized TBs (Berg et al., 2018). The daily data are in the 25 km grid format. Lee et al. (2018) reported that the satellite-derived $T_{si}$ is consistent with snow–ice interface temperatures observed by CRREL-IMB buoys, with the correlation coefficient, bias, and RMSE of 0.95, 0.15 K and 1.48 K, respectively. In this study, we also produced $T_{si}$ for March using the same algorithm of Lee et al. (2018) for evaluating results against OIB data which are mostly collected during spring. Monthly composites are

constructed by averaging daily data for grid cells where the data frequency is over 20 days. This product is called 'satellite interface temperature'.

## 3.3 OIB data

In this study, OIB snow depth ($h_s^{OIB}$) and total freeboard ($F_t^{OIB}$) are used as a reference in the evaluation of snow depth and ice thickness retrieved from the developed algorithm. NASA's OIB is an aircraft mission and it measures snow depth and total

freeboard over the Arctic using the snow radar, Digital Mapping System (DMS), and Airborne Topographic Mapper (ATM) (Kurtz et al., 2013). OIB ice thickness is derived from measured snow depth and total freeboard, for the given snow and ice densities using Eq. (4). In this study, the OIB radar freeboard ($F_r^{OIB}$) is derived from $F_t^{OIB}$ and $h_s^{OIB}$ using the combined relationship of $F_i = F_t - h_s$ and Eq. (9) as follows:

$$F_r^{OIB} = F_t^{OIB} - h_s^{OIB} - (f\eta_s - 1)h_s^{OIB} \tag{13}$$

Because the main objective of using OIB data is to evaluate the relative performance of the simultaneous retrieval method when the method is applied to CS2 data, the radar penetration factor ($f$) for OIB data processing is also set to be 0.84. In the data processing chain, $h_s^{OIB}$ is removed if it is smaller than the given uncertainty level of the dataset (~5.7 cm) or it is larger than the total freeboard $F_t^{OIB}$.

Five years of OIB data during 2011-2015 period are utilized in this study. The level 4 dataset (Kurtz et al., 2015) during 2011-2013 period and Quick look dataset during 2014-2015 period are obtained from the NSIDC website (see the data availability section). Because we use the November–March period for the buoy analysis, only March OIB data are considered for the evaluation. The OIB data are also reformatted into the 25 km grid format by averaging pixel-level OIB observations on the 25 km grid.

## 3.4 CS2 data

For examining the Arctic Ocean basin distribution of ice thickness and snow depth, CS2 freeboard measurement summary data are used (Kurtz and Harbeck, 2017). They are monthly mean composites of CS2 ice freeboard data in the 25 km Polar Stereographic SSM/I Grid format, covering the entire Arctic, and available from September 2010. Detailed descriptions of the retracker algorithm used in this dataset are found in the study by Kurtz et al. (2014). The dataset also includes MW99 ($h_s^{MW99}$) and W99 snow density climatology used for producing the ice freeboard.

The CS2 ice freeboard data ($F_i^{CS2}$) distributed by NSIDC (Kurtz and Harbeck, 2017) assumed that the radar scattering horizon is at the snow–ice interface and applied a wave propagation speed correction. However, the correction was made using $h_s^{MW99}$ and W99 snow density climatology with an erroneous form of $h_c = (1 - \eta_s^{-1}) h_s$, instead of the proper form of $h_c = (\eta_s - 1) h_s$ (Mallett et al., 2020). Thus, at this point, it is straightforward to derive the CS2 radar freeboard by removing the correction term as in the following equation.

$$F_r^{CS2} = F_i^{CS2} - (1 - \eta_s^{-1})h_s^{MW99} \tag{14}$$

Here, $\eta_s$ was parameterized as a function of the snow density, i.e. $\eta_s = (1 + 1.7\rho_s + 0.7\rho_s^2)^{0.5}$ (Tiuri et al., 1984), and $\rho_s$ is taken from the W99 climatology, after Kurtz and Harbeck (2017). Then CS2 ice thickness is re-produced from $F_r^{CS2}$ and $h_s^{MW99}$ by using Eq. (10) with the constant densities and the radar penetration factor described in Sect. 2.3. Those $h_s^{MW99}$ and $H_i$ ($h_s^{MW99}$, $F_r^{CS2}$) values are used for comparison with results from our simultaneous method.

## 3.5 Sea ice concentration

Calculation of $\alpha$ is done for those pixels where the monthly SIC is greater than 95% (as described in Sect. 2.3). To determine pixels that meet this SIC criterion, 'bootstrap sea ice concentrations from Nimbus-7 SMMR and DMSP SSM/I-SSMIS version 3' produced by Comiso (2017) are used. This SIC dataset is provided in the 25-km Polar Stereographic SSM/I grid format.

# 4 Results

## 4.1 The empirical relationship between $\alpha$ and $\Delta T_{snow}/\Delta T_{ice}$

We examine variables (i.e. $T_{as}$, $T_{si}$, $T_{iw}$, $H_i$, and $h_s$) obtained from buoy observations by applying the interface searching algorithm. In the scatter plot of weekly-averaged $\Delta T_{snow}/\Delta T_{ice}$ versus $\alpha$ (Fig. 4a), it appears that $\alpha$ linearly increases with $\Delta T_{snow}/\Delta T_{ice}$ when the ratio is smaller than 1.8, but the linear slope becomes smaller when $\Delta T_{snow}/\Delta T_{ice}$ is larger than 1.8. This pattern of the slopes is found to be nearly invariant from year to year, as observed in different colors appearing in the entire range of $\Delta T_{snow}/\Delta T_{ice}$ in Fig. 4a. We also found that this slope pattern is the consistent nature even for different data sets; two different data sets (red points for SHEBA and other points for CRREL) covering various ranges of $\Delta T_{snow}/\Delta T_{ice}$, show similar distributions along the two different slopes. Thus, the slope pattern is not due to different data sources or different data periods. Further analysis of the two slopes is found in Appendix A.

Taking such a two-slope pattern with $\Delta T_{snow}/\Delta T_{ice}$ into account, we introduce a piecewise linear function that may express the slope pattern, i.e.:

$$y = \begin{cases} a_1 x + b_1 & x \le x_0 \\ a_2 x + b_2 & x > x_0 \end{cases}, \quad x_0 = \frac{b_1 - b_2}{a_2 - a_1} \tag{15}$$

In Eq. (15), $x$ and $y$ correspond to $\Delta T_{snow}/\Delta T_{ice}$ and $\alpha$, respectively, and $x_0$ is the point where the slope transition takes place. Applying Eq. (15) to data points from buoy-based variables, the regression coefficients ($a_1$, $b_1$, $a_2$, $b_2$) and transition point ($x_0$) are determined by minimizing the total variance - obtained regression line is plotted in Fig. 4a. $\alpha$ is predicted using the determined regression equation (hereafter referred to as $\alpha$-prediction equation) and compared to the original $\alpha$ values to see how well the regression was performed. The comparison of $\alpha$ with predicted values in Fig. 4b shows that the regression equation is well fitted because of the zero bias and 91.9% of explained variance.

Although the slope pattern discussed with Eq. (15) and Fig. 4 is based on the weekly averages, the slope pattern seems to be consistent among the data averaging periods except for an averaging period shorter than five days. Regressions in the form of Eq. (15) are performed with buoy data averaged with different averaging periods to understand the slope pattern. Regression coefficients and transition point for the chosen averaging periods are examined, and results for four averaging periods are given in Table 2. Detailed information on the coefficients and associated statistics varying with the averaging period is given in Fig. 5. The positions of slope change ($x_0$) are located at approximately 1.8, delineating a nearly invariant slope pattern, regardless of different data averaging periods. Fig. 5a shows that coefficients do not vary much with different averaging periods while coefficients of the first part of the regression line ($a_1$ and $b_1$, $x \le x_0$) vary less than those of the second part ($a_2$ and $b_2$, $x > x_0$). The regression equations show that the explained variance ($R^2$) rises quickly when the averaging period is longer but levels off when data are averaged over a period that is longer than seven days. The bias appears to be near zero over the various averaging periods. Thus, regression performance is found to be comparable if data are averaged over a period that is longer than a week.

## 4.2 Evaluation against OIB estimates

According to the regression results, it is possible to estimate $\alpha$ from the $\Delta T_{snow}/\Delta T_{ice}$. Since the $\Delta T_{snow}/\Delta T_{ice}$ can be calculated from the satellite skin and interface temperature (as described in Sect. 3.2), the corresponding $\alpha$ can be estimated from satellite measurements. Thus, we are able to simultaneously retrieve sea ice thickness and snow depth from altimeter-based freeboard measurements, following Eqs. (11) and (12). We test and evaluate this simultaneous retrieval approach using OIB data. Accordingly, ice thickness and snow depth are simultaneously estimated from OIB freeboard measurements and evaluated against the OIB snow depth ($h_s^{OIB}$) and ice thickness ($H_i^{OIB}$).

To calculate $\alpha$, a data averaging period must be selected. Considering that the monthly composite of satellite freeboard measurements is needed to retrieve snow/ice thickness in the Arctic basin scale, it seems appropriate to use the monthly averaging period to calculate the monthly $\alpha$ distribution. Thus, we use the monthly averaged satellite temperatures and the coefficients for the 30-day averaging period (Table 2) to calculate $\alpha$.

We simultaneously retrieved $H_i$ and $h_s$ for each year's March during 2011–2015 period from the reformatted OIB freeboard measurements (Sect. 3.3) together with satellite-derived $\alpha$ ($\alpha^{sat}$). As expressed in Eqs. (11) and (12), two different ice thickness retrievals are possible, depending on the use of the freeboard type (i.e. total freeboard $F_t$ vs. radar freeboard $F_r$). Two accordingly associated retrievals of snow depth are available. Retrieved results of ice thickness ($H_i$) and snow depth ($h_s$) from the use of OIB total freeboard and radar freeboard are given in the first and second row of Fig. 6, respectively. Corresponding OIB measurements are given at the bottom of Fig. 6. The comparison between any snow/ice retrievals and OIB measurements appear to be consistent with each other for both snow depth and ice thickness, in terms of magnitudes and distribution.

To compare the results quantitatively, scatterplots comparing retrievals against OIB measurements are made, along with statistics for the snow depth and ice thickness retrievals, in the top four panels of Fig 7. The top-two left panels are derived from the use of OIB total freeboard ($F_t^{OIB}$) while the top-two right panels are derived from the OIB radar freeboard ($F_r^{OIB}$). The comparison is done only for pixels where all four products (i.e. snow/ice thicknesses from two different freeboards) are available. This indicates that the snow depth from the total freeboard (top left) is fairly consistent with the OIB snow depth, with a correlation coefficient of 0.73 and with a near-zero bias. The retrieved ice thickness from the total freeboard (middle left) appears to be consistent with OIB ice thickness, with a correlation coefficient of 0.93 and a bias around 8.5 cm. The RMSEs for snow depth and ice thickness are 6.8 cm and 44.3 cm, respectively. Based on the comparison results, Eq. (15) obtained from buoy measurements can be successfully implemented with space-borne total freeboard measurements for the simultaneous retrieval of snow depth and ice thickness.

Following Eq. (12), snow depth and ice thickness retrievals are made from the use of radar freeboard measurements, and results are presented in the top-two right panels in Fig. 7. On the one hand, the comparison of obtained ice thickness against the OIB ice thickness indicates that the retrieved ice thickness shows similar quality as that retrieved from the total freeboard measurements. On the other hand, snow retrievals from the radar freeboard show more scattered features, compared with snow retrieval results from the total freeboard. More scattered features found in the snow depth from the radar freeboard are likely

due to the larger sensitivity of the retrieved $\alpha$ and the prescribed densities, as noted in Eq. (12). Note that Eq. (12) has a smaller denominator than that for Eq. (11). Results of associated sensitivity analysis can be found in Appendix B.

We now examine how the use of MW99 for retrieving sea ice thickness from ICESat and CS2 measurements compares with results from our simultaneous method. To do so, OIB-measured total freeboard and radar freeboard are converted into ice thickness using MW99 as input to solve Eqs. (4) and (10). In this study, these two ice thickness retrievals with the use of MW99 are referred to as "ICESat-like" thickness and "CS2-like" thickness, respectively, and their comparisons are now observed in two panels at the bottom of Fig. 7. According to our analysis, ICESat-like thickness tends to underestimate the ice

thickness by about 47.9 cm when MW99 is used, in comparison to OIB thickness and CS2-like ice thickness shows an overestimate of about 25.5 cm. Nevertheless, their correlation coefficients and RMSEs are similar to the results obtained from the $\alpha$ method.

Better agreement of $H_i$ from the simultaneous method with $H_i^{OIB}$ may be due to the fact that the simultaneously estimated $h_s$ is more consistent with $h_s^{OIB}$ ($h_s^{MW99}$ is likely larger than $h_s^{OIB}$, as shown in Fig. S1). Note that all inputs are the same except the

snow depth. The negative bias of ICESat-like thickness and positive bias of CS2-like thickness reflect expected responses in different signs to the same snow depth error, as shown in different signs in the last terms of Eqs. (4) and (10) (also note Eq. (B2) in Appendix B). Because of this reasoning, if there are decreasing trends in not only ice thickness but also snow depth, the decreasing trend of ice thickness estimated from the constant snow depth will be diminished in radar, while being amplified in lidar. Because of this, the construction of the ice thickness (or volume) trend from the two different satellite altimeters would

be problematic if MW99 is used for the freeboard to thickness conversion. For example, it would be hard to compare the sea ice thickness records estimated from ICESat and CS2 observations and to extend the current ice thickness record from CS2 with recently launched NASA's ICESat-2 which carries a lidar altimeter, for the same reason.

### 4.3 Simultaneous retrieval of ice thickness and snow depth from CS2 measurements

We have demonstrated that the method of simultaneously retrieving the sea ice thickness and snow depth was successfully

implemented with OIB measurements. Now we extend the proposed approach to satellite freeboard measurements. Here, the method is tested with CS2 freeboard measurements, solving for $H_i$ in Eq. (12), and $\alpha$ is obtained from the collocated satellite skin and interface temperature data.

Monthly means of CS2-estimated freeboard ($F_r$), retrieved $\alpha$, ice thickness ($H_i$), and snow depth ($h_s$) for December 2013 to March 2014 are given in Fig. 8. The geographical distribution of $\alpha$ indicates that $\alpha$ is largest in January and becomes smaller

during the following months. Geographically, there seems to be no particular distribution of $\alpha$ between months, although interestingly the lowest $\alpha$ values are always found over the north of the Canadian Archipelago and the western part of the Arctic Ocean shows $\alpha$ values that are generally larger than those over the eastern part.

Retrieved ice thickness from the CS2 freeboard ($F_r$) using obtained $\alpha$ is presented in the third row of Fig. 8. As expected, as noted in Eq. (12), $H_i$ shows a similar geographical distribution to radar freeboard (the first row). The thickest area is located

north of the Canadian Archipelago, where the ice appears thicker than 4 m. On the other hand, most of the FYI thickness appears to range from 1.0 m to 2.0 m. The snow depth $h_s$ is obtained by multiplying $\alpha$ by $H_i$ (in 2nd and 3rd rows), following Eq. (3), and results are shown in the bottom row. The obtained snow distribution indicates that thicker (thinner) snow areas are generally coincident with thicker MYI (thinner FYI) areas. Such similarity should be consistent with the notion that MYI should accumulate more precipitation than FYI because of its longer existence.

To assess the accuracy of CS2 retrievals, reference snow depth and ice thickness collocated with CS2 freeboard in space and time are necessary. However, different from simultaneous retrievals from OIB freeboards in Sect. 4.2, evaluation with the required matching data may not be possible from the monthly composite of CS2 data used in this study. Here, instead of using monthly collocated match-up data, an indirect way is used to examine the accuracy of CS2 retrievals. We do so by examining whether the relationship between the simultaneous method and the MW99 method, based on retrievals from the OIB freeboard,

can be reproduced by CS2-based retrievals. If similar results are obtained, respective accuracies can be deduced against those noted from the evaluation against OIB measurements.

   The relationships, which can be obtained from analysis in Sect. 4.2 (i.e. $h_s$ ($\alpha^{sat}$, $F_r^{OIB}$) vs. $h_s^{MW99}$ and $H_i$ ($\alpha^{sat}$, $F_r^{OIB}$) vs. $H_i$ ($h_s^{MW99}$, $F_r^{OIB}$)), are compared with the relationships found in the current results in Fig. 8 (i.e. $h_s$ ($\alpha^{sat}$, $F_r^{CS2}$) vs. $h_s^{MW99}$ and $H_i$ ($\alpha^{sat}$, $F_r^{CS2}$) vs. $H_i$ ($h_s^{MW99}$, $F_r^{CS2}$)); the results are presented in Fig. 9. Observably, the relationships from CS2 freeboard data

(Fig. 9b, d) are very similar to the relationship obtained from the comparison results from OIB measurements (Fig. 9a, c). This similarity of the slope strongly indicates that the CS2-based sea ice thickness from the current $\alpha$ method has similar accuracy to that found in the evaluation against OIB measurements (Sect 4.2). Further uncertainty estimates for CS2-derived products can be found in Appendix C.

## 5. Conclusions and Discussion

A new approach towards simultaneously estimating snow depth and ice thickness from space-borne freeboard measurements was proposed and tested using OIB data and CS2 freeboard measurements. In developing the algorithm, the vertical temperature slopes were assumed to be linear within the snow and ice layers so that continuous heat flux could be maintained in both layers. This assumption allowed for the description of the snow–ice vertical thermal structure with snow skin temperature, snow–ice interface temperature, the water temperature at the ice–water interface, snow depth, and ice thickness.

Based on the continuous heat transfer assumption, the snow–ice thickness ratio ($\alpha = h_s / H_i$) was introduced and could then be embedded into the freeboard to ice thickness conversion equations. Thus, information on both ice thickness and snow depth can be derived once $\alpha$ is known in case of the availability of a freeboard, without relying on the snow depth information as an input to the conversion from freeboard to ice thickness. From the drifting buoy measurements of the temperature profile, snow depth, and ice thickness over the Arctic Ocean, we demonstrated that $\alpha$ can be reliably determined using the ratio of the vertical

difference of the snow-layer temperature to the vertical difference of ice-layer temperature ($\Delta T_{snow}/\Delta T_{ice}$). An empirical regression equation was obtained for predicting $\alpha$ from three interface temperatures.

    Before applying $\alpha$-prediction equation to simultaneously retrieve the ice thickness and snow depth from satellite-borne freeboard measurements, the algorithm was evaluated using OIB measurements, in conjunction with satellite-derived snow skin temperature and snow–ice interface temperature. Evaluation results demonstrated that our proposed algorithm adequately

retrieved both parameters simultaneously. As a matter of fact, the ice thickness results were more accurate than they were from the current retrieval methods relying on the input of snow depth (this time MW99 snow climatology), in terms of mean bias. It should be noted that in this case, snow depth is a retrieval product, instead of being input to the freeboard to ice thickness conversion adopted by CS2 or ICESat retrieval. The application was finally made for the retrieval of the snow depth and ice thickness from CS2 radar freeboard measurements from December 2013 to March 2014 using $\alpha$ as a constraint. Results showed

that the quality of the obtained ice thickness was similar to that obtained from evaluation results against OIB measurements. Retrieved snow depth distributions were also found to be consistent with expectations.

    In the retrieval process, we may be concerned about the applicability of the algorithm developed with buoy observations representing the point measurements, to the larger spatial and temporal scales of satellite measurements. This concern may be relevant upon observing the range of $\alpha$ values. $\alpha$ in the satellite's monthly and 25 km x 25 km spatial scales was found to be

generally smaller than 0.2. The smaller range of $\alpha$ compared to that shown in the buoy analysis results is likely due to the scale differences, indicating that extreme $\alpha$ values often shown in buoy measurements (due to very thick snow and/or very thin ice) may never be observed in satellite measurements. However, the range may not be a problem because the relationship (Eq. (3)) expresses the thermal equilibrium condition described by the temperature at three interfaces, the ratio of snow and ice thickness, and the ratio of thermal conductivity between snow and ice. Considering that the algorithm is based on the equilibrium

conditions, results should be valid regardless of spatial and temporal scales if the prerequisite equilibrium conditions are met. Apparently, buoy observations contain so many different cases that equilibrium conditions are met with different thermal and physical conditions of the snow–ice system. Sound evaluation results and the consistency between OIB and CS2 ice thickness retrieval results, which are subject to different scales, all suggest that point-measured $\alpha$-prediction equation can apply to satellite measurements.

Overall, the developed $\alpha$-based method yields ice thickness and snow depth, without relying on a priori 'uncertain' snow depth information (MW99), which results in uncertainty in the ice thickness retrieval. The proposed method applies to both lidar and radar altimeter data, although lidar-based altimeter data tend to offer relatively more suitable snow depth information with smaller RMSE. We expect to continuously monitor the Arctic scale snow depth and ice thickness by applying the proposed $\alpha$ method to total freeboard observations by the recently launched ICESat-2, using temperature observations from the upcoming

MetOp SG Meteorological Imager (MetImage), the Microwave Imager (MWI) and the proposed Copernicus Imaging Microwave Radiometer (CIMR).

## Appendix A: Physical interpretation of the piecewise linearity between $\alpha$ and $\Delta T_{snow}/\Delta T_{ice}$

The relationship found between $\alpha$ and $\Delta T_{snow}/\Delta T_{ice}$ showed a piecewise linearity, which is almost invariant with the data averaging period. Because the slope change is neither attributable to different data sources nor different data periods, it is likely caused by the physical properties of the snow and ice, as shown in Fig. A1. If the slope change is caused by the snow/ice condition, there will be a significant difference in snow/ice properties between the two parts showing different slopes. Here, we examine the possibility of different physical properties causing the difference in slopes. Through this comparison using buoy data, we may identify important properties that might be responsible for the piecewise linearity.

First, the averages of basic properties available from buoy measurements are compared. They include ice thickness, snow depth, snow–ice interface temperature, ice temperature ($T_{ice} = (T_{as} + T_{si}) / 2$), and so on. The comparison revealed that snow–ice system within the first part ($x \leq x_0$) is found to consist of relatively thicker ice (mean value: 1.84 m), thinner snow (0.29 m), and colder ice (-9.13 °C) while the second part ($x > x_0$) is found to consist of relatively thinner ice (1.10 m), thicker snow (0.46 m), and warmer ice (-5.00 °C). In general, a thicker snow or ice layer exhibits a greater temperature difference from top to bottom of the layer. There is no significant difference between the air–snow interface temperature ($T_{as}$) in the two slope parts.

The thermal conductivities, $k_{snow}$ and $k_{ice}$, are also compared because what connects $\alpha$ and $\Delta T_{snow}/\Delta T_{ice}$ is the ratio of thermal conductivities. Before showing the results, we describe how to calculate $k_{snow}$ and $k_{ice}$. First, the thermal conductivity ratio is calculated from buoy measured variables (i.e. $T_{as}$, $T_{si}$, $T_{iw}$, $h_s$, and $H_i$) using Eq. (3). Because the underlying physics in $k_{snow}$ is significantly more complex, $k_{ice}$ is estimated first, and then $k_{snow}$ is obtained by multiplying the calculated $k_{ice}$ and $k_{snow}/k_{ice}$. To calculate $k_{ice}$, the parameterization of Maykut and Untersteiner (1971), which describes $k_{ice}$ as a function of salinity and temperature, is used.

$$k_{ice} = 2.03 + 0.117 \frac{S_{ice}}{T_{ice}} \tag{A1}$$

Here, $S_{ice}$ and $T_{ice}$ is the salinity (in ppt) and temperature (in Celsius) of sea ice, respectively. For the calculation, $S_{ice}$ is estimated according to the empirical relationship between sea ice thickness and mean salinity from Cox and Weeks (1974) as follows:

$$S_{ice} = \begin{cases} 14.24 - 19.39 H_i, & H_i \leq 0.4\ m \\ 7.88 - 1.59 H_i, & H_i > 0.4\ m \end{cases} \tag{A2}$$

Although Trodahl et al. (2001) reported that $k_{ice}$ depends on depth and temperature; here we do not estimate accurate thermal conductivities but attempt to examine the physical consequences of the total ice layer.

The calculated thermal conductivities are presented in Fig. A2. The calculated $k_{ice}$ ranges from 1.8 W K$^{-1}$ m$^{-1}$ to 2.0 W K$^{-1}$ m$^{-1}$ (left two panels in Fig. A2). These values are consistent with the in-situ measurements by Pringle et al. (2006). The mean values of $k_{ice}$ of the first part (1.96 W K$^{-1}$ m$^{-1}$) and the second part (1.88 W K$^{-1}$ m$^{-1}$) show almost no difference. The calculated $k_{snow}$ ranges from 0.2 W K$^{-1}$ m$^{-1}$ to 1.05 W K$^{-1}$ m$^{-1}$ (right two panels in Fig. A2). This range is consistent with reported values

in Sturm et al. (1997). The first part shows the greater spread in the distribution of $k_{snow}$ compared to the second part. The mean $k_{snow}$ values are 0.44 and 0.27 for the first part and second part, respectively.

As a significant difference is observed in $k_{snow}$, we would like to find a possible reason for this difference. To do so, we should first review the factors determining $k_{snow}$; they are density, temperature, and crystal structure (Sturm et al., 1997). Snow is a mixture of ice particles and air, and air has lower thermal conductivity than ice. Thus, snow with a relatively lower density including a greater portion of air should have relatively lower thermal conductivity. Besides, the thermal conductivity of ice particles depends on the temperature, and the path of heat transfer depends on the crystal structure which describes how the particles are connected. The heat transfer occurs not only by conduction but also by water vapor latent heat transportation and convection through the pore spaces (Sturm et al, 2002), which are hard to quantify explicitly. These two factors are closely related to the temperature gradient (or difference) imposed within the snow layer.

Based on this knowledge, we can infer the condition of the snow layer of the two parts. The relatively higher and varying $k_{snow}$ of the first part would be related to the compaction process resulting in high density, and metamorphic diversity which changes the crystal structure. According to Sturm et al. (2002), the value of $k_{snow}$ of hard wind slab attains up to 0.5 W m$^{-1}$ K$^{-1}$, while that of $k_{snow}$ of depth hoar is below 0.1 W m$^{-1}$ K$^{-1}$. On the other hand, the lower and nearly constant $k_{snow}$ of the second part implies that the snow layer of the second part would consist of fresh and dry snow having relatively lower density and a relatively lower likelihood of experiencing particular metamorphism.

In summary, it is concluded that the physical properties of snow and ice can account for the piecewise linearity, based on the differences in the physical properties between the first and second parts. Especially, the thermal conductivity of the snow, $k_{snow}$, seems to play an important role. Nevertheless, further analysis is required to fully understand this phenomenon.

## Appendix B: Sensitivity test for the proposed method

Here we present results of a sensitivity test for showing how the snow depth and ice thickness retrieval results are dependent on the uncertainties in $\alpha$. To do so, the uncertainty in the snow depth ($\Delta h_s$) due to the $\alpha$ error (i.e. $\Delta\alpha$) and associated ice thickness error ($\Delta H_i$) are estimated. From this sensitivity test, we expect to understand why the simultaneous method for the radar freeboard shows more scattered features than those from the lidar total freeboard.

First, $\Delta h_s$ is defined by the difference of retrieved $h_s$ between with error ($\alpha + \Delta\alpha$) and without error ($\alpha$).

$$\Delta h_s = \begin{cases} h_s(\alpha + \Delta\alpha, F_t) - h_s(\alpha, F_t) & \text{(using } F_t) \\ h_s(\alpha + \Delta\alpha, F_r) - h_s(\alpha, F_r) & \text{(using } F_r) \end{cases} \tag{B1}$$

Then, $\Delta h_s$ can be converted to the error in the ice thickness ($\Delta H_i$) using the following equation derived from Eq. (10).

$$\Delta H_i = \frac{(f\eta_s - 1)\rho_w + \rho_s}{\rho_w - \rho_i} \Delta h_s = \begin{cases} -6.46\Delta h_s & \text{(using } F_t) \\ 3.44\Delta h_s & \text{(using } F_r) \end{cases} \tag{B2}$$

Because $H_i$ and $h_s$ are the combination of freeboard and $\alpha$, as in Eqs. (3), (11) and (12), we only examine the uncertainty with some typical sea ice types. Here physical states for thicker ice (type A), moderate ice (type B), and thinner ice (type C) are chosen, which are summarized in Table B1. Typical values for those three types are shown in the scatterplots of OIB-based ($\alpha^{OIB}$ vs. $F_t^{OIB}$) and of satellite-based ($\alpha^{sat}$ vs. $F_r^{CS2}$) – Fig. B1. It is shown that the majority of data points are located around type B, followed by type A. There seems a very small portion of total samples showing values around type C.

With $\Delta\alpha = \pm0.05$, which is the root mean square difference (RMSD) value between $\alpha^{OIB}$ and $\alpha^{sat}$, $\Delta h_s$ and $\Delta H_i$ are estimated for three ice types. Table B2 summarizing results show that $|\Delta h_s|$ is within 8 cm and it tends to decrease as the ice becomes thinner when the current method is applied to the total freeboard. On the other hand, the use of radar freeboard shows that $|\Delta h_s|$ tends to be more sensitive for the same $\Delta\alpha$. Especially, the sensitivity of type C is the greatest. This is because the denominator of Eq. (12) becomes smaller when $\alpha$ approaches to $\alpha_{crit}$, resulting in an unstable solution. For the ice thickness, $|\Delta H_i|$ is smaller

when the total freeboard is used since $\Delta H_i$ is proportional to $\Delta h_s$. However, the gap between the results from two freeboards has narrowed because $H_i$ from the total freeboard is more sensitive than the radar freeboard to $\Delta h_s$, according to Eq. (B2). The sensitivity characteristics shown here are consistent with the analysis results given in Sect 4.2. Because there is a much smaller number of data points belonging to type C, at least in the data used for this study, the overall sensitivity would likely be in between B and A types.

It is also of importance to ask to what degree of retrievals is successfully yielded. In this study, cases showing $T_{as} > T_{si}$ or retrieved $\alpha \geq \alpha_{crit}$ are considered to be failures. Statistics on success/fail ratio of $\alpha$ retrieval for December–March of 2011–2015 period are provided in Table B3. Overall, the success ratio was over 82% in December–February, while it was reduced to ~74% in March. Most of the failures appear associated with cases showing the temperature inversion (i.e. $T_{as} > T_{si}$), whose areas are shaded with grey in the $\alpha$-distributions of Fig. 8. Those failure areas are generally found around the marginal ice zones and in

the east of Greenland. On the other hand, there was a near-zero failure (0.02% of total pixels) for retrieved $\alpha \geq \alpha_{crit}$. This near-zero failure implies that almost all calculated $\alpha$ meet the satisfactory condition after the removal of cases showing the temperature inversion. It may be concluded that the calculated $\alpha$ appears to be physically reasonable (i.e. $\alpha < \alpha_{crit}$) as long as presumed thermodynamic conditions are met.

**Appendix C: Uncertainty estimation for CS2 retrievals**

Although the sensitivity test regarding uncertainty of satellite derived $\alpha$ has been conducted in Appendix B, the uncertainty of CS2 freeboard measurements and prescribed parameters should be considered as well for the satellite snow depth and ice thickness estimates. To do so, a simple propagation analysis of errors is performed, regarding the uncertainty of satellite products ($\alpha^{sat}$ and $F_r^{CS2}$) and prescribed parameters ($\rho_i$, $\rho_s$, and $f$). Uncertainty due to the variability of $\rho_w$ is neglected (Kurtz and Harbeck, 2017; Hendricks et al., 2016; Ricker et al., 2014). Here we assume that $\alpha^{sat}$ and $F_r^{CS2}$ are not correlated, with no

systematic bias. Such assumption may not be true in the real world. However, it allows us to estimate the retrieval uncertainty

from satellite-derived products, with a certain limit. Uncertainty of ice thickness can be estimated by following Gaussian error propagation equation.

$$\epsilon_{y,total}^2 = \sum_x \epsilon_y(x)^2 \tag{C1}$$

Here, $\varepsilon_{y,total}$ denotes the total uncertainty of retrieved variable $y$ ($h_s$ or $H_i$) and $\varepsilon_y(x)$ denotes the uncertainty of $y$ related to input variable $x$ ($\alpha$, $F_r$, $\rho_i$, $\rho_s$, or $f$). The uncertainties on the right-hand side are obtained by following equation.

$$\epsilon_y(x) = \frac{\partial y}{\partial x}\sigma_x = \lim_{\delta \to 0}\frac{y(x+\delta)-y(x)}{\delta}\sigma_x \tag{C2}$$

Here, $\sigma_x$ denotes the uncertainty of $x$ and $\delta$ is set to be $10^{-6}$ for numerical calculation of the partial derivative using Eqs. (3) and (12). $\sigma_\alpha$ is estimated to be an RMSD value between $\alpha^{OIB}$ and $\alpha^{sat}$. $\sigma_{Fr}$ is given by Kurtz and Harbeck (2017) and $\sigma_f$ is adopted from Armitage and Ridout (2015). Uncertainties of snow/ice densities are from relevant literatures (Alexandrov et al., 2020; Hendricks et al., 2016; Kern and Spreen, 2015; Ricker et al., 2014; Warren et al., 1999). Those values are summarized in Table C1.

Using Eqs. (C1) and (C2), uncertainties of snow depth and ice thickness retrievals can be estimated. Ice thickness uncertainty estimates are presented in Fig. C1. Total uncertainty of ice thickness estimate ranges from 0.8 m to 2.0 m. Generally, $F_r$-related uncertainty in the third row is greater than $\alpha$-related uncertainty in the second row. Snow depth uncertainty estimates are presented in Fig. C2. Total uncertainty of snow depth ranges from 0.04 m to 0.4 m. In the case of the snow depth, $\alpha$-related uncertainty is greater than $F_r$-related uncertainty. Both uncertainties of ice thickness and snow depth are greater for MYI region than FYI region. It is thought that the improvement of accuracy in satellite derived temperatures can reduce the snow depth uncertainty while the improvement of freeboard accuracy can reduce the ice thickness uncertainty. Uncertainties induced from densities and radar penetration factors are found to be relatively smaller than uncertainties related to $\alpha$ and $F_r$ (shown in Fig. S2 and Fig. S3).

**Data availability**

The SHEBA buoy data were obtained from NCAR/EOL (https://doi.org/10.5065/D6KS6PZ7, last access: 14 September 2019) and CRREL IMB buoy data were obtained from the CRREL-Dartmouth Mass Balance Buoy Program (http://imb-crrel-dartmouth.org, last access: 14 September 2019). AASTI-v2 and SIIT data are available upon request to authors. Other data sets were obtained from NSIDC; They are OIB data (https://doi.org/10.5067/G519SHCKWQV6, last access: 10 September 2019), OIB quick look data (https://doi.org/10.5067/GRIXZ91DE0L9, last access: 28 July 2020), CS2 data (https://doi.org/10.5067/96JO0KIFDAS8, last access: 10 September 2019), and SIC data (https://doi.org/10.5067/7Q8HCCWS4I0R, last access: 12 September 2019).

## Author contribution

HS and BJS conceptualized and developed the methodology and HS conducted data analysis and visualization. GD and RTT gave important feedback for the algorithm development and result interpretation. GD provided AASTI data. All of the authors participated in writing the manuscript; HS prepared the original draft under the supervision of BJS and GD, and BJS critically revised the manuscript.

## Competing interests

The authors declare that they have no conflict of interest.

## Acknowledgments

This study has been supported by the Space Core Technology Development Program (NRF-2018M1A3A3A02065661) of the National Research Foundation of Korea. Authors also acknowledge that this study is also supported by the International Network Programme of the Ministry of Higher Education and Science, Denmark (Grant ref. no. 8073-00079B). We appreciate NSIDC for producing and providing the OIB, CS2, and SIC dataset. We also give thanks to CRREL and NCAR/EOL under the sponsorship of the National Science Foundation for providing IMB and SHEBA buoy data. The authors express their sincere thanks to an anonymous reviewer and to Dr. Isobel R. Lawrence for their valuable comments that led to improve the paper.

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

**Table 1.** Information on the measurement sites of buoys whose observations were used in this study.

| Name | | Deployment Location | Ice Type | Initial Snow Depth [m] | Initial Ice Thickness [m] |
|---|---|---|---|---|---|
| CRREL | 2010F | Beaufort Sea | Multi-Year | 0.25 | 1.97 |
| | 2011M | Central Arctic | Multi-Year | 0.07 | 1.67 |
| | 2012G | Central Arctic | First-Year | 0.16 | 1.41 |
| | 2013F | Beaufort Sea | Multi-Year | 0.00 | 1.40 |
| | 2013G | Beaufort Sea | Multi-Year | 0.00 | 1.40 |
| | 2014G | Beaufort Sea | Multi-Year | 0.10 | 1.08 |
| | 2014I | Beaufort Sea | Multi-Year | 0.23 | 1.32 |
| SHEBA | Q2 | Beaufort Sea | Multi-Year | 0.06* | 1.75* |
| | PIT | Beaufort Sea | Multi-Year | 0.12* | 2.01* |
| | BALT | Beaufort Sea | First Year | 0.07* | 1.40* |
| | R4 | Beaufort Sea | Second-Year Ridge | 0.09* | 4.23* |
| | SEA | Beaufort Sea | Ponded Area | 0.10* | 1.54* |

*The initial snow depth and ice thickness of the SHEBA sites are average values of all thickness gauge measurements in the corresponding
site because there was one thermistor string but several thickness gauges in each measurement site


**Table 2.** Coefficients of the regression equation for averaging periods of 1, 7, 15, and 30 days. $a_1$, $b_1$, $a_2$, $b_2$, and $x_0$ are given in Eq. (15).

| Averaging Periods | $a_1$ | $b_1$ | $a_2$ | $b_2$ | $x_0$ |
|---|---|---|---|---|---|
| 1 day | 0.166 | 0.047 | 0.050 | 0.263 | 1.864 |
| 7 days | 0.179 | 0.028 | 0.053 | 0.254 | 1.796 |
| 15 days | 0.180 | 0.034 | 0.029 | 0.339 | 2.022 |
| 30 days | 0.185 | 0.022 | 0.076 | 0.214 | 1.769 |


**Table B1.** The physical state of typical cases of points A, B, and C.

| Type | $H_i$ [m] | $h_s$ [m] | $\alpha$ | $F_t$ [m] | $F_r$ [m] |
|------|-----------|-----------|----------|-----------|-----------|
| A | 3.961 | 0.332 | 0.084 | 0.65 | 0.30 |
| B | 1.646 | 0.123 | 0.075 | 0.26 | 0.13 |
| C | 0.616 | 0.152 | 0.246 | 0.17 | 0.01 |


**Table B2.** Errors of snow depth ($\Delta h_s$) and ice thickness ($\Delta H_i$) for snow depth to ice thickness ratio error ($\Delta\alpha$) of ±0.05.

| | Total Freeboard Method | | Radar Freeboard Method | |
|---|---|---|---|---|
| $\Delta\alpha$ | -0.05 | 0.05 | -0.05 | 0.05 |
| | $\Delta h_s$ (cm) | | | |
| A | -7.502 | 4.903 | -22.417 | 36.719 |
| B | -3.543 | 2.277 | -9.002 | 14.437 |
| C | -0.080 | 0.062 | -9.499 | Retrieval Fail* |
| | $\Delta H_i$ (m) | | | |
| A | 0.485 | -0.317 | -0.771 | 1.264 |
| B | 0.229 | -0.147 | -0.310 | 0.497 |
| C | 0.005 | -0.004 | -0.327 | Retrieval Fail* |

*Retrieval fail occurs if $\alpha + \Delta\alpha > \alpha_{crit}$ ($\alpha_{crit} = 0.291$ for $\rho_s = 320$ kg m$^{-3}$, $\rho_l = 915$ kg m$^{-3}$, $\rho_w = 1024$ kg m$^{-3}$, and $f = 0.84$).

**Table B3.** Statistics of success/fail ratio $\alpha$ retrieval for 2011-2015 winter.

| Year Month | Total Pixels (SIC > 95%) | Success | | Fail ($T_{as} > T_{si}$) | | Fail ($\alpha > \alpha_{crit}$) | |
|---|---|---|---|---|---|---|---|
| 2010 12 | 13879 | 12080 | (87.04%) | 1799 | (12.96%) | 0 | (0.00%) |
| 2011 01 | 16246 | 14004 | (86.20%) | 2242 | (13.80%) | 0 | (0.00%) |
| 2011 02 | 17986 | 14779 | (82.17%) | 3206 | (17.82%) | 1 | (0.01%) |
| 2011 03 | 17610 | 12871 | (73.09%) | 4738 | (26.91%) | 1 | (0.01%) |
| 2011 12 | 13915 | 11405 | (81.96%) | 2510 | (18.04%) | 0 | (0.00%) |
| 2012 01 | 16812 | 13765 | (81.88%) | 3047 | (18.12%) | 0 | (0.00%) |
| 2012 02 | 17528 | 14131 | (80.62%) | 3397 | (19.38%) | 0 | (0.00%) |
| 2012 03 | 18741 | 13586 | (72.49%) | 5155 | (27.51%) | 0 | (0.00%) |
| 2012 12 | 14059 | 11144 | (79.27%) | 2915 | (20.73%) | 0 | (0.00%) |
| 2013 01 | 16413 | 13510 | (82.31%) | 2903 | (17.69%) | 0 | (0.00%) |
| 2013 02 | 18640 | 15526 | (83.29%) | 3114 | (16.71%) | 0 | (0.00%) |
| 2013 03 | 19078 | 14134 | (74.09%) | 4944 | (25.91%) | 0 | (0.00%) |
| 2013 12 | 14515 | 12071 | (83.16%) | 2444 | (16.84%) | 0 | (0.00%) |
| 2014 01 | 16880 | 14201 | (84.13%) | 2678 | (15.86%) | 1 | (0.01%) |
| 2014 02 | 16987 | 14731 | (86.72%) | 2247 | (13.23%) | 9 | (0.05%) |
| 2014 03 | 17699 | 13300 | (75.15%) | 4391 | (24.81%) | 8 | (0.05%) |
| 2014 12 | 14071 | 11119 | (79.02%) | 2952 | (20.98%) | 0 | (0.00%) |
| 2015 01 | 17008 | 15095 | (88.75%) | 1913 | (11.25%) | 0 | (0.00%) |
| 2015 02 | 18076 | 15907 | (88.00%) | 2169 | (12.00%) | 0 | (0.00%) |
| 2015 03 | 17618 | 14042 | (79.70%) | 3576 | (20.30%) | 0 | (0.00%) |
| December | 70439 | 57819 | (82.08%) | 12620 | (17.92%) | 0 | (0.00%) |
| January | 83359 | 70575 | (84.66%) | 12783 | (15.33%) | 1 | (0.00%) |
| February | 89217 | 75074 | (84.15%) | 14133 | (15.84%) | 10 | (0.01%) |
| March | 90746 | 67933 | (74.86%) | 22804 | (25.13%) | 9 | (0.01%) |

$\alpha_{crit}$=0.291 for $\rho_s$=320 kg m$^{-3}$, $\rho_i$=915 kg m$^{-3}$, $\rho_w$=1024 kg m$^{-3}$, and $f$=0.84.


**Table C1.** Values and uncertainties of input variables for uncertainty estimation.

|  | $\alpha$ | $F_r$ [m] | $\rho_i$ [kg m$^{-3}$] | $\rho_s$ [kg m$^{-3}$] | $f$ |
|---|---|---|---|---|---|
| Value | $\alpha^{sat}$ | $F_r^{CS2}$ | 915 | 320 | 0.84 |
| Uncertainty | 0.05 | 0.065 | 20 | 50 | 0.04 |

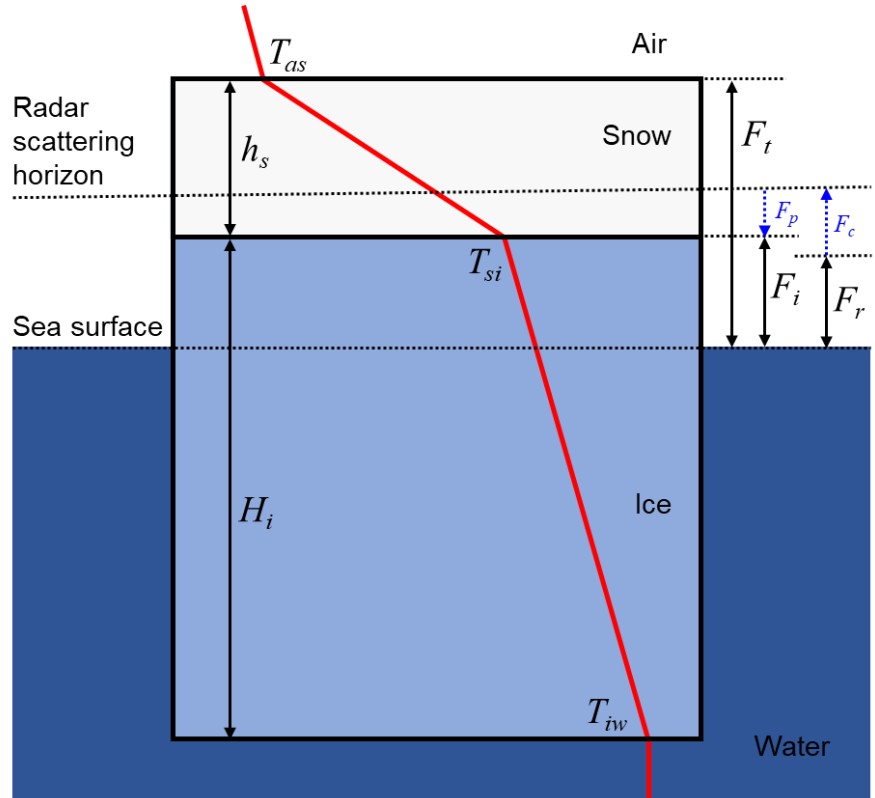

**Figure 1.** Schematic diagram of a typical snow–ice system during the winter. Snow depth ($h_s$), ice thickness ($H_i$), total freeboard ($F_t$), radar freeboard ($F_r$), and ice freeboard ($F_i$) are indicated. Correction terms regarding the wave propagation speed change in snow layer ($F_c$) and the displacement of the scattering horizon from the ice surface ($F_p$) are indicated by blue arrows. The red line denotes a typical temperature profile with air–snow interface temperature ($T_{as}$), snow–ice interface temperature ($T_{si}$), and ice–water interface temperature ($T_{iw}$).

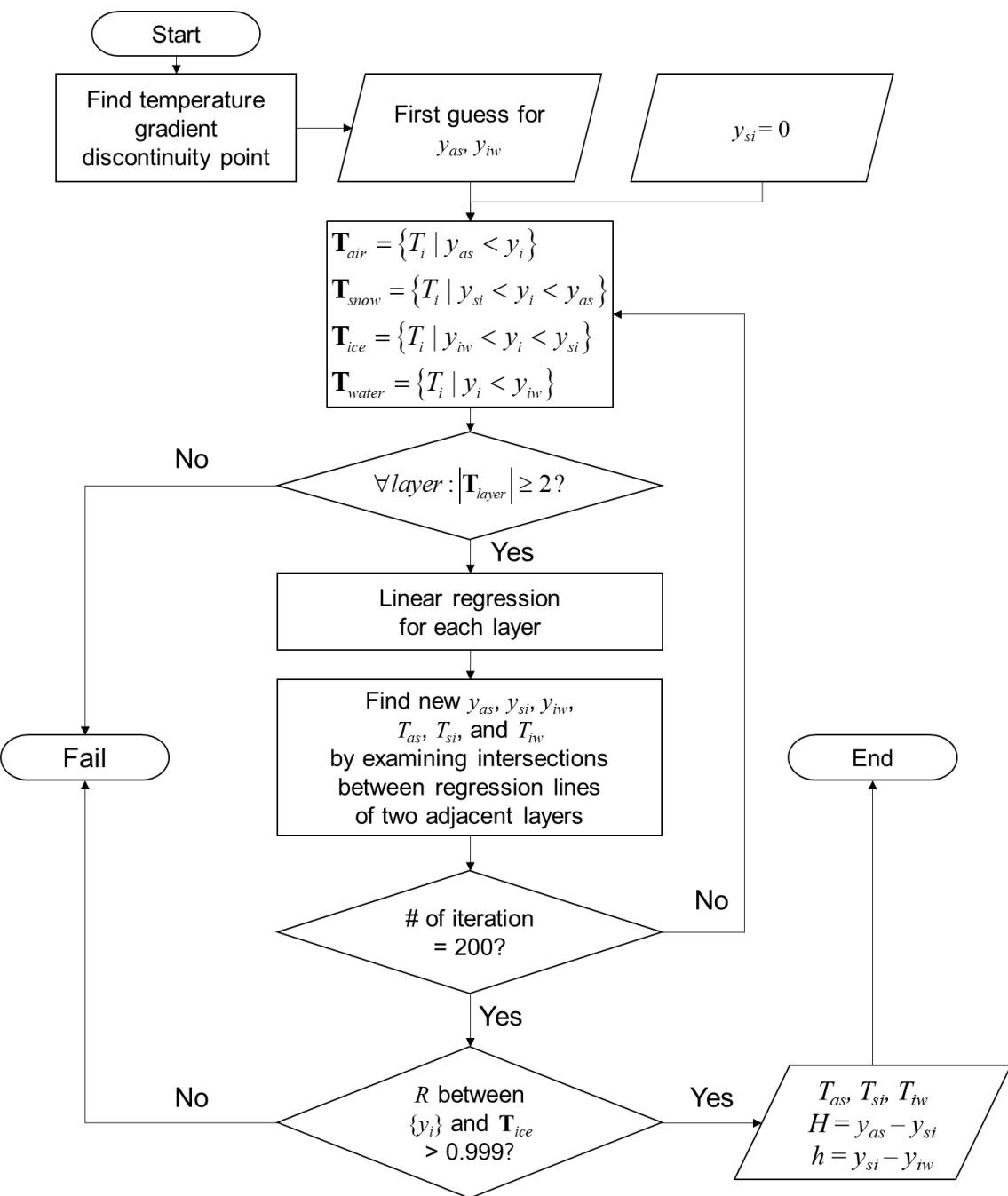


**Figure 2.** The flow chart of the interface searching algorithm. $y_i$ and $T_i$ denote the position and temperature of a data point in the temperature profile. $y_{as}$, $y_{si}$, and $y_{iw}$ denote the position of the interfaces, and $\mathbf{T}_{layer}$ denotes a set of temperature data points.

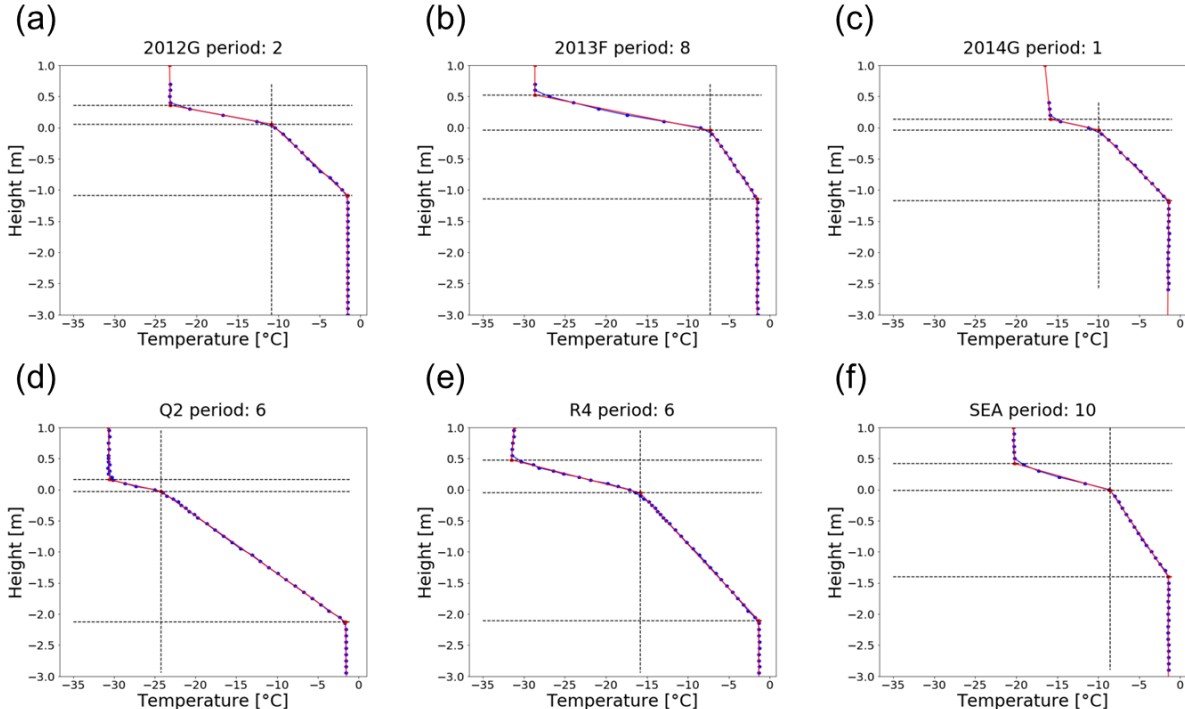

**Figure 3.** Examples of interface searching results with an averaging period of 15 days: (a) 2012G period 2, (b) 2013F period 8, (c) 2014G period 1, (d) Q2 period 6, (e) R4 period 6, and (f) SEA period 10. The period number indicates the sequential 15-day period from November 1 (e.g. 'period: 2' denotes a time-averaging period of November 16th to November 30th). Blue dots are buoy-measured temperature profiles and red lines are regression lines. Black dashed lines indicate the intersections between adjacent regression lines.

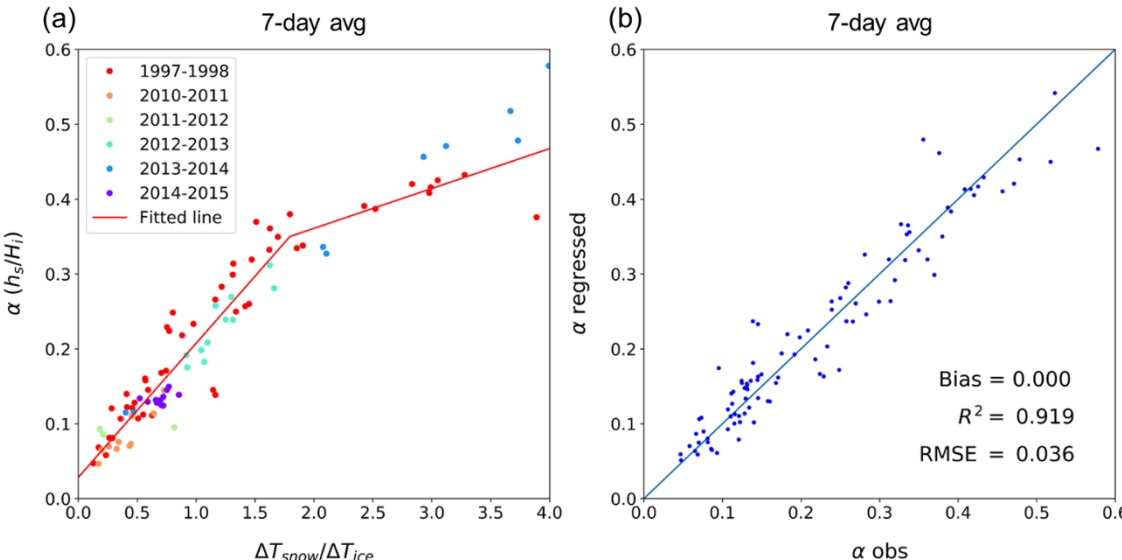


**Figure 4.** (a) Scatterplots of the temperature difference ratio of the snow and ice layer ($\Delta T_{snow}/\Delta T_{ice}$) and the snow–ice thickness ratio ($\alpha$). Color denotes the collected year of buoy data. The red lines are the regression lines (defined in Eq. (15)). (b) The scatter plot of observed and regressed $\alpha$.

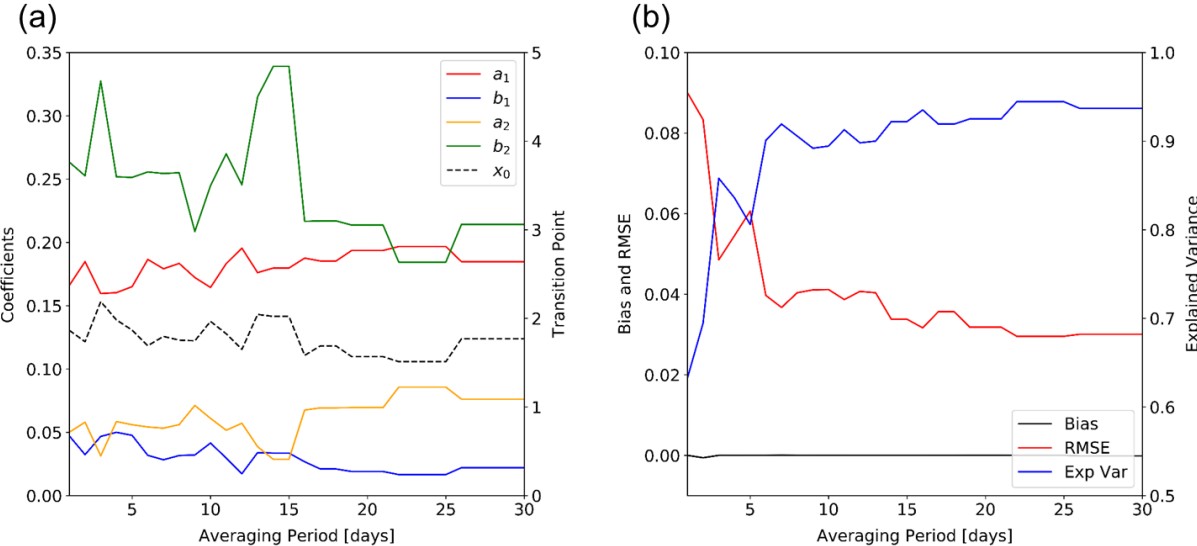


**Figure 5.** (a) The regression coefficients ($a_1$, $b_1$, $a_2$, $b_2$) in Eq. (15) and (b) the error statistics of the regression with averaging periods from 1 to 30 days.

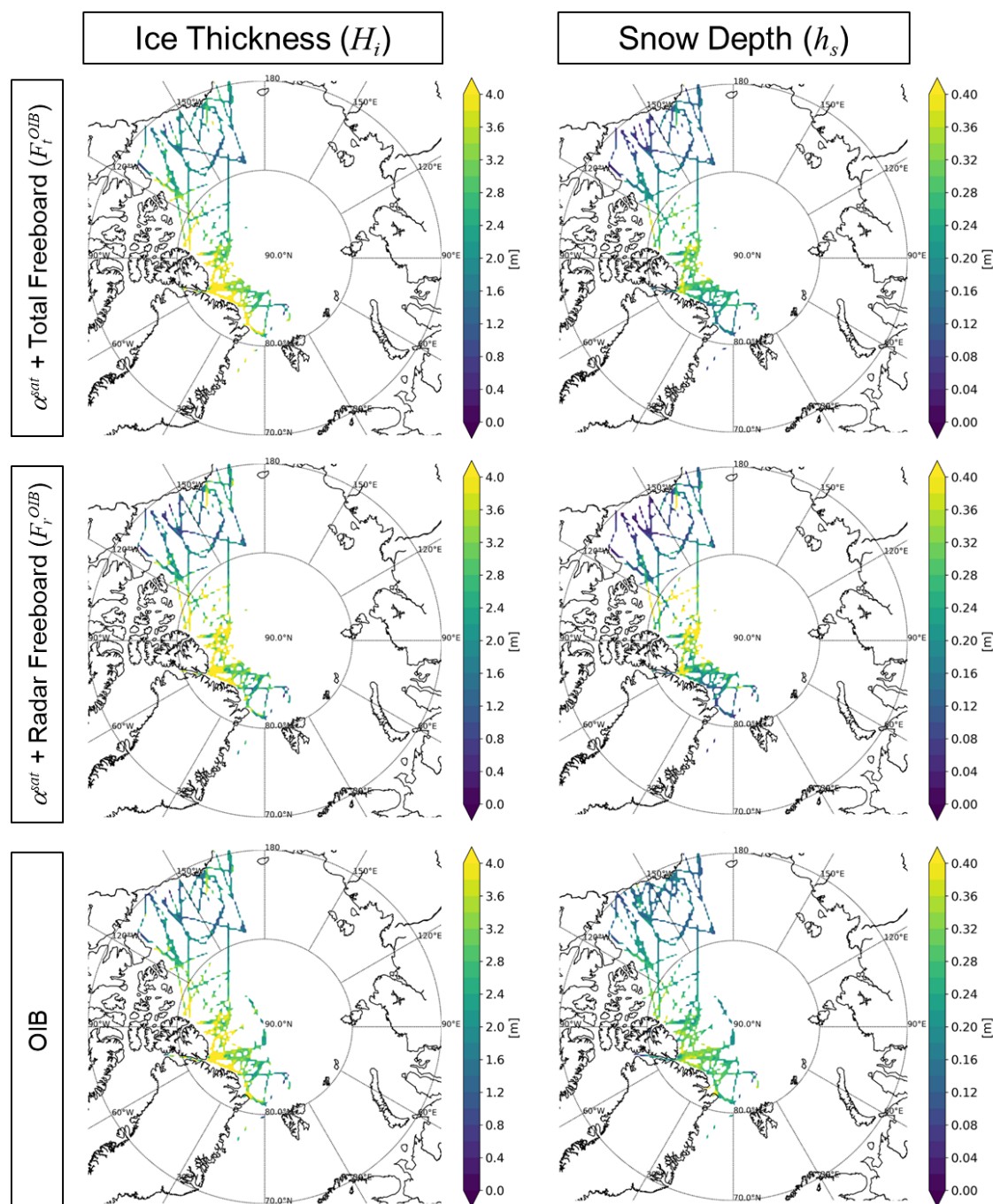

**Figure 6.** Simultaneously retrieved ice thickness and snow depth from OIB total/radar freeboard in March of the 2011–2015 period. Corresponding OIB products are at the bottom.

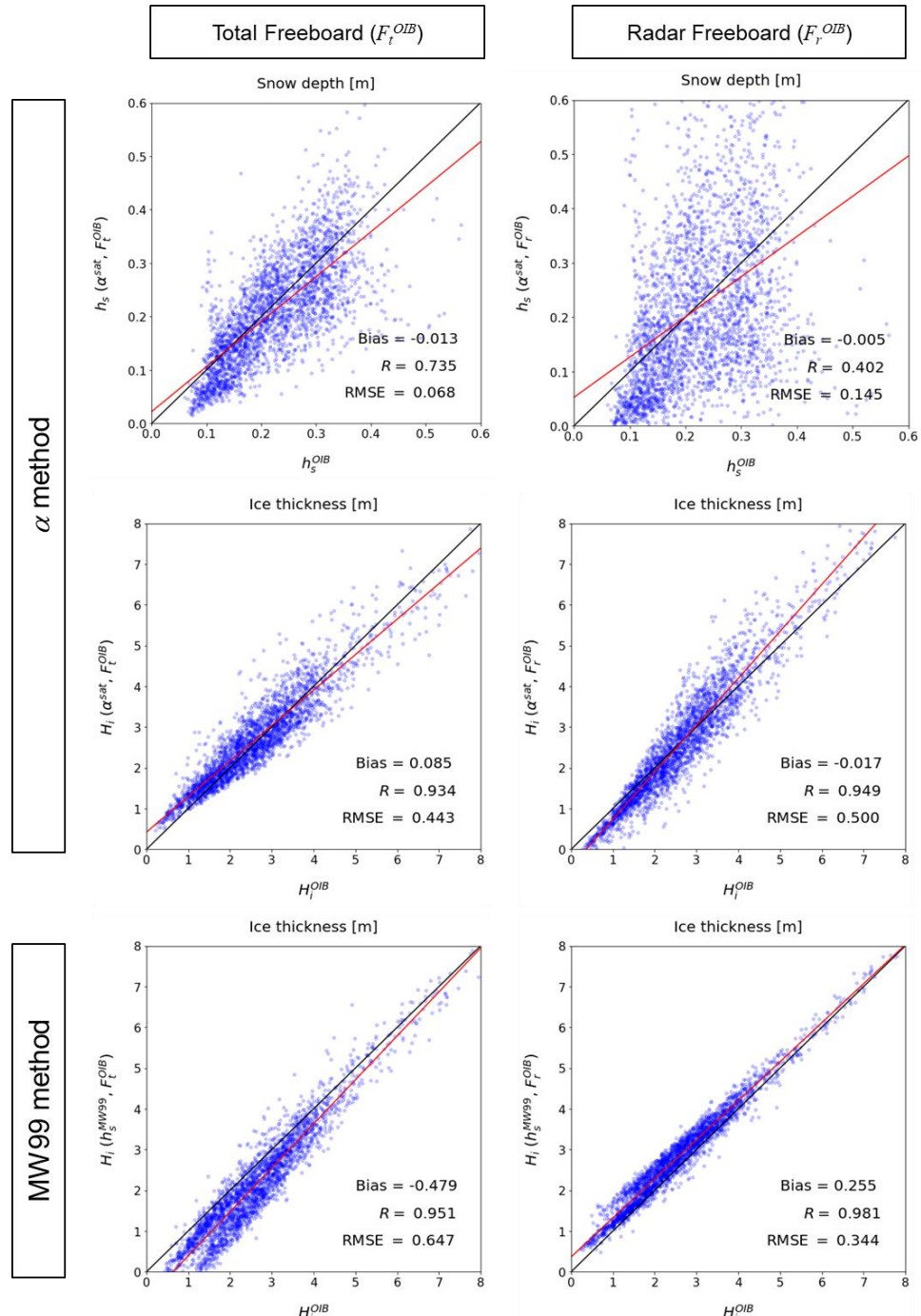

**Figure 7.** Scatter plots between OIB products and the simultaneously retrieved snow depth and ice thickness from OIB total/radar freeboards
during the March 2011–2015 period. Corresponding ice thicknesses estimated from MW99 are in the third row. The red lines are linear
regression lines.

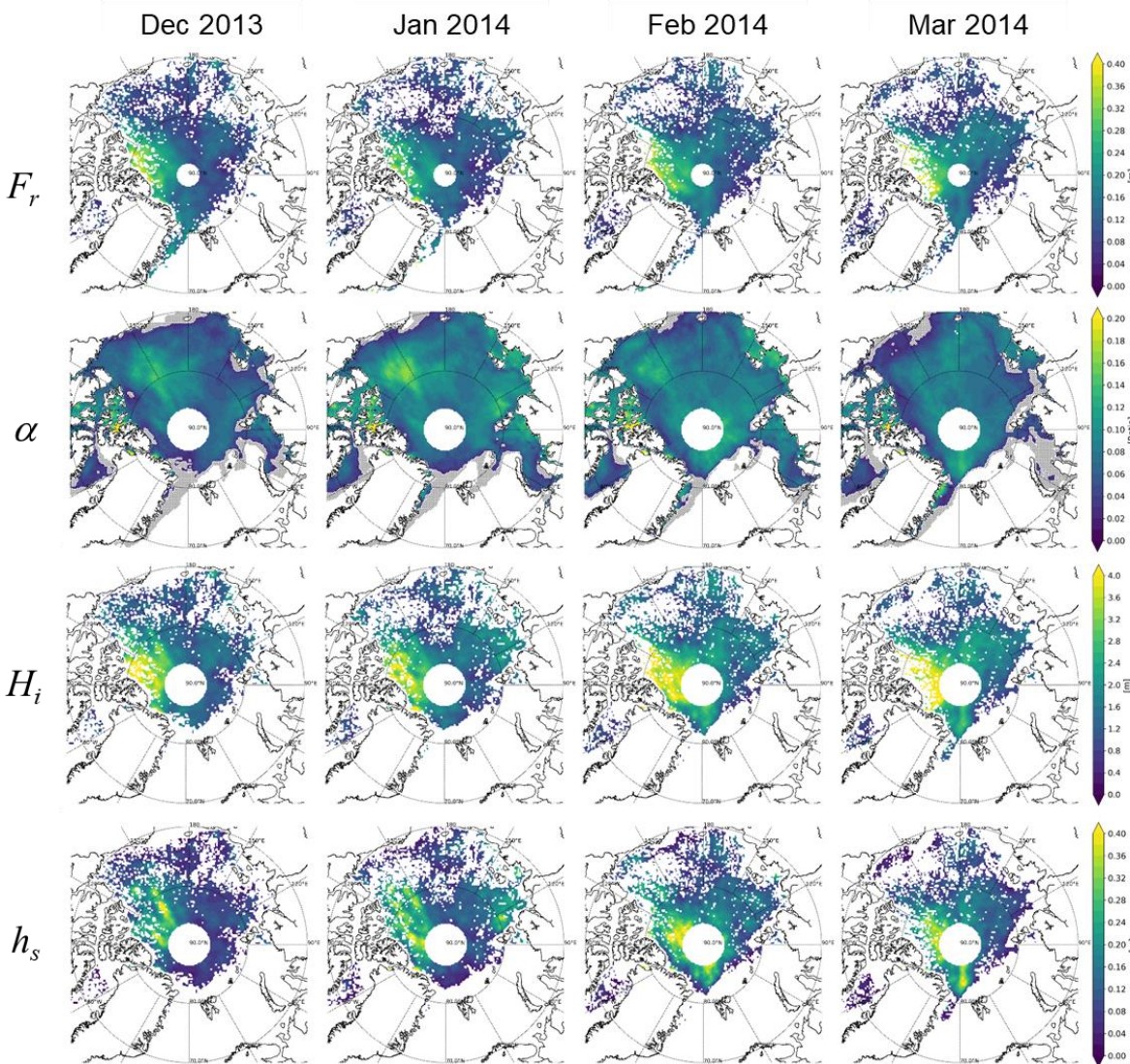

**Figure 8.** Geographical distributions of observed CS2 radar freeboard ($F_r$) and estimated snow–ice thickness ratio ($\alpha$), ice thickness ($H_i$), and snow depth ($h_s$) from December 2013 to March 2014. Grey areas in the second row denote where $\alpha$ retrieval failed because $T_{as}$ is warmer than $T_{si}$.

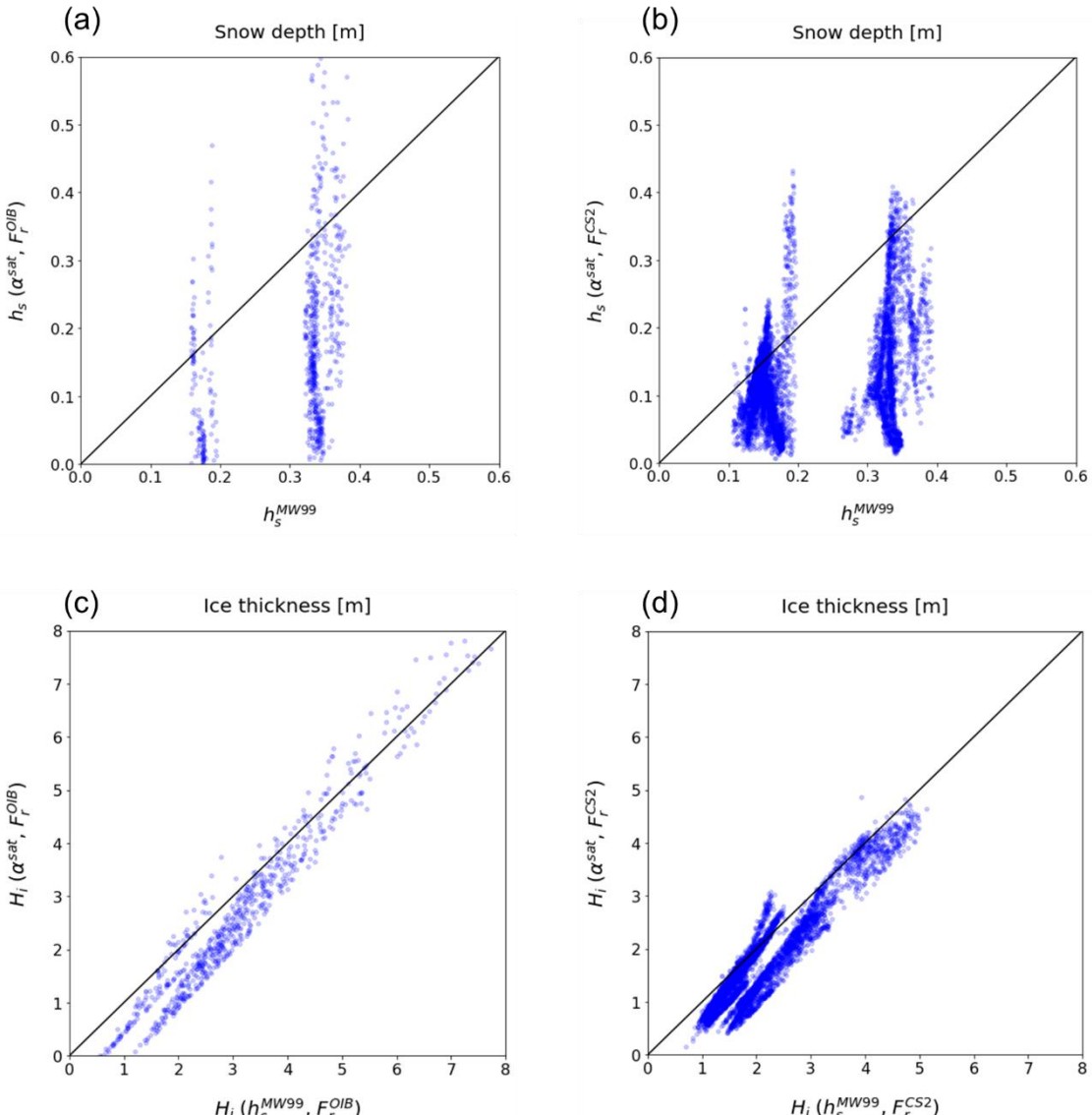

**Figure 9.** Comparison of simultaneous retrieved snow depth and ice thickness to those from the MW99 method. (a) Snow depth from OIB radar freeboard, (b) snow depth from CS2 radar freeboard, (c) ice thickness from OIB radar freeboard, and (d) ice thickness from CS2 radar freeboard.

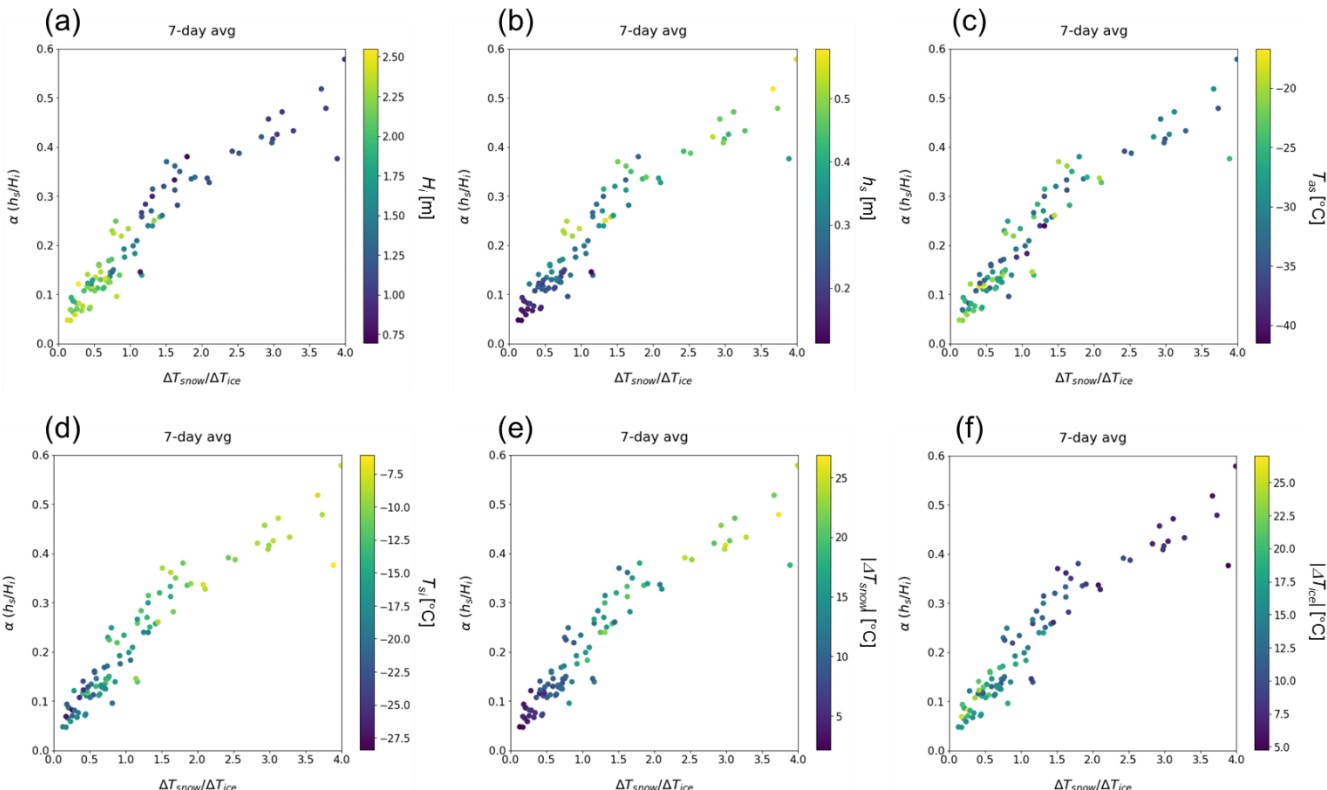

**Figure A1.** Distribution of physical variables on scatterplots of the temperature difference ratio of snow and ice layer ($\Delta T_{snow}/\Delta T_{ice}$) and the snow–ice thickness ratio ($\alpha$). Color denotes the value of physical variables: (a) ice thickness ($H_i$), (b) snow depth ($h_s$), (c) air–snow interface temperature ($T_{as}$), (d) snow–ice interface temperature ($T_{si}$), (e) temperature difference within snow layer ($|\Delta T_{snow}|$), and (f) temperature difference within ice layer ($|\Delta T_{ice}|$).

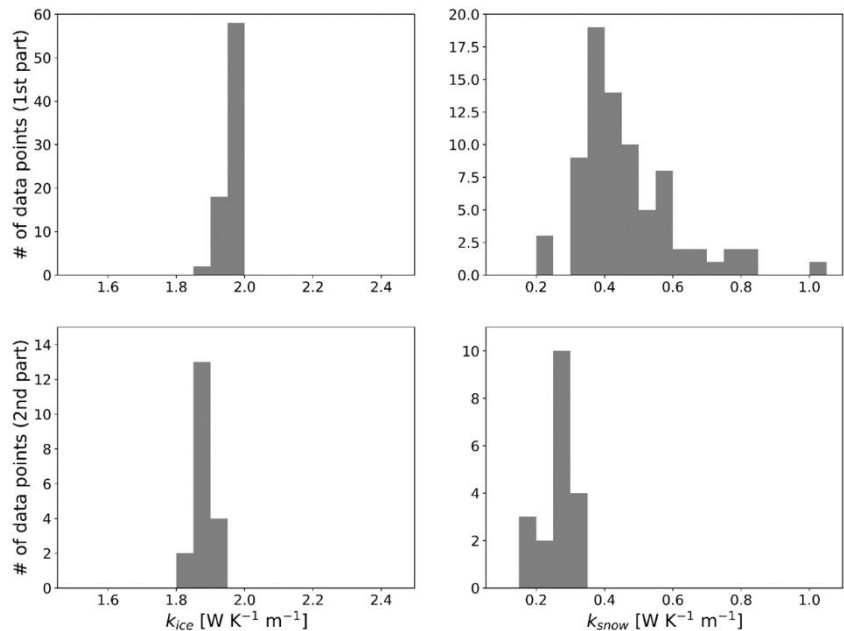


**Figure A2.** Histogram of estimated (left column) $k_{ice}$ and (right column) $k_{snow}$. The top and bottom row denote the first and the second part, respectively. The size of the bins is 0.05 W K$^{-1}$ m$^{-1}$.

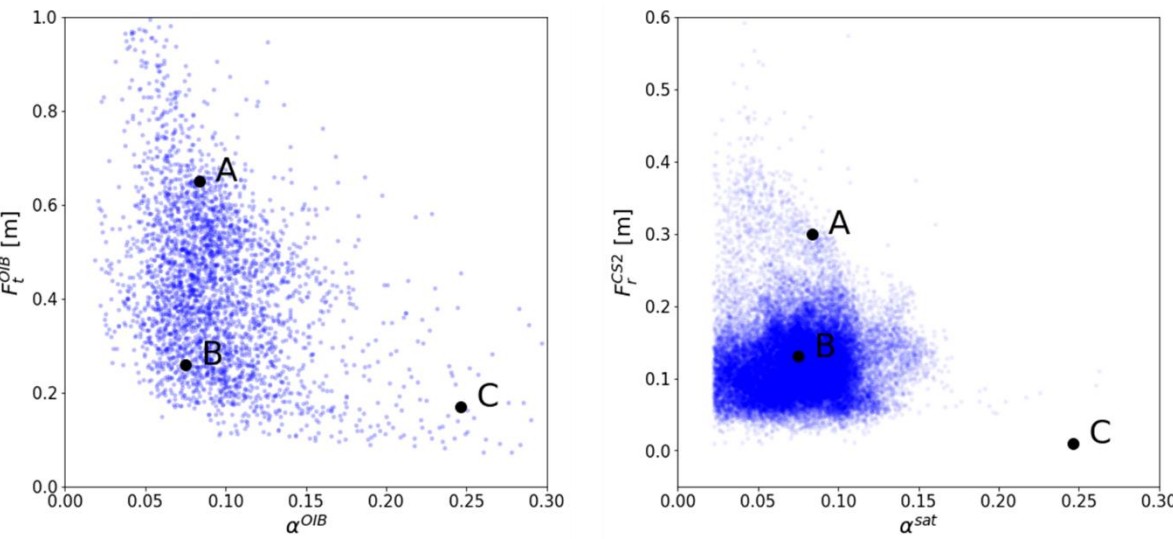

**Figure B1.** Locations of physical states for typical types (A, B, C) on the freeboard-thickness ratio space. Blue dots are from (left) OIB data and (right) retrieved thickness ratio and CS2 radar freeboard.

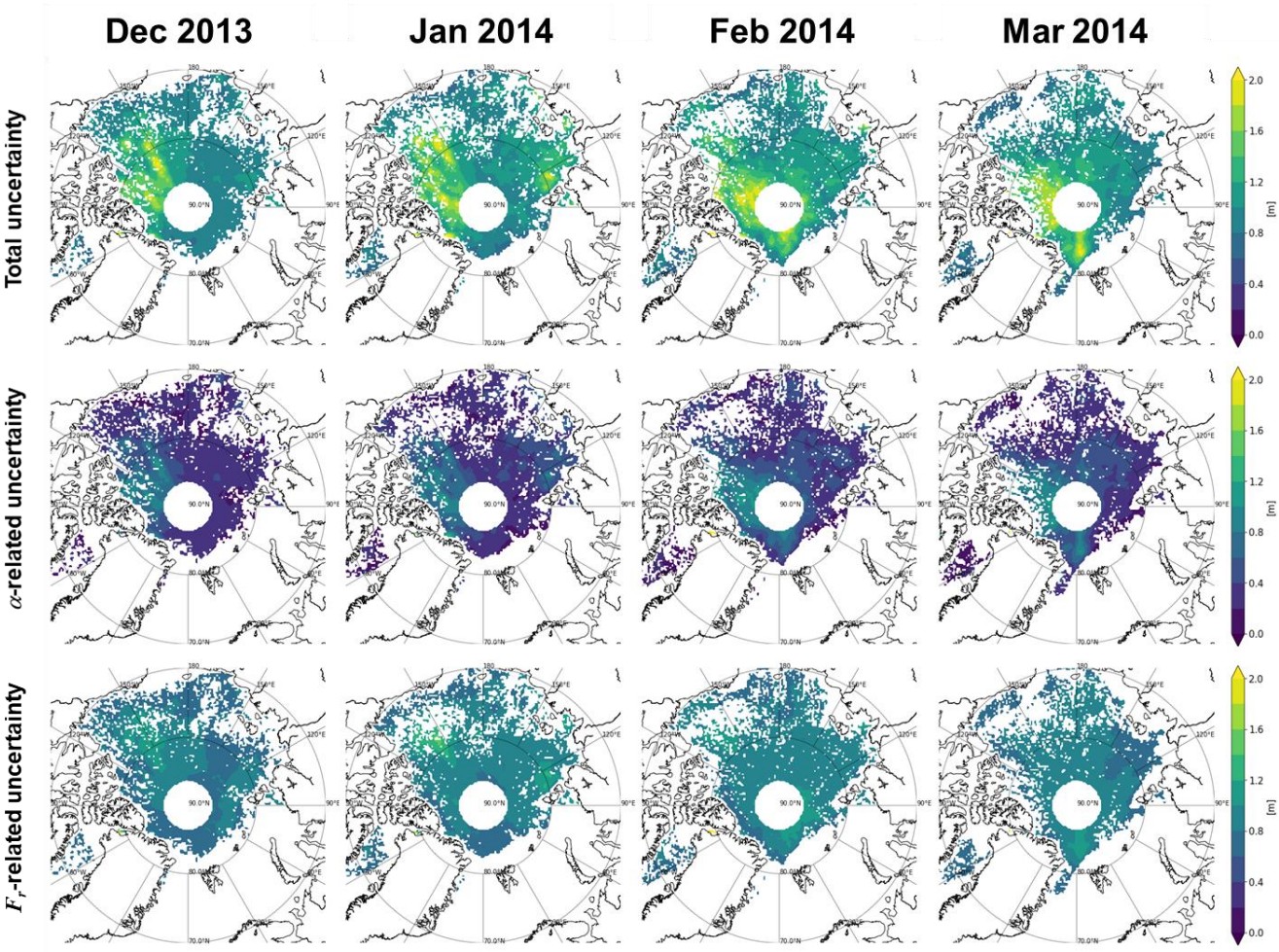

**Figure C1.** Geographical distributions of sea ice thickness uncertainty: (first row) total uncertainty, (second row) $\alpha$-related uncertainty, and (third row) $F_r$–related uncertainty for the period from December 2013 to March 2014.

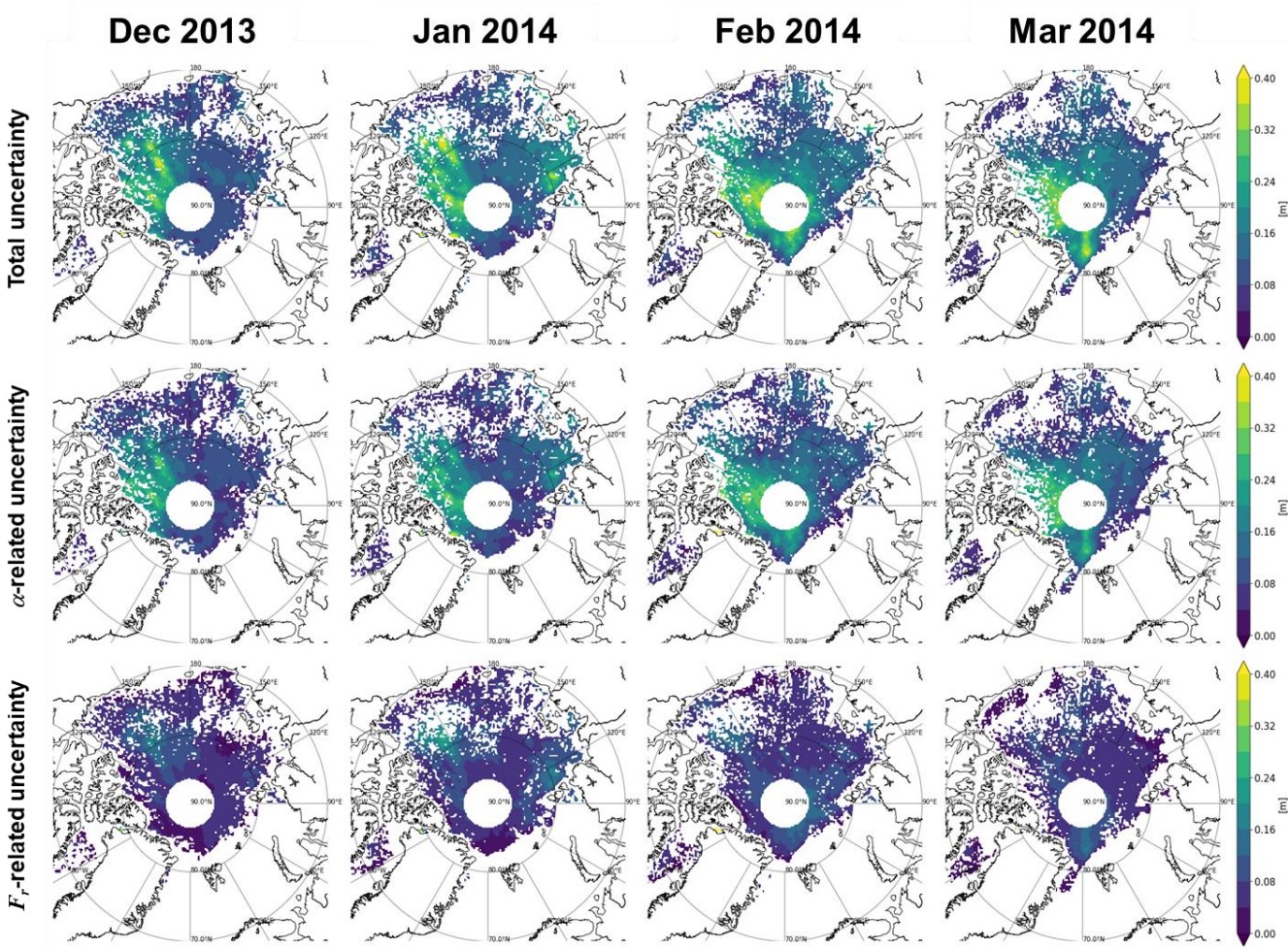

**Figure C2.** Geographical distributions of snow depth uncertainty: (first row) total uncertainty, (second row) $\alpha$-related uncertainty, and (third row) $F_r$-related uncertainty for the period from December 2013 to March 2014.