# Peer review of "Simultaneous estimation of wintertime sea ice thickness and snow depth from space-borne freeboard measurements"

_The Cryosphere, 2020_

## Referee Comment (RC1) · Anonymous Referee #1 · 27 Apr 2020

**Review of: Estimating the snow depth, the snow–ice interface temperature, and the effective temperature of Arctic sea ice using Advanced Microwave Scanning Radiometer 2 and ice mass balance buoy data**

**1 Introduction**

**1.1 Summary of Review**

This manuscript addresses the issue of converting satellite radar ranging data to sea ice thickness estimates, while simultaneously retrieving the depth of the overlying snow. This is done through a novel linear mapping of the temperature gradients in the snow and ice to the thickness values of the materials.

However the manuscript does contain a technical issue which affects its application to CryoSat-2 data (described in Sect. 2.2), which must be remedied before publication. In short, the freeboard data used in the study have been calculated using the modified Warren climatology, so any thickness estimates resulting from them cannot be independent of that climatology.

Otherwise, this manuscript contains a novel approach to estimating the depth of snow on sea ice, and the authors show that the related thickness values match OIB thickness values more closely than those resulting from use of the modified Warren climatology.

I believe that if the issues presented in this review can be resolved, subsequent publication of this manuscript would be of significant interest to the sea ice community.

**1.2 Summary of the Technique Presented**

Sea ice thickness is conventionally estimated from ice freeboard as such (using the notation of this paper):

$$H = \frac{h_f \rho_w + h \rho_s}{\rho_w - \rho_i} \tag{1}$$

Where $H$ is sea ice thickness, $h_f$ the height of the snow ice interface above the waterline, and $h$ the depth of the overlying snow. $\rho$ values represent the densities of overlying snow, water and ice. From the above, it can be seen that for $h_f$ to be converted to $H$, $h$ must be known (as well as all the density values). Unfortunately despite a major research effort to map the spatial distribution of $h$, it remains highly uncertain.

One of the novelties of this paper is the authors' framing of the problem thus:

$$H = h_f \frac{\rho_w}{\rho_w - \rho_i - (h/H)\rho_s} \tag{2}$$

This framing allows $h_f$ to be converted to $H$ without direct knowledge of $h$, instead it is sufficient just to know the ratio $h/H$ - this is referred to as $\alpha$. The authors then show that this formulation is useful, by estimating (h/H) in a Pan-Arctic way using satellite-measured temperatures of the air-snow and snow-ice interfaces.

A byproduct of this approach is that snow depth ($h$) is also trivially extracted:

$$h = h_f \frac{\rho_w}{\rho_w - \rho_i - (h/H)\rho_s} \times (h/H) \tag{3}$$

[Figure]

Figure 1: Graph of $H, h$ space for a case with typical values of densities, snow depth and ice thickness

It should be noted that this approach of 'simultaneously' working out $h$ and $H$ is mathematically equivalent to first working out $h$ using Eq. (3) and substituting the result into Eq. (1).

The method can be further understood graphically in $h, H$ space, as in Figure 1. When the ice freeboard is known, the corresponding snow depth ($h$) and ice thickness ($H$) lie unconstrained in $h, H$ space along the blue line described by Eq. (1).

Thickness is then traditionally determined by a priori knowledge of the snow depth ($h_0$) - this is depicted in the left hand plot by the intersection of the green line (known snow depth) with the blue line.

This paper introduces a different relationship to constrain sea ice thickness, depicted on the right, revolving around the empirical parameter $\alpha$. By finding the intersection with the red ($\alpha$) line and the blue line, the authors are able to simultaneously derive $h$ and $H$.

**2 Major Points**

**2.1 Difference between radar freeboard and scattering horizon height**

L22: The statement "altimeters ... measure sea ice freeboard" is only approximately correct in the case of radar altimeters.

The instrument on board CS2 measures a time of flight, which can be related to the height of some radar scattering horizon only where no snow lies in between the scattering horizon and the instrument. When snow is in between and fully penetrated by the radar, the radar range to the scattering surface is overestimated due to slower pulse propagation in the overlying snow. Correcting for this and estimating the height of the ice-snow interface requires knowledge of the overlying snow (Mallett et al., 2020).

This issue surfaces again when the authors identify the radar freeboard as the height from the sea surface to the radar scattering horizon in L28. This is only the case for bare ice. Where overlying snow is present and fully penetrated, the radar freeboard is a finite distance below the ice freeboard (the assumed scattering surface). In the freeboard product used in this manuscript (Kurtz et al., 2014) this displacement is $h_s(1 - c_s/c)$.

This is relevant to Fig. 1, where $h_{rf}$ is depicted as being above the ice freeboard. While it may be true that the radar scattering horizon is above the snow-ice interface, in products that assume full radar penetration of the snowpack the radar freeboard is lower than the ice freeboard. Theoretically for total radar penetration and a freeboard depressed to near the water by snow, the radar freeboard can be below the waterline (while the ice freeboard and scattering horizon are above).

**2.2 The freeboard product used by the authors has been created with mW99**

In the final sentence of the abstract, the authors state:

> "In conclusion, the developed $\alpha$-based method has the capacity to derive ice thickness and snow depth, without relying on the snow depth information as input to the buoyancy equation for converting freeboard to ice thickness."

However, the method presented here works directly from ice freeboard data which can only be derived by relying on snow depth information (Sect. 5.1 & Eq. 15 of Kurtz et al., 2014).

I feel that what the authors would like to present is a way to convert *radar freeboards* to ice thickness without relying on snow depth data, and this should be done before publication. I think it is possible for the authors to adapt their processing chain to deal with this, although it may complicate things.

**2.3 Uncertainty Analysis**

The authors state in their Discussion and Conclusions section:

> Overall, the developed $\alpha$-based method yields ice thickness and snow depth, without relying on a priori 'uncertain' snow depth information, which results in uncertainty in the ice thickness retrieval.

They are of course correct to identify that uncertainty in snow depth leads to uncertainty in the ice thickness retrieval. To avoid having to quantify snow depth, they instead rely on a parameter equal to $h/H$, which they empirically derive from the temperature of the air-snow and ice-snow interfaces.

Clearly there is significant uncertainty in the value of $\alpha$, and the authors should try to quantify how this propagates through into uncertainty in ice thickness. It's possible that their $\alpha$ parameter is more uncertain than other published data for $h$, and if so this method will deliver lower quality estimates of $H$ than the traditional method.

It seems (looking at Fig. 1 of this review) that a given error in $\alpha$ would have a more serious impact on $H$ than the same error in $h$, because the gradients of the lines are much more similar in on the left panel of Fig. 1 than on the right. This issue scales with the alpha parameter (i.e. as the freeboard goes down), and at high $\alpha$ very small uncertainties in alpha will lead to large uncertainties in $H$.

As alpha becomes so large that the freeboard tends to zero (not that uncommon in the Atlantic sector of the Arctic), the method seems to lose its usefulness, whereas the traditional method continues to function. That is to say in the case of near-zero freeboard, the traditional method still provides an estimate of $H$, but that proposed by the authors does not (see Eq. 7 as $h_f \rightarrow 0$ ).

This is addressed in L 161/162, where a critical value is given for alpha, and it is explained that for alpha above this value data are not produced. How often does this occur? And what is the effect on $H$ of small errors in alpha just below this critical threshold?

**3 Minor Points**

**3.1 General**

- L21: The authors should consider directing the reader to Laxon et al. (2003) when illustrating that thickness has been estimated for nearly two decades.

- When discussing studies indicating the height difference between the scattering horizon and the snow-ice interface, the authors should consider directing the reader to Nandan et al. (2017) and Willatt et al. (2010, 2011).

- L62 & 64: Define RTM before using the acronym.

- The font sizes of some annotations to Figure 3 should be increased so as to be legible and comparable to the (a), (b), (c) lettering.

- In Fig. 2, the box that reads 'Find temperature discontinuity point'. It is my understanding that the temperature is continuous (but not a smooth function), and therefore it has no discontinuities (but its gradient does). Should this box then read 'Find temperature gradient discontinuity point'?

- I think the notation of $H$ and $h$ in combination with $h_f$, $h_{rf}$ and $h_{tf}$ is confusing to the casual reader. For instance, the fact that $h$ and $h_f$ look so similar but are in fact unrelated confused me initially. Even changing $H \to H_{ice}$ and $h \to h_{snow}$ would clarify this.

**3.2 Validation of $H$ against OIB Data**

The authors are able to create two products from freeboard data obtained by OIB and CryoSat-2, one for snow depth ($h$), and one for ice thickness ($H$). They then rightly try to assess the quality of these data products against other datasets, namely the OIB snow depth and ice thickness data.

There are at least five algorithms published to process the raw OIB radar returns into along-track snow depth data, and they produce a spread in the mean snow depth (Kwok et al., 2017). 'Validation' of a model or data implies comparison to true or certain values, and it is unclear which OIB snow depth product (if any) represents *the truth*. This limits the strength of the validation exercise. Nonetheless, I understand that OIB snow depth values have historically been taken as *the truth* in published work so this is a perhaps not a big issue. It might also be argued that the spread of different OIB data is sufficiently small relative to other methods of snow depth estimation to allow OIB to approximate the truth for validation purposes.

I feel that there is however a more significant issue with the authors' claims to have 'validated' their ice thickness data against OIB ice thickness data ($H_{OIB}$). OIB aircraft instruments do not measure thickness ($H_{true}$) directly, but instead estimate it based on freeboard, snow depth, snow density and ice density values. As such, OIB thickness data (while likely to be the most accurate data on $H_{true}$ outside of in-situ measurement), undoubtedly suffer from biases involving snow depth, snow density and ice density, and therefore should not be mistaken for $H_{true}$.

The technique for determining $H_{OIB}$ is very similar to that presented in this manuscript: the authors use identical freeboard, snow density and ice density values to estimate thickness with the hydrostatic equilibrium assumption. Given these similarities, comparing the thickness estimates in this paper with OIB thickness estimates doesn't really qualify as independent validation.

It seems more like the exercise of comparing $H$ estimates is in fact comparing the novel snow depth estimates with OIB snow depths (Fig 7 top row; a valuable analysis), and then investigating how that singular difference propagates into sea ice thickness estimates.

I suspect that the strong agreement between the two datasets presented in the middle row of Fig. 7. is largely a result of the identical radar freeboards and geophysical parameters used in each processing chain. After all, much of sea ice thickness is determined by radar freeboard information, independent of snow data. The fact that the 'simultaneous' method matches $H_{OIB}$ data more closely than MW99 is therefore evidence that the snow depth product produced by the 'simultaneous' method is closer to OIB snow depths than MW99 (because everything else is equal).

I think it is perfectly reasonable (and in fact expected) to compare $H$ estimates from the new method with $H_{OIB}$. However, I think this should be presented as a 'comparison with' or 'evaluation against' OIB data, rather that implying that the new data are being validated against some true value. It is also an understandable bit of reasoning to say that values which are closer to $H_{OIB}$ are likely to be closer to $H_{true}$, but if this assumption is made it should be stated explicitly.

**3.3 Limitations of Other Data**

L66 - 69:

> Other approaches worth mentioning are snow depth retrieval using dual-frequency altimetry (Guerreiro et al., 2016; Lawrence et al., 2018, Kwok and Markus, 2018), snow on sea ice model accumulating snowfall from reanalysis (Petty et al., 2018), multilinear regression (Kilic et al., 2019), and the neural network approach (Braakmann-Folgmann and Donlon, 2019). However, these methods do not satisfactorily meet the criteria required for freeboard to ice conversion over the entire Arctic Ocean basin scale or multi-year time scale.

The approach of Guerreiro et al. (2016) and Lawrence et al. (2018) are limited latitudinally by the AltiKa orbital inclination and Lawrence et al. (2018) additionally through calibration with OIB which only operates in Spring. As the authors identify, they are limited in spatial or temporal extent.

While there are limitations to the data products of Petty et al. (2018), Kilic et al. (2019) and Braakmann-Folgmann and Donlon (2019), it's not obvious that these can be characterised by failure to cover the entire basin on a multiyear timescale. As such, the statement on L69 that they do not satisfactorily meet these criteria should be clarified.

**3.4 Rainbow Color Schemes**

Where possible, authors should avoid presenting continuous data with 'rainbow' color schemes as in Figures 6 & 8. This is because (among other reasons) the scheme tends to imply sharp transitions in the data where they do not exist (Borland and Taylor, 2007). Alternatives for geoscientists are given by Light and Bartlein (2004), Stauffer et al. (2015) and Thyng et al. (2016).

**References**

Borland, D. and Taylor, R. M.: Rainbow color map (still) considered harmful, IEEE Computer Graphics and Applications, 27, 14–17, https://doi.org/10.1109/MCG.2007.323435, 2007.

Braakmann-Folgmann, A. and Donlon, C.: Estimating snow depth on Arctic sea ice using satellite microwave radiometry and a neural network, The Cryosphere, 13, 2421–2438, https://doi.org/10.5194/tc-13-2421-2019, URL https://www.the-cryosphere.net/13/2421/2019/, 2019.

Guerreiro, K., Fleury, S., Zakharova, E., Rémy, F., and Kouraev, A.: Potential for estimation of snow depth on Arctic sea ice from CryoSat-2 and SARAL/AltiKa missions, Remote Sensing of Environment, 186, 339–349, https://doi.org/10.1016/j.rse.2016.07.013, URL http://dx.doi.org/10.1016/j.rse.2016.07.013, 2016.

Kilic, L., Tage Tonboe, R., Prigent, C., and Heygster, G.: Estimating the snow depth, the snow-ice interface temperature, and the effective temperature of Arctic sea ice using Advanced Microwave Scanning Radiometer 2 and ice mass balance buoy data, Cryosphere, 13, 1283–1296, https://doi.org/10.5194/tc-13-1283-2019, 2019.

Kurtz, N. T., Galin, N., and Studinger, M.: An improved CryoSat-2 sea ice freeboard retrieval algorithm through the use of waveform fitting, Cryosphere, 8, 1217–1237, https://doi.org/10.5194/tc-8-1217-2014, URL https://www.the-cryosphere.net/8/1217/2014/, 2014.

Kwok, R., Kurtz, N. T., Brucker, L., Ivanoff, A., Newman, T., Farrell, S. L., King, J., Howell, S., Webster, M. A., Paden, J., Leuschen, C., MacGregor, J. A., Richter-Menge, J., Harbeck, J., and Tschudi, M.: Intercomparison of snow depth retrievals over Arctic sea ice from radar data acquired by Operation IceBridge, The Cryosphere, 11, 2571–2593, https://doi.org/10.5194/tc-11-2571-2017, URL https://www.the-cryosphere.net/11/2571/2017/, 2017.

Lawrence, I. R., Tsamados, M. C., Stroeve, J. C., Armitage, T. W., and Ridout, A. L.: Estimating snow depth over Arctic sea ice from calibrated dual-frequency radar freeboards, Cryosphere, 12, 3551–3564, https://doi.org/10.5194/tc-12-3551-2018, URL https://www.the-cryosphere.net/12/3551/2018/, 2018.

Laxon, S., Peacock, H., and Smith, D.: High interannual variability of sea ice thickness in the Arctic region, Nature, 425, 947–950, https://doi.org/10.1038/nature02050, 2003.

Light, A. and Bartlein, P. J.: The end of the rainbow? color schemes for improved data graphics, Eos, 85, https://doi.org/10.1029/2004EO400002, 2004.

Mallett, R. D. C., Lawrence, I. R., Stroeve, J. C., Landy, J. C., and Tsamados, M.: Brief communication: Conventional assumptions involving the speed of radar waves in snow introduce systematic underestimates to sea ice thickness and seasonal growth rate estimates, The Cryosphere, 14, 251–260, https://doi.org/10.5194/tc-14-251-2020, URL `https://www.the-cryosphere.net/14/251/2020/`, 2020.

Nandan, V., Geldsetzer, T., Yackel, J., Mahmud, M., Scharien, R., Howell, S., King, J., Ricker, R., and Else, B.: Effect of Snow Salinity on CryoSat-2 Arctic First-Year Sea Ice Freeboard Measurements, Geophysical Research Letters, 44, 419–10, https://doi.org/10.1002/2017GL074506, URL `http://doi.wiley.com/10.1002/2017GL074506`, 2017.

Petty, A. A., Webster, M., Boisvert, L., and Markus, T.: The NASA Eulerian Snow on Sea Ice Model (NESOSIM) v1.0: Initial model development and analysis, Geoscientific Model Development, 11, 4577–4602, https://doi.org/10.5194/gmd-11-4577-2018, 2018.

Stauffer, R., Mayr, G. J., Dabernig, M., and Zeileis, A.: Somewhere over the rainbow: How to make effective use of colors in meteorological visualizations, Bulletin of the American Meteorological Society, 96, 203–216, https://doi.org/10.1175/BAMS-D-13-00155.1, 2015.

Thyng, K. M., Greene, C. A., Hetland, R. D., Zimmerle, H. M., and DiMarco, S. F.: True colors of oceanography: Guidelines for effective and accurate colormap selection, https://doi.org/10.5670/oceanog.2016.66, 2016.

Willatt, R., Laxon, S., Giles, K., Cullen, R., Haas, C., and Helm, V.: Ku-band radar penetration into snow cover on Arctic sea ice using airborne data, Annals of Glaciology, 52, 197–205, https://doi.org/10.3189/172756411795931589, 2011.

Willatt, R. C., Giles, K. A., Laxon, S. W., Stone-Drake, L., and Worby, A. P.: Field investigations of Ku-band radar penetration into snow cover on antarctic sea ice, IEEE Transactions on Geoscience and Remote Sensing, 48, 365–372, https://doi.org/10.1109/TGRS.2009.2028237, URL `http://ieeexplore.ieee.org/document/5282596/`, 2010.

---

## Referee Comment (RC2) · Isobel Lawrence (Referee) · 16 May 2020

Review of 'Simultaneous estimation of wintertime sea ice thickness and snow depth from space-borne freeboard measurements', Shi Hoyeon et al.
By Isobel R Lawrence.

**Paper synopsis**

A methodology is presented for estimating sea ice thickness and snow depth (on sea ice) using ice or snow freeboard, and snow surface and snow/ice interface temperatures (available from satellite data). The method is based on the assumption that conductive heat flux through the snow/ice interface is continuous, therefore the bulk temperature difference across the ice layer, divided by the ice thickness, is proportional to the temperature difference across the snow layer divided by its depth. The proportionality is a function of the conductivities of ice and snow, but can be derived empirically using temperature profiles from drifting buoys. The authors find a piecewise (2-part) linear relationship between the ratio of the temperature differential across the snow layer to the temperature change through the ice, and the ratio of snow to ice thickness. They present a thorough analysis of the possible physical mechanisms behind the two-part function, in the appendix.

Using the derived relationship, the ratio of snow depth to ice thickness can be estimated purely from satellite measurements of snow surface and snow/ice interface temperature. Then, combined with either snow or ice freeboard, this ratio can be used to simultaneously estimate snow depth and ice thickness. The appeal of the methodology lies in the fact that independent estimates of snow depth are not required to calculate ice thickness. The authors compare their ice thickness and snow depth estimates to those from OIB, and also against 'CryoSat-like' and 'ICESat-like' ice thicknesses, where snow depth from the Warren climatology is implemented in the traditional freeboard-to-thickness buoyancy equations. Finally the method is extended to the basin scale, using satellite-derived snow surface and snow/ice interface temperatures and sea ice freeboard data from CryoSat-2.

The manuscript is very well written, coherent and extremely thorough in its explanations, however something major has been overlooked which I believe means that this method cannot be used to retrieve snow depth and ice thickness from *radar* altimetry. That said, the method remains applicable to laser altimetry, e.g. IceSat-2, and is therefore still very valuable. I therefore recommend the paper for publication following this major revision:

**Major comments**

In order to use this methodology with satellite data from CryoSat-2, ice freeboard (the elevation of the snow/ice interface above the ocean surface) is required. However it is impossible to retrieve ice freeboard from CryoSat-2 without a-priori knowledge of the snow layer. Since the radar pulse slows down as it travels through the snow, snow depth is required in order to correct for the slower speed of propagation and estimate sea ice freeboard.

To compound the issue, the equation to convert radar freeboard into ice freeboard is incorrectly reported in a number of studies, including that of Kurtz et al. (2014, eq. 16) which describes the CS2 ice freeboard dataset you use in your analysis. Please see Mallett et al. (2020) for the correct derivation of the equation and details of its misreporting in the literature.

The correct equation for sea ice thickness from *radar* altimetry (assuming full snow penetration) is:

$$H = \left( f_r + h\left(\frac{c}{c_s} - 1\right) \right) \left(\frac{\rho_w}{\rho_w - \rho_i}\right) + h\left(\frac{\rho_s}{\rho_w - \rho_i}\right),$$

where $f_r$ is radar freeboard, as estimated from radar altimeters like CryoSat-2. If this equation cannot be solved by the proposed methodology (I do not believe it can be), then the paper should be restructured to focus on the laser case. The methodology remains valid for use with snow freeboards, and these are available from ICESat and now ICESat-2, so perhaps section 4.3 could be changed to an application to ICESat data. I appreciate that this will require a substantial amount of work, which is why I consider this revision major. However I find this methodology novel and valuable and the results in section 4.2 are encouraging; I would like to reiterate therefore that I think the paper deserves publication subject to this alteration and the following minor revisions:

**Minor comments**

I think you need to include an uncertainty budget for your sea ice thickness and snow depth estimates.

L30: "However, the radar scattering horizon is often treated as the snow–ice interface". Include Hendricks et al. (2016), Guerreiro et al. (2017), Tilling et al. (2018) as refs here since AWI, LEGOS and CPOM CryoSat-2 ice thickness products all make the same assumption.

L72: …"for given densities and freeboard" – (and assuming no snow penetration for laser and full snow penetration for radar)

L138: Could you say how many are discarded based on this criterion, and out of how many total.

L159: Can you provide a reference for the OIB data processing document where the densities are given?

L160: I understand that you keep ice density constant in order to compare with OIB data, but later when comparing with satellite-derived ice thickness should you not then use the densities used in those products for a fair comparison?

L198: "It was reported…" – By who?

L200: Where does $T_{si}$ for March come from if the Lee et al. (2018) dataset is only December-February ?

L201: "…if data frequency is over 20". Do you mean if 20 days out of the month contain data? Or are you referring to a number of points per grid cell?

L205: Please could you provide the details and a reference for which OIB dataset you used and where it is available from? i.e. L2, L4, Quicklook?

Figure 4: Why do you choose to show us the 7-day averaged plot in Fig.4 when Figure 3 was showing 15-day averaged temperature profiles?

L235: At the end of this sentence you could refer the reader to the appendix.

L244: bias is not near-zero in Fig 4b, it *is* zero.

L269: Did you calculate a different alpha for each year, and apply the different alpha to each year of OIB data? Or did you just average all the years together? Please clarify this in the text.

L295: Do you get the MW99 for input into Eqs. (4) and (5) from the CS2 data? If so is it monthly grid-averaged? How do you assign each OIB point a snow depth?

L301: "Therefore, if there are decreasing trends in both ice thickness and snow depth, the decreasing trend of ice thickness estimated from the constant snow depth will be diminished in radar, while being amplified in lidar" – This sentence seems overcomplicated. To me, all that the bottom two plots of Figure 7 demonstrate is that MW99 snow depth is larger than OIB snow depth. For the laser case, this means that using W99 causes ice thickness to be *underestimated* compared to H(OIB), and for the radar case using W99 results in ice thicknesses that are *too thick* compared to OIB. Perhaps you could plot MW99 against h(OIB) to clarify this? The retrieval of sea ice thickness from ICESat has not traditionally used the Warren climatology- see Kwok and Cunningham (2008) and Petty et al. (2020). Therefore I don't think it's justified to call this 'ICESat-like thickness' since you are not using the same snow depth product that they do.

**Typos / Grammar**

L128: "Therefore, the interface searching algorithm…" -> "Therefore, an interface searching algorithm…"

L165: "Sea ice thicknesses converted from MW99 using Eqs. (4) and (5) are also compared to examine how simultaneous retrievals…" **->** "Sea ice thicknesses are also calculated from Eqs. (4) and (5), using MW99 as snow depth, to examine how simultaneous retrievals…"

L194: "This reformatted AASTI-v2 data are called…" -> "This reformatted AASTI-v2 dataset is called…"

L293: "Examining how the current practices of retrieving the sea ice thickness through ICESat and CS2 measurements are compared with the simultaneous retrievals is of interest" **->** "We now examine how the current practises of retrieving sea ice thickness from ICESat and CS2 measurements compare with our method."

L294: "In doing so, OIB-measured…" -> "To do so, OIB-measured…"

L297: "Apparently, ICESat-like thickness tends…." -> "According to our analysis, ICESat-like thickness tends…."

L416: "…which are hard to be quantified explicitly." -> "…which are hard to quantify explicitly."

---

## Author Comment (AC1) · 29 May 2020

**Response to Review #1**

**1. Major Points**

**1.1. Difference between radar freeboard and scattering horizon height**

*L22: The statement "altimeters . . . measure sea ice freeboard" is only approximately correct in the case of radar altimeters. The instrument on board CS2 measures a time of flight, which can be related to the height of some radar scattering horizon only where no snow lies in between the scattering horizon and the instrument. When snow is in between and fully penetrated by the radar, the radar range to the scattering surface is overestimated due to slower pulse propagation in the overlying snow. Correcting for this and estimating the height of the ice-snow interface requires knowledge of the overlying snow (Mallett et al., 2020).*

*This issue surfaces again when the authors identify the radar freeboard as the height from the sea surface to the radar scattering horizon in L28. This is only the case for bare ice. Where overlying snow is present and fully penetrated, the radar freeboard is a finite distance below the ice freeboard (the assumed scattering surface). In the freeboard product used in this manuscript (Kurtz et al., 2014) this displacement is $h_s (1 − c_s/c)$.*

*This is relevant to Fig. 1, where $h_{rf}$ is depicted as being above the ice freeboard. While it may be true that the radar scattering horizon is above the snow-ice interface, in products that assume full radar penetration of the snowpack the radar freeboard is lower than the ice freeboard. Theoretically for total radar penetration and a freeboard depressed to near the water by snow, the radar freeboard can be below the waterline (while the ice freeboard and scattering horizon are above).*

We became aware of that the definition of 'radar freeboard' used in the manuscript is not consistent with the one used in CS2 data-related references (e.g., Kurtz et al., 2014), which is the estimated height of radar scattering horizon before applying the wave propagation speed correction. Therefore, Fig. 1 in the manuscript is replaced by figure AC1-1, by following your suggestion on using different notations for clarity.

Now, the system includes correction terms regarding the wave propagation speed change in the snow layer ($F_c$), and the displacement of the scattering horizon from the ice surface ($F_p$) following Kwok and Cunningham (2015) and Armitage and Ridout (2015).

$$F_i = F_r + \left( F_c - F_p \right) \tag{AC1-1}$$

$$F_c = (\eta_s - 1) f h_s \tag{AC1-2}$$

$$F_p = (1 - f)h_s \tag{AC1-3}$$

Here, $\eta_s$ denotes the refractive index of snow layer ($\eta_s = c/c_s$) and $f$ denotes the radar penetration factor, respectively. Combining three equations yields the following relationship.

$$F_i = F_r + (f\eta_s - 1)h_s \tag{AC 1-4}$$

In the revision, new formulae are introduced for simultaneously solving snow depth and ice thickness even for radar-based freeboard measurements.

**1.2 The freeboard product used by the authors has been created with mW99**

*In the final sentence of the abstract, the authors state:*

> *"In conclusion, the developed α-based method has the capacity to derive ice thickness and snow depth, without relying on the snow depth information as input to the buoyancy equation for converting freeboard to ice thickness."*

*However, the method presented here works directly from ice freeboard data which can only be derived by relying on snow depth information (Sect. 5.1 & Eq. 15 of Kurtz et al., 2014).*

*I feel that what the authors would like to present is a way to convert radar freeboards to ice thickness without relying on snow depth data, and this should be done before publication. I think it is possible for the authors to adapt their processing chain to deal with this, although it may complicate things.*

Thanks for the suggestion which in fact led to our deeper understanding of radar altimeter remote sensing. Luckily, we were able to include this issue in the retrieval, by modifying Eqs. (5) and (7) of the manuscript (see below for Eqs. (5) and (7) written with new notation), using Eq. (AC1-4).

Eq. (5): $H_i = \left(\frac{\rho_w}{\rho_w - \rho_i}\right) F_i + \left(\frac{\rho_s}{\rho_w - \rho_i}\right) h_s$

Eq. (7): $H_i = \frac{\rho_w}{\rho_w - \rho_i - \alpha \rho_s} F_i$

It is because Eq. (AC1-4) does not include additional unknowns, for given parameterization and assumption on the radar penetration. We assumed $f = 0.84$ for CS2 (Armitage and Ridout, 2015). $\eta_s$ can be parameterized as a function of the snow density, i.e., $\eta_s = (1 + 0.51\rho_s)^{1.5}$ (Ulaby et al., 1986). Here we present how the equations were solved.

First, the traditional method for the ice thickness retrieval with the snow depth as input can be written in the following equation by substituting $F_i$ with $F_r$ using Eq. (AC1-4).

$$H_i = \frac{\rho_w}{\rho_w - \rho_i} F_r + \frac{(f\eta_s - 1)\rho_w + \rho_s}{\rho_w - \rho_i} h_s \qquad \text{(AC1-5)}$$

Then, substituting $h_s$ with $\alpha H_i$ and rearranging the equation yield the equation for $H_i$ as a function of radar freeboard and $\alpha$, without snow depth information.

$$H_i = \frac{\rho_w}{\rho_w - \rho_i - \alpha\{(f\eta_s - 1)\rho_w + \rho_s\}} F_r \qquad \text{(AC1-6)}$$

Note that Eq. (AC1-6) becomes equivalent to the equation for the total freeboard if $f = 0$ (no wave penetration into snow layer).

This new setup requires the data processing chain to be modified as well. Here we describe what changes were made (in Sect. 3.3 and 3.4 of the manuscript). First, CS2-like radar freeboard was derived from OIB total freeboard ($F_t^{OIB}$) and snow depth ($h_s^{OIB}$). From Eq. (AC1-4) and the relationship $F_i = F_t - h_s$, the radar freeboard can be expressed as follows.

$$F_r^{OIB} = F_t^{OIB} - h_s^{OIB} - (f\eta_s - 1)h_s^{OIB} \qquad \text{(AC1-7)}$$

Because the main objective of using OIB data is to evaluate the relative performance of the simultaneous retrieval method when the method is applied to CS2 data, the radar penetration factor (*f*)

for OIB data processing is also set to be 0.84 (Artimage and Ridout, 2015). In the data processing chain, $h_s^{OIB}$ is removed if it is smaller than the given uncertainty level of the dataset (~5.7 cm) or it is larger than the total freeboard $F_t^{OIB}$.

The CS2 radar freeboard ($F_r^{CS2}$) was obtained from CS2 ice freeboard dataset. The CS2 ice freeboard data ($F_i^{CS2}$) distributed by NSIDC (Kurtz et al. 2017) assume that radar scattering horizon locates at snow–ice interface and applies a wave propagation speed correction. However, the correction was made using the MW99 snow depth ($h_s^{MW99}$) climatology with an erroneous correction form of $h_c = (1-\eta_s^{-1})\,h_s$, instead of the correct correction form of $h_c = (\eta_s - 1)\,h_s$ (Mallet et al. 2020). Thus, at this point, it is straightforward to derive the CS2 radar freeboard by removing the correction term as in the following equation.

$$F_r^{CS2} = F_i^{CS2} - (1 - \eta_s^{-1})h_s^{MW99} \qquad\qquad (AC1\text{-}8)$$

Finally, analyses in the first version of the manuscript are conducted again using the radar freeboard rather than using the ice freeboard. This time SIC criteria for $\alpha$ calculation was set to be 95% (original: 98%) for a wider coverage. Figs. AC1-2 to AC1-5 are the reprocessed results which will replace the figures in the manuscript. Despite of these changes, we find little changes in the conclusions we made in the first version of the manuscript. In addition, for more comprehensive information, snow depth comparison results are provided in Fig. AC1-5.

**1.3 Uncertainty Analysis**

*The authors state in their Discussion and Conclusions section:*

> *Overall, the developed α-based method yields ice thickness and snow depth, without relying on a priori 'uncertain' snow depth information, which results in uncertainty in the ice thickness retrieval.*

*They are of course correct to identify that uncertainty in snow depth leads to uncertainty in the ice thickness retrieval. To avoid having to quantify snow depth, they instead rely on a parameter equal to h/H, which they empirically derive from the temperature of the air-snow and ice-snow interfaces.*

*Clearly there is significant uncertainty in the value of α, and the authors should try to quantify how this propagates through into uncertainty in ice thickness. It's possible that their α parameter is more uncertain than other published data for h, and if so this method will deliver lower quality estimates of H than the traditional method.*

*It seems (looking at Fig. 1 of this review) that a given error in α would have a more serious impact on H than the same error in h, because the gradients of the lines are much more similar in on the left panel of Fig. 1 than on the right. This issue scales with the alpha parameter (i.e. as the freeboard goes down), and at high α very small uncertainties in alpha will lead to large uncertainties in H.*

*As alpha becomes so large that the freeboard tends to zero (not that uncommon in the Atlantic sector of the Arctic), the method seems to lose its usefulness, whereas the traditional method continues to function. That is to say in the case of near-zero freeboard, the traditional method still provides an*

*estimate of H, but that proposed by the authors does not (see Eq. 7 as $h_f \rightarrow 0$).*

*This is addressed in L161/162, where a critical value is given for alpha, and it is explained that for alpha above this value data are not produced. How often does this occur? And what is the effect on H of small errors in alpha just below this critical threshold?*

To identify the uncertainty of simultaneous method, snow depth error ($\Delta h_s$) equivalent to $\alpha$ error ($\Delta \alpha$) is calculated for chosen three representative sea ice conditions (thicker / moderate / thinner). The Simultaneous method showed a small sensitivity to $\Delta \alpha = 0.03$, which is RMSE value of the regression equation. On the other hand, sensitivity was greater when radar freeboard was used, especially for thinner ice where $\alpha$ is close to $\alpha_{crit}$. In case of ice thickness error ($\Delta H_i$), gap of sensitivity between total freeboard and radar freeboard methods was reduced because $\Delta H_i$ is more sensitive to $\Delta h_s$ when the total freeboard is used. This characteristic of sensitivity is consistent with results from OIB analysis.

Majority of data used in this study belong to moderate or thicker ice and retrieved $\alpha$ rarely exceeds $\alpha_{crit}$. Therefore, there should not be many cases having great uncertainty that might be expected from thinner ice condition. As a matter of fact, it seems that retrieved $\alpha$ shows reasonable values upon presumed thermodynamic condition. Areas where thermodynamic condition is not met are located at around the marginal ice zones and in the east of Greenland.

Details can be found in Appendix B (will be included in the revised manuscript). In addition, we clarify that "a priori 'uncertain' snow depth in formation' is MW99 snow depth climatology.

**2. Minor Points**

**2.1. General**

*• L21: The authors should consider directing the reader to Laxon et al. (2003) when illustrating that thickness has been estimated for nearly two decades.*

Laxon et al. (2003) is now included in the manuscript.

*• When discussing studies indicating the height difference between the scattering horizon and the snow-ice interface, the authors should consider directing the reader to Nandan et al. (2017) and Willatt et al. (2010, 2011).*

Nandan et al. (2017) and Willatt et al. (2011) are now referred in the manuscript. Because characteristic of sea ice is different between Arctic and Antarctic, study on Antarctic sea ice by Willatt et al. (2010) is not included.

*• L62 & 64: Define RTM before using the acronym*

The acronym 'RTM' stands for 'Radiative Transfer Model'. It is now defined in the manuscript.

• *The font sizes of some annotations to Figure 3 should be increased so as to be legible and comparable to the (a), (b), (c) lettering.*

Annotation is now increased to be legible (see Fig. AC1-6).

• *In Fig. 2, the box that reads 'Find temperature discontinuity point'. It is my understanding that the temperature is continuous (but not a smooth function), and therefore it has no discontinuities (but its gradient does). Should this box then read 'Find temperature gradient discontinuity point'?*

Yes, what discontinuous is temperature gradient, not temperature. The text now reads 'Find temperature gradient discontinuity point' (see Fig. AC1-7).

• *I think the notation of H and h in combination with $h_f$, $h_{rf}$ and $h_{tf}$ is confusing to the casual reader. For instance, the fact that h and $h_f$ look so similar but are in fact unrelated confused me initially. Even changing $H \rightarrow H_{ice}$ and $h \rightarrow h_{snow}$ would clarify this.*

Following your comment, we have changed our notations (see Sect 1.1 in this response).

**2.2. Validation of H against OIB Data**

*The authors are able to create two products from freeboard data obtained by OIB and CryoSat-2, one for snow depth (h), and one for ice thickness (H). They then rightly try to assess the quality of these data products against other datasets, namely the OIB snow depth and ice thickness data. There are at least five algorithms published to process the raw OIB radar returns into along-track snow depth data, and they produce a spread in the mean snow depth (Kwok et al., 2017). 'Validation' of a model or data implies comparison to true or certain values, and it is unclear which OIB snow depth product (if any) represents the truth. This limits the strength of the validation exercise. Nonetheless, I understand that OIB snow depth values have historically been taken as the truth in published work so this is a perhaps not a big issue. It might also be argued that the spread of different OIB data is sufficiently small relative to other methods of snow depth estimation to allow OIB to approximate the truth for validation purposes.*

*I feel that there is however a more significant issue with the authors' claims to have 'validated' their ice thickness data against OIB ice thickness data ($H_{OIB}$). OIB aircraft instruments do not measure thickness ($H_{true}$) directly, but instead estimate it based on freeboard, snow depth, snow density and ice density values. As such, OIB thickness data (while likely to be the most accurate data on $H_{true}$ outside of in-situ measurement), undoubtedly suffer from biases involving snow depth, snow density and ice density, and therefore should not be mistaken for $H_{true}$.*

*The technique for determining $H_{OIB}$ is very similar to that presented in this manuscript: the authors use identical freeboard, snow density and ice density values to estimate thickness with the hydrostatic equilibrium assumption. Given these similarities, comparing the thickness estimates in this paper with OIB thickness estimates doesn't really qualify as independent validation.*

*It seems more like the exercise of comparing H estimates is in fact comparing the novel snow depth estimates with OIB snow depths (Fig 7 top row; a valuable analysis), and then investigating how that singular difference propagates into sea ice thickness estimates. I suspect that the strong agreement between the two datasets presented in the middle row of Fig. 7. is largely a result of the identical radar freeboards and geophysical parameters used in each processing chain.*

*After all, much of sea ice thickness is determined by radar freeboard information, independent of snow data. The fact that the 'simultaneous' method matches $H_{OIB}$ data more closely than MW99 is therefore evidence that the snow depth product produced by the 'simultaneous' method is closer to OIB snow depths than MW99 (because everything else is equal).*

*I think it is perfectly reasonable (and in fact expected) to compare H estimates from the new method with $H_{OIB}$. However, I think this should be presented as a 'comparison with' or 'evaluation against' OIB data, rather that implying that the new data are being validated against some true value. It is also an understandable bit of reasoning to say that values which are closer to $H_{OIB}$ are likely to be closer to $H_{true}$, but if this assumption is made it should be stated explicitly.*

We agree upon your notion that snow depth and ice thickness comparisons are the same problem. To address your comment, we first clarified how $H_i^{OIB}$ is calculated in Sect 3.3. Then, we changed the subtitle of Sect 4.2 from '*Validation* against OIB estimates' to '*Evaluation* against OIB estimates'. Finally, *validation* on ice thickness contents were modified in the direction to address that the estimated snow depth showing a more consistency with $h_s^{OIB}$ implies improved ice thickness. Accordingly, the snow depth comparisons between $h_s(\alpha^{sat}, F_r^{OIB})$ vs. $h_s^{MW99}$ and $h_s(\alpha^{sat}, F_r^{CS2})$ vs. $h_s^{MW99}$ are included in Fig. AC1-5.

**2.3. Limitations of Other Data**

*L66 - 69: Other approaches worth mentioning are snow depth retrieval using dual-frequency altimetry (Guerreiro et al., 2016; Lawrence et al., 2018, Kwok and Markus, 2018), snow on sea ice model accumulating snowfall from reanalysis (Petty et al., 2018), multilinear regression (Kilic et al., 2019), and the neural network approach (Braakmann-Folgmann and Donlon, 2019). However, these methods do not satisfactorily meet the criteria required for freeboard to ice conversion over the entire Arctic Ocean basin scale or multi-year time scale.*

*The approach of Guerreiro et al. (2016) and Lawrence et al. (2018) are limited latitudinally by the AltiKa orbital inclination and Lawrence et al. (2018) additionally through calibration with OIB which only operates in Spring. As the authors identify, they are limited in spatial or temporal extent. While there are limitations to the data products of Petty et al. (2018), Kilic et al. (2019) and Braakmann-Folgmann and Donlon (2019), it's not obvious that these can be characterized by failure to cover the entire basin on a multiyear timescale. As such, the statement on L69 that they do not satisfactorily meet these criteria should be clarified.*

The purpose of this paragraph was to introduce recent researches to readers. Therefore, the last sentence describing limitations of other data, which is not necessary, is removed. In case of Petty et al. (2018), we decided to not include it in the text to keep manuscript's focus on remote sensing, as characteristic of their product seems to be closer to model than remote sensing. Accordingly, paragraph

will be modified as:

"Other *satellite remote sensing* approaches worth mentioning are snow depth retrieval using dual-frequency altimetry (Guerreiro et al., 2016; Lawrence et al., 2018, Kwok and Markus, 2018), multilinear regression (Kilic et al., 2019), and the neural network approach (Braakmann-Folgmann and Donlon, 2019)."

**2.4. Rainbow Color Schemes**

*Where possible, authors should avoid presenting continuous data with 'rainbow' color schemes as in Figures 6 & 8. This is because (among other reasons) the scheme tends to imply sharp transitions in the data where they do not exist (Borland and Taylor, 2007). Alternatives for geoscientists are given by Light and Bartlein (2004), Stauffer et al. (2015) and Thyng et al. (2016).*

Thanks for valuable comment. We changed our color scheme to generate figures from 'jet' from 'viridis', which is perceptually uniform colormap, available in matplotlib/python. Figs. AC1-2 and AC1-4 are new plots with the new color scheme.

[revised manuscript text omitted]

Because $h_s$ is a combination of freeboard and $\alpha$, as in Eq. (AC1-6), we only examine the uncertainty with some representative sea ice types. Here physical states for thicker ice (type A), moderate ice (type B) and thinner ice (type C) are chosen, which are summarized in Table B1. Those typical values are for three types are shown over the scatterplots of OIB-based ($\alpha$ vs. $F_t$) and of satellite-based ($\alpha$ vs. $F_r$) – Fig. B1. It is shown that the majority of data points are located around type B, followed by type A. There seem a very small portion of total samples showing values around the type C.

With $\Delta\alpha = \pm0.03$, which is an RMSE range of the $\alpha$-prediction equation, $\Delta h_s$ and $\Delta H_i$ are estimated for the three ice types. Results summarized in Table B2 show that $|\Delta h_s|$ is within 5 cm and it tends to decrease as the ice becomes thinner, when the current method is applied to the total freeboard. On the other hand, $|\Delta h_s|$ shows more sensitive behavior for the same $\Delta\alpha$ when the radar freeboard is used for the retrieval. Especially, the sensitivity of type C is the greatest. This is because the denominator of Eq (AC1-6) becomes smaller when $\alpha$ approaches to $\alpha_{crit}$, resulting in unstable solution. For the ice thickness, $|\Delta H_i|$ is smaller when the total freeboard is used since $\Delta H_i$ is proportional to $\Delta h_s$. However, the gap between the results from two freeboards has narrowed because $H_i$ from the total freeboard is more sensitive than the radar freeboard to $\Delta h_s$, according to Eq. (B2). Sensitivity characteristics shown here are consistent with analysis results given in Sect 4.2. Because there is a much small number of data points belonging to the type C, at least in the data used for this study, the overall sensitivity would likely be in between the B and A types.

[revised manuscript text omitted]

---

## Author Comment (AC2) · 29 May 2020

**Response to the review by Isobel R. Lawrence.**

**Major comments**

*In order to use this methodology with satellite data from CryoSat-2, ice freeboard (the elevation of the snow/ice interface above the ocean surface) is required. However, it is impossible to retrieve ice freeboard from CryoSat-2 without a-priori knowledge of the snow layer. Since the radar pulse slows down as it travels through the snow, snow depth is required in order to correct for the slower speed of propagation and estimate sea ice freeboard.*

*To compound the issue, the equation to convert radar freeboard into ice freeboard is incorrectly reported in a number of studies, including that of Kurtz et al. (2014, eq. 16) which describes the CS2 ice freeboard dataset you use in your analysis.*

*Please see Mallett et al. (2020) for the correct derivation of the equation and details of its misreporting in the literature. The correct equation for sea ice thickness from radar altimetry (assuming full snow penetration) is:*

$$ H = \left( f_r + h \left( \frac{c}{c_s} - 1 \right) \right) \left( \frac{\rho_w}{\rho_w - \rho_i} \right) + h \left( \frac{\rho_s}{\rho_w - \rho_i} \right), $$

*where $f_r$ is radar freeboard, as estimated from radar altimeters like CryoSat-2. If this equation cannot be solved by the proposed methodology (I do not believe it can be), then the paper should be restructured to focus on the laser case. The methodology remains valid for use with snow freeboards, and these are available from ICESat and now ICESat-2, so perhaps section 4.3 could be changed to an application to ICESat data. I appreciate that this will require a substantial amount of work, which is why I consider this revision major. However I find this methodology novel and valuable and the results in section 4.2 are encouraging; I would like to reiterate therefore that I think the paper deserves publication subject to this alteration and the following minor revisions:*

Thanks for the comment on the fact that the radar algorithm depends on the snow depth, even before retrieving the snow depth. As a matter of fact, another reviewer raised the similar concern, and thus following responses are nearly same.

Recognizing the problems related to the radar altimetry, we modified equations for the model system to handle the radar freeboard as well. The modified model system is delineated in Figure AC2-1 (with slightly changed notations).

Now, the system includes correction terms regarding the wave propagation speed change in the snow layer ($F_c$), and the displacement of the scattering horizon from the ice surface ($F_p$) following Kwok and Cunningham (2015) and Armitage and Ridout (2015).

$$ F_i = F_r + \left( F_c - F_p \right) \tag{AC2-1} $$

$$ F_c = \left( \eta_s - 1 \right) f h_s \tag{AC2-2} $$

$$F_p = (1 - f)h_s \tag{AC2-3}$$

Here, $\eta_s$ denotes the refractive index of snow layer ($\eta_s = c/c_s$) and $f$ denotes the radar penetration factor, respectively. Combining three equations yields the following relationship.

$$F_i = F_r + (f\eta_s - 1)h_s \tag{AC 2-4}$$

Luckily, we were able to include this issue in the retrieval, by modifying Eqs. (5) and (7) of the manuscript (see below for Eqs. (5) and (7) written with new notation), using Eq. (AC2-4).

Eq. (5): $H_i = \left(\frac{\rho_w}{\rho_w - \rho_i}\right) F_i + \left(\frac{\rho_s}{\rho_w - \rho_i}\right) h_s$

Eq. (7): $H_i = \frac{\rho_w}{\rho_w - \rho_i - \alpha\rho_s} F_i$

It is because Eq. (AC2-4) does not include additional unknowns, for given parameterization and assumption on the radar penetration. We assumed $f = 0.84$ for CS2 (Armitage and Ridout, 2015). $\eta_s$ can be parameterized as a function of the snow density, i.e., $\eta_s = (1 + 0.51\rho_s)^{1.5}$ (Ulaby et al., 1986). Here we present how the equations were solved.

First, the traditional method for the ice thickness retrieval with the snow depth as input can be written in the following equation by substituting $F_i$ with $F_r$ using Eq. (AC2-4).

$$H_i = \frac{\rho_w}{\rho_w - \rho_i} F_r + \frac{(f\eta_s - 1)\rho_w + \rho_s}{\rho_w - \rho_i} h_s \tag{AC2-5}$$

Please note that this equation is equivalent to the equation assuming full snow penetration which you presented. Then, substituting $h_s$ with $\alpha H_i$ and rearranging the equation yield the equation for $H_i$ as a function of radar freeboard and $\alpha$, without snow depth information.

$$H_i = \frac{\rho_w}{\rho_w - \rho_i - \alpha\{(f\eta_s - 1)\rho_w + \rho_s\}} F_r \tag{AC2-6}$$

Also note that Eq. (AC2-6) becomes equivalent to the equation for the total freeboard if $f = 0$ (no wave penetration into snow layer).

This new setup requires the data processing chain to be modified as well. Here we describe what changes were made (in Sect. 3.3 and 3.4 of the manuscript). First, CS2-like radar freeboard was derived from OIB total freeboard ($F_t^{OIB}$) and snow depth ($h_s^{OIB}$). From Eq. (AC2-4) and the relationship $F_i = F_t - h_s$, the radar freeboard can be expressed as follows.

$$F_r^{OIB} = F_t^{OIB} - h_s^{OIB} - (f\eta_s - 1)h_s^{OIB} \tag{AC2-7}$$

Because the main objective of using OIB data is to evaluate the relative performance of the simultaneous retrieval method when the method is applied to CS2 data, the radar penetration factor ($f$) for OIB data processing is also set to be 0.84 (Artimage and Ridout, 2015). In the data processing chain, $h_s^{OIB}$ is removed if it is smaller than the given uncertainty level of the dataset (~5.7 cm) or it is

larger than the total freeboard $F_t^{OIB}$.

The CS2 radar freeboard ($F_r^{CS2}$) was obtained from CS2 ice freeboard dataset. The CS2 ice freeboard data ($F_i^{CS2}$) distributed by NSIDC (Kurtz et al. 2017) assume that radar scattering horizon locates at snow–ice interface and applies a wave propagation speed correction. However, the correction was made using the MW99 snow depth ($h_s^{MW99}$) climatology with an erroneous correction form of $h_c = (1 - \eta_s^{-1}) h_s$, instead of the correct correction form of $h_c = (\eta_s - 1) h_s$ (Mallet et al. 2020). Thus, at this point, it is straightforward to derive the CS2 radar freeboard by removing the correction term as in the following equation.

$$F_r^{CS2} = F_i^{CS2} - (1 - \eta_s^{-1})h_s^{MW99} \tag{AC2-8}$$

Finally, analyses in the first version of the manuscript are conducted again using the radar freeboard rather than using the ice freeboard. This time SIC criteria for $\alpha$ calculation was set to be 95% (original: 98%) for a wider coverage. Figs. AC2-2 to AC2-5 are the reprocessed results which will replace the figures in the manuscript. Despite of these changes, we find little changes in the conclusions we made in the first version of the manuscript. In addition, for more comprehensive information, snow depth comparison results are provided in Fig. AC2-5.

**Minor comments**

• *I think you need to include an uncertainty budget for your sea ice thickness and snow depth estimates.*

Sensitivity test for our method was conducted and the result will be included as Appendix B in the revised manuscript. In Appendix B, snow depth error caused by $\alpha$ error is presented for different cases of $\alpha$ and freeboard.

• *L30: "However, the radar scattering horizon is often treated as the snow–ice interface". Include Hendricks et al. (2016), Guerreiro et al. (2017), Tilling et al. (2018) as refs here since AWI, LEGOS and CPOM CryoSat-2 ice thickness products all make the same assumption.*

Hendricks et al. (2016), Guerreiro et al. (2017) and Tilling et al. (2018) are now referred in the manuscript.

• *L72: …"for given densities and freeboard" – (and assuming no snow penetration for laser and full snow penetration for radar)*

We rewrote the sentence as follow:

    "… for given densities, freeboard and assumptions on wave penetration"

*• L138: Could you say how many are discarded based on this criterion, and out of how many total.*

By examining the outputs from the program, we found no outputs discarded by this criterion, therefore, we removed this sentence from the text. Here we provide total number of buoy data obtained from different averaging periods for your information (Table AC2-1, will not be included in the text).

*• L159: Can you provide a reference for the OIB data processing document where the densities are given?*

We now referred Kurtz et al. (2013) in the text.

*• L160: I understand that you keep ice density constant in order to compare with OIB data, but later when comparing with satellite-derived ice thickness should you not then use the densities used in those products for a fair comparison?*

As you mentioned, '*CS2 H*' in Fig 9 of the manuscript should not be same as the CS2 product available from NSIDC. Instead, CS2 ice thickness is reproduced with the same densities and MW99 snow depth for the fair comparison. New figures are given in Fig. AC2-5. We also clarified this data processing in the comparison section.

*• L198: "It was reported…" – By who?*

It was reported by Lee et al. (2018). We clarified this in the manuscript.

*• L200: Where does $T_{si}$ for March come from if the Lee et al. (2018) dataset is only December-February?*

We produced $T_{si}$ for March by applying the same algorithm. We clarified this in the text.

*• L201: "…if data frequency is over 20". Do you mean if 20 days out of the month contain data? Or are you referring to a number of points per grid cell?*

Monthly mean temperature was calculated by gird cell by grid cell and the average was done only for the grid cells where there are more than 20 data available during a month. We clarified this in the manuscript.

*• L205: Please could you provide the details and a reference for which OIB dataset you used and where it is available from? i.e. L2, L4, Quicklook?*

We utilized L4 dataset for 2011-2013 period, and Quick look dataset for 2014-2015 period. Details on OIB data are now provided in the manuscript. Reference and accessibility information were already included in 'Data availability' section.

> "Five years of OIB data during 2011-2015 period are utilized in this study. Accordingly, OIB level 4 dataset (Kurtz et al., 2015) during 2011-2013 period and Quick look dataset (https://doi.org/10.5067/7Q8HCCWS4I0R, last access: 20 May 2020) during 2014-2015 period are obtained from NSIDC."

*• Figure 4: Why do you choose to show us the 7-day averaged plot in Fig.4 when Figure 3 was showing 15-day averaged temperature profiles?*

We intended to show results from various averaging period to readers. The results for different averaging period can be found in Figs. 5 and 6 in the text. For your information, same figures for different averaging periods are presented in Fig. AC2-6 (will not be included in the text).

*• L235: At the end of this sentence you could refer the reader to the appendix.*

Appendix A is now referred at the end of the sentence.

*• L244: bias is not near-zero in Fig 4b, it is zero.*

Yes, it is zero. The comment is now applied in the text.

*• L269: Did you calculate a different alpha for each year, and apply the different alpha to each year of OIB data? Or did you just average all the years together? Please clarify this in the text.*

We calculated and applied a different $\alpha$ for each year. It is now clarified in the text.

*• L295: Do you get the MW99 for input into Eqs. (4) and (5) from the CS2 data? If so is it monthly grid-averaged? How do you assign each OIB point a snow depth?*

Yes, MW99 was obtained from the CS2 dataset. OIB data were reformatted in a 25 km polar stereographic grid by method described in Sect. 3.3 in the manuscript. As OIB data is reformatted into the grid format, it is straightforward to assign the OIB snow depth to the MW99.

One possible concern is that the monthly mean of daily MW99 might differ significantly from the daily MW99, because the MW99 depends on the sea ice type. However, it may not be a critical issue when following points are considered: 1) W99 is already a monthly climatology, 2) ice type distribution would not be changed significantly by the sea ice drift in March because the Arctic Ocean is nearly filled with sea ice and thus the sea ice mobility is reduced.

*• L301: "Therefore, if there are decreasing trends in both ice thickness and snow depth, the decreasing trend of ice thickness estimated from the constant snow depth will be diminished in radar, while being amplified in lidar" – This sentence seems overcomplicated. To me, all that the bottom two plots of Figure 7 demonstrate is that MW99 snow depth is larger than OIB snow depth. For the laser case, this means that using W99 causes ice thickness to be underestimated compared to H(OIB), and for the radar case using W99 results in ice thicknesses that are too thick compared to OIB. Perhaps you could plot MW99 against h(OIB) to clarify this? The retrieval of sea ice thickness from ICESat has not traditionally used the Warren climatology- see Kwok and Cunningham (2008) and Petty et al. (2020). Therefore I don't think it's justified to call this 'ICESat-like thickness' since you are not using the same snow depth product that they do.*

Yes, what you mentioned is the appropriate interpretation of Fig. 7; MW99 is larger than OIB snow depth. This can be verified by Fig. S1 (will be included as a supplementary figure in the text). However, we attempted to address a possible unintended result of 'diverging direction in errors in ice thickness retrieval, when the same snow depth error is applied to two different satellite altimetry measurements'. We modified this paragraph to deliver such message.

> "The negative bias of ICESat-like thickness and positive bias of CS2-like thickness compared to $H_i^{OIB}$ demonstrate that $h_s^{MW99}$ is greater than $h_s^{OIB}$ (as shown in Fig. S1), according to Eqs. (4) and (5). In other words, the sensitivity of ice thickness diverges for two different types of altimeter to the same snow depth error. Therefore, …"

Regarding the naming issue, we are not referring existing products when we call ICESat-like and CS2-like thickness. Those are explicitly defined in the text:

> "In doing so, OIB-measured total freeboard and ice freeboard are converted into ice thickness using MW99 as input to solve Eqs. (4) and (5). These two ice thickness retrievals are referred to as ICESat-like thickness and CS2-like thickness, respectively"

Besides, there is an ice thickness dataset from ICESat total freeboard distributed by NSIDC (Yi and Zwally, 2009; doi: 10.5067/SXJVJ3A2XIZT) which uses MW99. However, we think that the expression "current practices of retrieving sea ice thickness" might confuse readers. Therefore, we will replace such expression with "MW99 method" for clarity.

**Typos / Grammar**

• *L128: "Therefore, the interface searching algorithm…" -> "Therefore, an interface searching algorithm…"*

• *L165: "Sea ice thicknesses converted from MW99 using Eqs. (4) and (5) are also compared to examine how simultaneous retrievals…" -> "Sea ice thicknesses are also calculated from Eqs. (4) and (5), using MW99 as snow depth, to examine how simultaneous retrievals…"*

• *L194: "This reformatted AASTI-v2 data are called…" -> "This reformatted AASTI-v2 dataset is called…"*

• *L293: "Examining how the current practices of retrieving the sea ice thickness through ICESat and CS2 measurements are compared with the simultaneous retrievals is of interest" -> "We now examine how the current practises of retrieving sea ice thickness from ICESat and CS2 measurements compare with our method."*

• *L294: "In doing so, OIB-measured…" -> "To do so, OIB-measured…"*

• *L297: "Apparently, ICESat-like thickness tends…." -> "According to our analysis, ICESat-like thickness tends…."*

• *L416: "…which are hard to be quantified explicitly." -> "…which are hard to quantify explicitly."*

All comments are applied in the manuscript.

**Table AC2-1.** Number of outputs obtained from the interface searching algorithm

| Averaging period | # of obtained outputs |
|:---:|:---:|
| 1 day | 542 |
| 7 days | 97 |
| 15 days | 59 |
| 30 days | 36 |

[Figure]

**Figure AC2-1.** Schematic diagram of a typical snow–ice system during the winter. Snow depth ($h_s$), ice thickness ($H_i$), total freeboard ($F_t$), radar freeboard ($F_r$), and ice freeboard ($F_i$) are indicated. Correction terms regarding the wave propagation speed change in snow layer ($F_c$) and the displacement of the scattering horizon from the ice surface ($F_p$) are indicated by blue arrows. Red line denotes a typical temperature profile with air–snow interface temperature ($T_{as}$), snow–ice interface temperature ($T_{si}$) and ice–water interface temperature ($T_{iw}$). *In the text, change was also made on the notation for the bulk densities of materials ($\rho_i$: density of sea ice, $\rho_s$: density of snow, $\rho_w$: density of sea water).*

[Figure]

**Figure AC2-2.** Simultaneously retrieved ice thickness and snow depth from OIB total/radar freeboard in March of the 2011–2015 period. Corresponding OIB products are at the bottom.

[Figure]

**Figure AC2-3.** Scatter plots between OIB products and the simultaneously retrieved snow depth and ice thickness from OIB total/radar freeboards during the March 2011–2015 period. Corresponding ice thicknesses estimated from MW99 snow depth are in the third row. The red lines are linear regression lines.

[Figure]

**Figure AC2-4.** Geographical distributions of observed CS2 radar freeboard ($F_r$) and estimated snow–ice thickness ratio ($\alpha$), ice thickness ($H_i$), and snow depth ($h_s$) from December 2013 to March 2014. Grey area in the second row denote where $\alpha$ retrieval is failed because $T_{as}$ is warmer than $T_{as}$.

[Figure]

**Figure AC2-5.** Comparison of simultaneous retrieved snow depth and ice thickness to those from MW99 method. (a) Snow depth from OIB radar freeboard, (b) snow depth from CS2 radar freeboard, (c) ice thickness from OIB radar freeboard, and (d) ice thickness from CS2 radar freeboard.

[Figure]

**Figure AC2-6.** (Left column) Scatterplots of the temperature difference ratio of the snow and ice layer ($\Delta T_{snow}/\Delta T_{ice}$) and the snow–ice thickness ratio ($\alpha$) for averaging period of 1,7,15 and 30 days. Color denotes collected year of buoy data. The red lines are the regression lines (defined in Eq. (8) in the text). (Right column) The corresponding scatter plot of observed and regressed $\alpha$.

[Figure]

**Figure S1.** Comparison between OIB snow depth and MW99 snow depth during March of 2011-2015 period.

**Appendix B. Sensitivity test for the proposed method**

Here we present results of the sensitivity test of showing how the snow depth and ice thickness retrieval results are dependent on the uncertainties in $\alpha$. To do so, the uncertainty in the snow depth ($\Delta h_s$) due to the $\alpha$ error (i.e. $\Delta\alpha$) and associated ice thickness error ($\Delta H_i$) are estimated. From this sensitivity test, we expect to understand why the simultaneous method for the radar freeboard shows more scattered features than those from the method for the lidar total freeboard.

First, $\Delta h_s$ is defined by the difference of retrieved $h_s$ between with error ($\alpha + \Delta\alpha$) and without error ($\alpha$).

$$\Delta h_s = \begin{cases} h_s(\alpha + \Delta\alpha, F_t) - h_s(\alpha, F_t) & \text{(using } F_t) \\ h_s(\alpha + \Delta\alpha, F_r) - h_s(\alpha, F_r) & \text{(using } F_r) \end{cases} \tag{B1}$$

Then, $\Delta h_s$ can be converted to the error in the ice thickness ($\Delta H_i$) using the following equation derived from Eq. (AC2-5).

$$\Delta H_i = \frac{(f\eta_s - 1)\rho_w + \rho_s}{\rho_w - \rho_i} \Delta h_s = \begin{cases} -6.46\Delta h_s & \text{(using } F_t) \\ 3.44\Delta h_s & \text{(using } F_r) \end{cases} \tag{B2}$$

Because $h_s$ is a combination of freeboard and $\alpha$, as in Eq. (AC2-6), we only examine the uncertainty with some representative sea ice types. Here physical states for thicker ice (type A), moderate ice (type B) and thinner ice (type C) are chosen, which are summarized in Table B1. Those typical values are for three types are shown over the scatterplots of OIB-based ($\alpha$ vs. $F_t$) and of satellite-based ($\alpha$ vs. $F_r$) – Fig. B1. It is shown that the majority of data points are located around type B, followed by type A. There seem a very small portion of total samples showing values around the type C.

With $\Delta\alpha = \pm0.03$, which is an RMSE range of the $\alpha$-prediction equation, $\Delta h_s$ and $\Delta H_i$ are estimated for the three ice types. Results summarized in Table B2 show that $|\Delta h_s|$ is within 5 cm and it tends to decrease as the ice becomes thinner, when the current method is applied to the total freeboard. On the other hand, $|\Delta h_s|$ shows more sensitive behavior for the same $\Delta\alpha$ when the radar freeboard is used for the retrieval. Especially, the sensitivity of type C is the greatest. This is because the denominator of Eq (AC2-6) becomes smaller when $\alpha$ approaches to $\alpha_{crit}$, resulting in unstable solution. For the ice thickness, $|\Delta H_i|$ is smaller when the total freeboard is used since $\Delta H_i$ is proportional to $\Delta h_s$. However, the gap between the results from two freeboards has narrowed because $H_i$ from the total freeboard is more sensitive than the radar freeboard to $\Delta h_s$, according to Eq. (B2). Sensitivity characteristics shown here are consistent with analysis results given in Sect 4.2. Because there is a much small number of data points belonging to the type C, at least in the data used for this study, the overall sensitivity would likely be in between the B and A types.

[revised manuscript text omitted]

---

## Editor Decision (ED1)

**Review of: Estimating the snow depth, the snow–ice interface temperature, and the effective temperature of Arctic sea ice using Advanced Microwave Scanning Radiometer 2 and ice mass balance buoy data**

**1 Introduction**

**1.1 Summary of Review**

This manuscript addresses the issue of converting satellite radar ranging data to sea ice thickness estimates, while simultaneously retrieving the depth of the overlying snow. This is done through a novel linear mapping of the temperature gradients in the snow and ice to the thickness values of the materials.

However the manuscript does contain a technical issue which affects its application to CryoSat-2 data (described in Sect. 2.2), which must be remedied before publication. In short, the freeboard data used in the study have been calculated using the modified Warren climatology, so any thickness estimates resulting from them cannot be independent of that climatology.

Otherwise, this manuscript contains a novel approach to estimating the depth of snow on sea ice, and the authors show that the related thickness values match OIB thickness values more closely than those resulting from use of the modified Warren climatology.

I believe that if the issues presented in this review can be resolved, subsequent publication of this manuscript would be of significant interest to the sea ice community.

**1.2 Summary of the Technique Presented**

Sea ice thickness is conventionally estimated from ice freeboard as such (using the notation of this paper):

$$H = \frac{h_f \rho_w + h \rho_s}{\rho_w - \rho_i} \qquad (1)$$

Where $H$ is sea ice thickness, $h_f$ the height of the snow ice interface above the waterline, and $h$ the depth of the overlying snow. $\rho$ values represent the densities of overlying snow, water and ice. From the above, it can be seen that for $h_f$ to be converted to $H$, $h$ must be known (as well as all the density values). Unfortunately despite a major research effort to map the spatial distribution of $h$, it remains highly uncertain.

One of the novelties of this paper is the authors' framing of the problem thus:

$$H = h_f \frac{\rho_w}{\rho_w - \rho_i - (h/H)\rho_s} \qquad (2)$$

This framing allows $h_f$ to be converted to $H$ without direct knowledge of $h$, instead it is sufficient just to know the ratio $h/H$ - this is referred to as $\alpha$. The authors then show that this formulation is useful, by estimating (h/H) in a Pan-Arctic way using satellite-measured temperatures of the air-snow and snow-ice interfaces.

A byproduct of this approach is that snow depth ($h$) is also trivially extracted:

$$h = h_f \frac{\rho_w}{\rho_w - \rho_i - (h/H)\rho_s} \times (h/H) \qquad (3)$$

[Figure]

Figure 1: Graph of $H, h$ space for a case with typical values of densities, snow depth and ice thickness

It should be noted that this approach of 'simultaneously' working out $h$ and $H$ is mathematically equivalent to first working out $h$ using Eq. (3) and substituting the result into Eq. (1).

The method can be further understood graphically in $h, H$ space, as in Figure 1. When the ice freeboard is known, the corresponding snow depth ($h$) and ice thickness ($H$) lie unconstrained in $h, H$ space along the blue line described by Eq. (1).

Thickness is then traditionally determined by a priori knowledge of the snow depth ($h_0$) - this is depicted in the left hand plot by the intersection of the green line (known snow depth) with the blue line.

This paper introduces a different relationship to constrain sea ice thickness, depicted on the right, revolving around the empirical parameter $\alpha$. By finding the intersection with the red ($\alpha$) line and the blue line, the authors are able to simultaneously derive $h$ and $H$.

**2 Major Points**

**2.1 Difference between radar freeboard and scattering horizon height**

L22: The statement "altimeters ... measure sea ice freeboard" is only approximately correct in the case of radar altimeters.

The instrument on board CS2 measures a time of flight, which can be related to the height of some radar scattering horizon only where no snow lies in between the scattering horizon and the instrument. When snow is in between and fully penetrated by the radar, the radar range to the scattering surface is overestimated due to slower pulse propagation in the overlying snow. Correcting for this and estimating the height of the ice-snow interface requires knowledge of the overlying snow (Mallett et al., 2020).

This issue surfaces again when the authors identify the radar freeboard as the height from the sea surface to the radar scattering horizon in L28. This is only the case for bare ice. Where overlying snow is present and fully penetrated, the radar freeboard is a finite distance below the ice freeboard (the assumed scattering surface). In the freeboard product used in this manuscript (Kurtz et al., 2014) this displacement is $h_s(1 - c_s/c)$.

This is relevant to Fig. 1, where $h_{rf}$ is depicted as being above the ice freeboard. While it may be true that the radar scattering horizon is above the snow-ice interface, in products that assume full radar penetration of the snowpack the radar freeboard is lower than the ice freeboard. Theoretically for total radar penetration and a freeboard depressed to near the water by snow, the radar freeboard can be below the waterline (while the ice freeboard and scattering horizon are above).

**2.2 The freeboard product used by the authors has been created with mW99**

In the final sentence of the abstract, the authors state:

> "In conclusion, the developed $\alpha$-based method has the capacity to derive ice thickness and snow depth, without relying on the snow depth information as input to the buoyancy equation for converting freeboard to ice thickness."

However, the method presented here works directly from ice freeboard data which can only be derived by relying on snow depth information (Sect. 5.1 & Eq. 15 of Kurtz et al., 2014).

I feel that what the authors would like to present is a way to convert *radar freeboards* to ice thickness without relying on snow depth data, and this should be done before publication. I think it is possible for the authors to adapt their processing chain to deal with this, although it may complicate things.

**2.3 Uncertainty Analysis**

The authors state in their Discussion and Conclusions section:

> Overall, the developed $\alpha$-based method yields ice thickness and snow depth, without relying on a priori 'uncertain' snow depth information, which results in uncertainty in the ice thickness retrieval.

They are of course correct to identify that uncertainty in snow depth leads to uncertainty in the ice thickness retrieval. To avoid having to quantify snow depth, they instead rely on a parameter equal to $h/H$, which they empirically derive from the temperature of the air-snow and ice-snow interfaces.

Clearly there is significant uncertainty in the value of $\alpha$, and the authors should try to quantify how this propagates through into uncertainty in ice thickness. It's possible that their $\alpha$ parameter is more uncertain than other published data for $h$, and if so this method will deliver lower quality estimates of $H$ than the traditional method.

It seems (looking at Fig. 1 of this review) that a given error in $\alpha$ would have a more serious impact on $H$ than the same error in $h$, because the gradients of the lines are much more similar in on the left panel of Fig. 1 than on the right. This issue scales with the alpha parameter (i.e. as the freeboard goes down), and at high $\alpha$ very small uncertainties in alpha will lead to large uncertainties in $H$.

As alpha becomes so large that the freeboard tends to zero (not that uncommon in the Atlantic sector of the Arctic), the method seems to lose its usefulness, whereas the traditional method continues to function. That is to say in the case of near-zero freeboard, the traditional method still provides an estimate of $H$, but that proposed by the authors does not (see Eq. 7 as $h_f \rightarrow 0$ ).

This is addressed in L 161/162, where a critical value is given for alpha, and it is explained that for alpha above this value data are not produced. How often does this occur? And what is the effect on $H$ of small errors in alpha just below this critical threshold?

**3 Minor Points**

**3.1 General**

- L21: The authors should consider directing the reader to Laxon et al. (2003) when illustrating that thickness has been estimated for nearly two decades.

- When discussing studies indicating the height difference between the scattering horizon and the snow-ice interface, the authors should consider directing the reader to Nandan et al. (2017) and Willatt et al. (2010, 2011).

- L62 & 64: Define RTM before using the acronym.

- The font sizes of some annotations to Figure 3 should be increased so as to be legible and comparable to the (a), (b), (c) lettering.

- In Fig. 2, the box that reads 'Find temperature discontinuity point'. It is my understanding that the temperature is continuous (but not a smooth function), and therefore it has no discontinuities (but its gradient does). Should this box then read 'Find temperature gradient discontinuity point'?

- I think the notation of $H$ and $h$ in combination with $h_f$, $h_{rf}$ and $h_{tf}$ is confusing to the casual reader. For instance, the fact that $h$ and $h_f$ look so similar but are in fact unrelated confused me initially. Even changing $H \to H_{ice}$ and $h \to h_{snow}$ would clarify this.

**3.2 Validation of $H$ against OIB Data**

The authors are able to create two products from freeboard data obtained by OIB and CryoSat-2, one for snow depth ($h$), and one for ice thickness ($H$). They then rightly try to assess the quality of these data products against other datasets, namely the OIB snow depth and ice thickness data.

There are at least five algorithms published to process the raw OIB radar returns into along-track snow depth data, and they produce a spread in the mean snow depth (Kwok et al., 2017). 'Validation' of a model or data implies comparison to true or certain values, and it is unclear which OIB snow depth product (if any) represents *the truth*. This limits the strength of the validation exercise. Nonetheless, I understand that OIB snow depth values have historically been taken as *the truth* in published work so this is a perhaps not a big issue. It might also be argued that the spread of different OIB data is sufficiently small relative to other methods of snow depth estimation to allow OIB to approximate the truth for validation purposes.

I feel that there is however a more significant issue with the authors' claims to have 'validated' their ice thickness data against OIB ice thickness data ($H_{OIB}$). OIB aircraft instruments do not measure thickness ($H_{true}$) directly, but instead estimate it based on freeboard, snow depth, snow density and ice density values. As such, OIB thickness data (while likely to be the most accurate data on $H_{true}$ outside of in-situ measurement), undoubtedly suffer from biases involving snow depth, snow density and ice density, and therefore should not be mistaken for $H_{true}$.

The technique for determining $H_{OIB}$ is very similar to that presented in this manuscript: the authors use identical freeboard, snow density and ice density values to estimate thickness with the hydrostatic equilibrium assumption. Given these similarities, comparing the thickness estimates in this paper with OIB thickness estimates doesn't really qualify as independent validation.

It seems more like the exercise of comparing $H$ estimates is in fact comparing the novel snow depth estimates with OIB snow depths (Fig 7 top row; a valuable analysis), and then investigating how that singular difference propagates into sea ice thickness estimates.

I suspect that the strong agreement between the two datasets presented in the middle row of Fig. 7. is largely a result of the identical radar freeboards and geophysical parameters used in each processing chain. After all, much of sea ice thickness is determined by radar freeboard information, independent of snow data. The fact that the 'simultaneous' method matches $H_{OIB}$ data more closely than MW99 is therefore evidence that the snow depth product produced by the 'simultaneous' method is closer to OIB snow depths than MW99 (because everything else is equal).

I think it is perfectly reasonable (and in fact expected) to compare $H$ estimates from the new method with $H_{OIB}$. However, I think this should be presented as a 'comparison with' or 'evaluation against' OIB data, rather that implying that the new data are being validated against some true value. It is also an understandable bit of reasoning to say that values which are closer to $H_{OIB}$ are likely to be closer to $H_{true}$, but if this assumption is made it should be stated explicitly.

**3.3 Limitations of Other Data**

L66 - 69:

> Other approaches worth mentioning are snow depth retrieval using dual-frequency altimetry (Guerreiro et al., 2016; Lawrence et al., 2018, Kwok and Markus, 2018), snow on sea ice model accumulating snowfall from reanalysis (Petty et al., 2018), multilinear regression (Kilic et al., 2019), and the neural network approach (Braakmann-Folgmann and Donlon, 2019). However, these methods do not satisfactorily meet the criteria required for freeboard to ice conversion over the entire Arctic Ocean basin scale or multi-year time scale.

The approach of Guerreiro et al. (2016) and Lawrence et al. (2018) are limited latitudinally by the AltiKa orbital inclination and Lawrence et al. (2018) additionally through calibration with OIB which only operates in Spring. As the authors identify, they are limited in spatial or temporal extent.

While there are limitations to the data products of Petty et al. (2018), Kilic et al. (2019) and Braakmann-Folgmann and Donlon (2019), it's not obvious that these can be characterised by failure to cover the entire basin on a multiyear timescale. As such, the statement on L69 that they do not satisfactorily meet these criteria should be clarified.

**3.4 Rainbow Color Schemes**

Where possible, authors should avoid presenting continuous data with 'rainbow' color schemes as in Figures 6 & 8. This is because (among other reasons) the scheme tends to imply sharp transitions in the data where they do not exist (Borland and Taylor, 2007). Alternatives for geoscientists are given by Light and Bartlein (2004), Stauffer et al. (2015) and Thyng et al. (2016).

**References**

Borland, D. and Taylor, R. M.: Rainbow color map (still) considered harmful, IEEE Computer Graphics and Applications, 27, 14–17, https://doi.org/10.1109/MCG.2007.323435, 2007.

Braakmann-Folgmann, A. and Donlon, C.: Estimating snow depth on Arctic sea ice using satellite microwave radiometry and a neural network, The Cryosphere, 13, 2421–2438, https://doi.org/10.5194/tc-13-2421-2019, URL https://www.the-cryosphere.net/13/2421/2019/, 2019.

Guerreiro, K., Fleury, S., Zakharova, E., Rémy, F., and Kouraev, A.: Potential for estimation of snow depth on Arctic sea ice from CryoSat-2 and SARAL/AltiKa missions, Remote Sensing of Environment, 186, 339–349, https://doi.org/10.1016/j.rse.2016.07.013, URL http://dx.doi.org/10.1016/j.rse.2016.07.013, 2016.

Kilic, L., Tage Tonboe, R., Prigent, C., and Heygster, G.: Estimating the snow depth, the snow-ice interface temperature, and the effective temperature of Arctic sea ice using Advanced Microwave Scanning Radiometer 2 and ice mass balance buoy data, Cryosphere, 13, 1283–1296, https://doi.org/10.5194/tc-13-1283-2019, 2019.

Kurtz, N. T., Galin, N., and Studinger, M.: An improved CryoSat-2 sea ice freeboard retrieval algorithm through the use of waveform fitting, Cryosphere, 8, 1217–1237, https://doi.org/10.5194/tc-8-1217-2014, URL https://www.the-cryosphere.net/8/1217/2014/, 2014.

Kwok, R., Kurtz, N. T., Brucker, L., Ivanoff, A., Newman, T., Farrell, S. L., King, J., Howell, S., Webster, M. A., Paden, J., Leuschen, C., MacGregor, J. A., Richter-Menge, J., Harbeck, J., and Tschudi, M.: Intercomparison of snow depth retrievals over Arctic sea ice from radar data acquired by Operation IceBridge, The Cryosphere, 11, 2571–2593, https://doi.org/10.5194/tc-11-2571-2017, URL https://www.the-cryosphere.net/11/2571/2017/, 2017.

Lawrence, I. R., Tsamados, M. C., Stroeve, J. C., Armitage, T. W., and Ridout, A. L.: Estimating snow depth over Arctic sea ice from calibrated dual-frequency radar freeboards, Cryosphere, 12, 3551–3564, https://doi.org/10.5194/tc-12-3551-2018, URL https://www.the-cryosphere.net/12/3551/2018/, 2018.

Laxon, S., Peacock, H., and Smith, D.: High interannual variability of sea ice thickness in the Arctic region, Nature, 425, 947–950, https://doi.org/10.1038/nature02050, 2003.

Light, A. and Bartlein, P. J.: The end of the rainbow? color schemes for improved data graphics, Eos, 85, https://doi.org/10.1029/2004EO400002, 2004.

Mallett, R. D. C., Lawrence, I. R., Stroeve, J. C., Landy, J. C., and Tsamados, M.: Brief communication: Conventional assumptions involving the speed of radar waves in snow introduce systematic underestimates to sea ice thickness and seasonal growth rate estimates, The Cryosphere, 14, 251–260, https://doi.org/10.5194/tc-14-251-2020, URL `https://www.the-cryosphere.net/14/251/2020/`, 2020.

Nandan, V., Geldsetzer, T., Yackel, J., Mahmud, M., Scharien, R., Howell, S., King, J., Ricker, R., and Else, B.: Effect of Snow Salinity on CryoSat-2 Arctic First-Year Sea Ice Freeboard Measurements, Geophysical Research Letters, 44, 419–10, https://doi.org/10.1002/2017GL074506, URL `http://doi.wiley.com/10.1002/2017GL074506`, 2017.

Petty, A. A., Webster, M., Boisvert, L., and Markus, T.: The NASA Eulerian Snow on Sea Ice Model (NESOSIM) v1.0: Initial model development and analysis, Geoscientific Model Development, 11, 4577–4602, https://doi.org/10.5194/gmd-11-4577-2018, 2018.

Stauffer, R., Mayr, G. J., Dabernig, M., and Zeileis, A.: Somewhere over the rainbow: How to make effective use of colors in meteorological visualizations, Bulletin of the American Meteorological Society, 96, 203–216, https://doi.org/10.1175/BAMS-D-13-00155.1, 2015.

Thyng, K. M., Greene, C. A., Hetland, R. D., Zimmerle, H. M., and DiMarco, S. F.: True colors of oceanography: Guidelines for effective and accurate colormap selection, https://doi.org/10.5670/oceanog.2016.66, 2016.

Willatt, R., Laxon, S., Giles, K., Cullen, R., Haas, C., and Helm, V.: Ku-band radar penetration into snow cover on Arctic sea ice using airborne data, Annals of Glaciology, 52, 197–205, https://doi.org/10.3189/172756411795931589, 2011.

Willatt, R. C., Giles, K. A., Laxon, S. W., Stone-Drake, L., and Worby, A. P.: Field investigations of Ku-band radar penetration into snow cover on antarctic sea ice, IEEE Transactions on Geoscience and Remote Sensing, 48, 365–372, https://doi.org/10.1109/TGRS.2009.2028237, URL `http://ieeexplore.ieee.org/document/5282596/`, 2010.

Review of 'Simultaneous estimation of wintertime sea ice thickness and snow depth from space-borne freeboard measurements', Shi Hoyeon et al.
By Isobel R Lawrence.

**Paper synopsis**

A methodology is presented for estimating sea ice thickness and snow depth (on sea ice) using ice or snow freeboard, and snow surface and snow/ice interface temperatures (available from satellite data). The method is based on the assumption that conductive heat flux through the snow/ice interface is continuous, therefore the bulk temperature difference across the ice layer, divided by the ice thickness, is proportional to the temperature difference across the snow layer divided by its depth. The proportionality is a function of the conductivities of ice and snow, but can be derived empirically using temperature profiles from drifting buoys. The authors find a piecewise (2-part) linear relationship between the ratio of the temperature differential across the snow layer to the temperature change through the ice, and the ratio of snow to ice thickness. They present a thorough analysis of the possible physical mechanisms behind the two-part function, in the appendix.

Using the derived relationship, the ratio of snow depth to ice thickness can be estimated purely from satellite measurements of snow surface and snow/ice interface temperature. Then, combined with either snow or ice freeboard, this ratio can be used to simultaneously estimate snow depth and ice thickness. The appeal of the methodology lies in the fact that independent estimates of snow depth are not required to calculate ice thickness. The authors compare their ice thickness and snow depth estimates to those from OIB, and also against 'CryoSat-like' and 'ICESat-like' ice thicknesses, where snow depth from the Warren climatology is implemented in the traditional freeboard-to-thickness buoyancy equations. Finally the method is extended to the basin scale, using satellite-derived snow surface and snow/ice interface temperatures and sea ice freeboard data from CryoSat-2.

The manuscript is very well written, coherent and extremely thorough in its explanations, however something major has been overlooked which I believe means that this method cannot be used to retrieve snow depth and ice thickness from *radar* altimetry. That said, the method remains applicable to laser altimetry, e.g. IceSat-2, and is therefore still very valuable. I therefore recommend the paper for publication following this major revision:

**Major comments**

In order to use this methodology with satellite data from CryoSat-2, ice freeboard (the elevation of the snow/ice interface above the ocean surface) is required. However it is impossible to retrieve ice freeboard from CryoSat-2 without a-priori knowledge of the snow layer. Since the radar pulse slows down as it travels through the snow, snow depth is required in order to correct for the slower speed of propagation and estimate sea ice freeboard.

To compound the issue, the equation to convert radar freeboard into ice freeboard is incorrectly reported in a number of studies, including that of Kurtz et al. (2014, eq. 16) which describes the CS2 ice freeboard dataset you use in your analysis. Please see Mallett et al. (2020) for the correct derivation of the equation and details of its misreporting in the literature.

The correct equation for sea ice thickness from *radar* altimetry (assuming full snow penetration) is:

$$H = \left( f_r + h \left( \frac{c}{c_s} - 1 \right) \right) \left( \frac{\rho_w}{\rho_w - \rho_i} \right) + h \left( \frac{\rho_s}{\rho_w - \rho_i} \right),$$

where $f_r$ is radar freeboard, as estimated from radar altimeters like CryoSat-2. If this equation cannot be solved by the proposed methodology (I do not believe it can be), then the paper should be restructured to focus on the laser case. The methodology remains valid for use with snow freeboards, and these are available from ICESat and now ICESat-2, so perhaps section 4.3 could be changed to an application to ICESat data. I appreciate that this will require a substantial amount of work, which is why I consider this revision major. However I find this methodology novel and valuable and the results in section 4.2 are encouraging; I would like to reiterate therefore that I think the paper deserves publication subject to this alteration and the following minor revisions:

**Minor comments**

I think you need to include an uncertainty budget for your sea ice thickness and snow depth estimates.

L30: "However, the radar scattering horizon is often treated as the snow–ice interface". Include Hendricks et al. (2016), Guerreiro et al. (2017), Tilling et al. (2018) as refs here since AWI, LEGOS and CPOM CryoSat-2 ice thickness products all make the same assumption.

L72: …"for given densities and freeboard" – (and assuming no snow penetration for laser and full snow penetration for radar)

L138: Could you say how many are discarded based on this criterion, and out of how many total.

L159: Can you provide a reference for the OIB data processing document where the densities are given?

L160: I understand that you keep ice density constant in order to compare with OIB data, but later when comparing with satellite-derived ice thickness should you not then use the densities used in those products for a fair comparison?

L198: "It was reported…" – By who?

L200: Where does $T_{si}$ for March come from if the Lee et al. (2018) dataset is only December-February ?

L201: "…if data frequency is over 20". Do you mean if 20 days out of the month contain data? Or are you referring to a number of points per grid cell?

L205: Please could you provide the details and a reference for which OIB dataset you used and where it is available from? i.e. L2, L4, Quicklook?

Figure 4: Why do you choose to show us the 7-day averaged plot in Fig.4 when Figure 3 was showing 15-day averaged temperature profiles?

L235: At the end of this sentence you could refer the reader to the appendix.

L244: bias is not near-zero in Fig 4b, it *is* zero.

L269: Did you calculate a different alpha for each year, and apply the different alpha to each year of OIB data? Or did you just average all the years together? Please clarify this in the text.

L295: Do you get the MW99 for input into Eqs. (4) and (5) from the CS2 data? If so is it monthly grid-averaged? How do you assign each OIB point a snow depth?

L301: "Therefore, if there are decreasing trends in both ice thickness and snow depth, the decreasing trend of ice thickness estimated from the constant snow depth will be diminished in radar, while being amplified in lidar" – This sentence seems overcomplicated. To me, all that the bottom two plots of Figure 7 demonstrate is that MW99 snow depth is larger than OIB snow depth. For the laser case, this means that using W99 causes ice thickness to be *underestimated* compared to H(OIB), and for the radar case using W99 results in ice thicknesses that are *too thick* compared to OIB. Perhaps you could plot MW99 against h(OIB) to clarify this? The retrieval of sea ice thickness from ICESat has not traditionally used the Warren climatology- see Kwok and Cunningham (2008) and Petty et al. (2020). Therefore I don't think it's justified to call this 'ICESat-like thickness' since you are not using the same snow depth product that they do.

**Typos / Grammar**

L128: "Therefore, the interface searching algorithm…" -> "Therefore, an interface searching algorithm…"

L165: "Sea ice thicknesses converted from MW99 using Eqs. (4) and (5) are also compared to examine how simultaneous retrievals…" **->** "Sea ice thicknesses are also calculated from Eqs. (4) and (5), using MW99 as snow depth, to examine how simultaneous retrievals…"

L194: "This reformatted AASTI-v2 data are called…" -> "This reformatted AASTI-v2 dataset is called…"

L293: "Examining how the current practices of retrieving the sea ice thickness through ICESat and CS2 measurements are compared with the simultaneous retrievals is of interest" **->** "We now examine how the current practises of retrieving sea ice thickness from ICESat and CS2 measurements compare with our method."

L294: "In doing so, OIB-measured…" -> "To do so, OIB-measured…"

L297: "Apparently, ICESat-like thickness tends…." -> "According to our analysis, ICESat-like thickness tends…."

L416: "…which are hard to be quantified explicitly." -> "…which are hard to quantify explicitly."

---

## Editor Decision (ED2)

2nd review of 'Simultaneous estimation of wintertime sea ice thickness and snow depth from space-borne freeboard measurements', Shi Hoyeon et al.
By Isobel R Lawrence

**General comment**

The authors have done a thorough job responding to and addressing my comments. In particular, the application of the methodology to *radar* freeboard from CryoSat-2 has been demonstrated (I apologise for assuming it would not be possible!). The only thing outstanding is an uncertainty estimate for the CS2-derived snow depth and ice thickness. The sensitivity analysis that has been included in the appendix explores the impact $\Delta\alpha$ has on snow depth and sea ice thickness, but this alone is not sufficient to describe the uncertainty on these products since, for example, error on radar freeboard should also be accounted for. If these products are to be made available, in particular to the modelling community, they must come with an error budget.

The methodology presented in this manuscript is a novel and valuable addition to the sea ice community. I therefore recommend the paper for publication following minor revisions.

**Minor revisions**

**The accuracies of CS2 retrievals / Error budget**

I find the final paragraph of section 4.3 (L359-369) extremely difficult to follow. Indeed, after studying Figure 9 for some ten minutes I am still at a loss as to what these plots actually tell us. If the aim is to perform a validation against OIB measurements, why not just show a scatter plot of $h_s^{OIB}$ vs $h_s(\alpha^{sat}, F_r^{CS2})$ and $H_i^{OIB}$ vs $H_i(\alpha^{sat}, F_r^{CS2})$? That would be a far simpler and more relevant plot which the reader will understand immediately.

I think an uncertainty estimate for the satellite-derived products needs including in section 4.3. A simple propagation of errors could be performed on Equation 12 to estimate the error on $H_i$, and similarly error on $h_s$ could then be propagated from Equation 3. For this, the uncertainty on radar freeboard and $\alpha$ are required. In the manuscript $\Delta\alpha$ is estimated to be 0.036, equal to the RMSE between observed and regressed $\alpha$, where regressed $\alpha$ are derived from buoy-measured interface temperatures. Does the same $\Delta\alpha$ apply for $\alpha$ derived from satellite temperatures? Evidently errors in $T_{si}^{sat}$ and $T_{as}^{sat}$ will result in errors in $\alpha$. This should at least be discussed in section 4.3, even if it is not possible to incorporate errors on $T_{si}^{sat}$ and $T_{as}^{sat}$ into the final uncertainty budget (if for example the satellite temperature products do not come with an uncertainty). Errors on radar freeboard should be available with the CS2 product you are using. If not, see discussion in Tilling et al (2018) for their estimate of CS2 radar freeboard uncertainty.

**Other minor comments**

L15: "retrieved ice thickness was found to be better than the methods relying on the use of snow depth climatology as input, in terms of mean bias and RMSE." - This is

not true, RMSE on ice thickness from radar freeboard is smaller using the MW99 method (0.344 vs 0.5 $\alpha$-method, figure 7)

L68: "Other satellite remote sensing approaches include the snow depth retrieval using dual-frequency altimetry (Guerreiro et al., 2016; Lawrence et al., 2018, 70 Kwok and Markus, 2018), multilinear regression (Kilic et al., 2019), and a neural network approach (Braakmann-Folgmann and Donlon, 2019)." – I think here you need to add something about the limitations of these methods. Otherwise it is unclear why a new snow product is necessary.

L170: "A sensitivity test indicated that the influence of a 0.3°C difference in the freezing temperature on $\alpha$ was negligible". Could you give a percentage value or some quantification of it being negligible?

L233: The Quicklook dataset URL you provide takes you to 'Bootstrap Sea ice concentrations". Please check the DOI. Also I suggest moving the url to the end of the sentence.

L235: "The OIB data are also reformatted into the 25 km grid format for comparison. If the location of one OIB individual data point falls within a certain 25 km grid area, then the point data is binned in a corresponding grid. After completing the grid assignment, grid value is determined by calculating a simple arithmetic mean of all data within that grid area." – Do you just mean "the OIB data are averaged on the same 25km grid" ?

L251: I find lines 245 to 250 slightly confusing. I suggest you move the equation for $\eta_s$ to after equation 14. I.e:
"[Eq 14],
where $\eta_s$ = […] and $\rho_s$ is taken from the Warren climatology, after Kurtz (2017)"

Figure 3 caption: "The period number is equivalent to the number of time-averaging bin." – I do not understand what the period number is.

**Typos / Grammar**

L19: "…buoyancy equation and radar penetration…" -> "…buoyancy equation or the radar penetration…"

L26: "…the height from the sea surface in cracks and leads to the snow surface." -> "…the height from the sea surface in leads, to the snow surface."

L51: "variation of snow–ice system" -> "variation of the snow–ice system"

L58: "TB's" -> "TBs"

L156: Remove "respectively"

L181: "…by multiplying the obtained sea ice thickness and $\alpha$." -> "…by multiplying the obtained sea ice thickness and $\alpha$ (Eq. (3))."

L194 "…as parts…" -> "…as part…"

L201: "depending" -> "dependent"

L248: "In this dataset, $\eta_s$ was parameterized as a function of the snow density" -> "$\eta_s$ was parameterized as a function of the snow density" – The 'in this dataset' suggests to me that you mean your dataset!

L253: "…values are used for comparison." -> "…values are used for comparison with results from our simultaneous method."

L307: "scatterplots of comparing retrievals" -> "scatterplots comparing retrievals"

L351: "shows $\alpha$ values that is generally larger than that over" -> "shows $\alpha$ values that are generally larger than those over"

L353: "$H_i$ shows a similar geographical distribution as shown in the freeboard (the first row)" -> "$H_i$ shows a similar geographical distribution to radar freeboard (the first row)"

L356: "and results are given at the bottom" -> "and results are shown in the bottom row"

L356: "The obtained snow distribution indicates that thicker snow areas are generally coincident with thicker MYI areas. Likewise, the thinner snow area coincides with the thinner FYI area" -> "The obtained snow distribution indicates that thicker (thinner) snow areas are generally coincident with thicker MYI (thinner FYI) areas."

L385: "As a matter of fact, the ice thickness results were more accurate than they were from the current retrieval methods relying on the input of snow depth (this time MW99 snow climatology), in terms of mean bias and RMSE." – This sentence is not accurate. RMSE on ice thickness from radar freeboard is smaller using the MW99 method (0.344 vs 0.5 $\alpha$-method, figure 7)

L406: "The results that radar freeboard and the total freeboard yielded had nearly the same outputs when the $\alpha$ -approach was used" – This sentence does not make sense to me.

L455: "hard wind slap" -> "hard wind slab"

L471: "Because $h_s$ is a combination of freeboard and $\alpha$" - Do you mean "Because $H_i$ is a combination of freeboard and $\alpha$" ?

L476: "With $\Delta\alpha = \pm0.03$, which is an RMSE range in the $\alpha$ -prediction equation" – From figure 4b, the RMSE = 0.04, not 0.03.

L483: "a much small number" -> "a much smaller number"

Figure 8 caption: "Grey areas in the second row denote where $\alpha$ retrieval is failed because $T_{as}$ is warmer than $T_{as}$." -> "Grey areas in the second row denote where $\alpha$ retrieval failed because $T_{as}$ is warmer than $T_{si}$."

---

## Author Response (AR3)

Authors appreciate thoughtful and constructive comments and suggestions given by anonymous referee #1 and Isobel R. Lawrence. By resolving the issues raised by the referee, the authors believe that the logical flow of this study was strengthened, especially by counting the radar freeboard.

5     Please find the following point-by-point responses to the referee's comments below. A marked-up version of the revised manuscript regarding the changes is attached at the end of this authors' response.

**Response to anonymous referee #1**

**1. Major Points**

10    **1.1. Difference between radar freeboard and scattering horizon height**

*L22: The statement "altimeters . . . measure sea ice freeboard" is only approximately correct in the case of radar altimeters. The instrument on board CS2 measures a time of flight, which can be related to the height of some radar scattering horizon only where no snow lies in between the scattering horizon and the instrument. When snow is in between and fully penetrated by the radar, the radar range to the scattering surface is*
15    *overestimated due to slower pulse propagation in the overlying snow. Correcting for this and estimating the height of the ice-snow interface requires knowledge of the overlying snow (Mallett et al., 2020).*

*This issue surfaces again when the authors identify the radar freeboard as the height from the sea surface to the radar scattering horizon in L28. This is only the case for bare ice. Where overlying snow is present and fully penetrated, the radar freeboard is a finite distance below the ice freeboard (the assumed scattering surface). In*
20    *the freeboard product used in this manuscript (Kurtz et al., 2014) this displacement is $\underline{h_s}(1 - c_s/c)$.*

*This is relevant to Fig. 1, where $h_{rf}$ is depicted as being above the ice freeboard. While it may be true that the radar scattering horizon is above the snow-ice interface, in products that assume full radar penetration of the snowpack the radar freeboard is lower than the ice freeboard. Theoretically for total radar penetration and a freeboard depressed to near the water by snow, the radar freeboard can be below the waterline (while the ice*
25    *freeboard and scattering horizon are above).*

We became aware of that the definition of 'radar freeboard' used in the manuscript was not consistent with the one used in CS2 data-related references (e.g., Kurtz et al., 2014), which is difference in the radar ranging between the sea surface and the radar scattering horizon. Therefore, Fig. 1 in the manuscript was modified, by following your suggestion on using different notations for clarity.

30    Now, the system includes correction terms regarding the wave propagation speed change in the snow layer ($F_c$), and the displacement of the scattering horizon from the ice surface ($F_p$) following Kwok and Cunningham (2015) and Armitage and Ridout (2015).

$$F_i = F_r + \left(F_c - F_p\right) \tag{AR1}$$

$$F_c = (\eta_s - 1)\, f h_s \tag{AR2}$$

35    $$F_p = (1 - f)h_s \tag{AR3}$$

Here, $\eta_s$ denotes the refractive index of snow layer ($\eta_s = c/c_s$) and $f$ denotes the radar penetration

factor, which is the depth of the radar scattering horizon relative to the snow depth (e.g. $f = 1$ if the radar scattering horizon is at snow–ice interface and $f = 0$ if the radar scattering horizon is at air-snow interface), respectively. Combining three equations yields the following relationship.

40 $$F_i = F_r + (f\eta_s - 1)h_s \tag{AR4}$$

In the revision, new formulae are introduced for simultaneously solving snow depth and ice thickness even for radar-based freeboard measurements. These changes can be found in introduction, Sect. 2.3 and Fig. 1 in the revised manuscript.

45 **1.2 The freeboard product used by the authors has been created with mW99**

*In the final sentence of the abstract, the authors state:*

> *"In conclusion, the developed α-based method has the capacity to derive ice thickness and snow depth, without relying on the snow depth information as input to the buoyancy equation for converting freeboard to ice thickness."*

50 *However, the method presented here works directly from ice freeboard data which can only be derived by relying on snow depth information (Sect. 5.1 & Eq. 15 of Kurtz et al., 2014).*

*I feel that what the authors would like to present is a way to convert radar freeboards to ice thickness without relying on snow depth data, and this should be done before publication. I think it is possible for the authors to adapt their processing chain to deal with this, although it may complicate things.*

55 Thanks for the suggestion which in fact led us to deeper understanding of radar altimeter remote sensing. Luckily, we were able to include this issue in the retrieval, by modifying Eqs. (5) and (7) of the first version of manuscript (see below for Eqs. (5) and (7) written with new notation), using Eq. (AR4).

$$\text{Eq. (5): } H_i = \left(\frac{\rho_w}{\rho_w - \rho_i}\right) F_i + \left(\frac{\rho_s}{\rho_w - \rho_i}\right) h_s$$

60 $$\text{Eq. (7): } H_i = \frac{\rho_w}{\rho_w - \rho_i - \alpha\rho_s} F_i$$

It was possible because Eq. (AR4) does not include additional unknowns, for given parameterization and assumption on the radar penetration. We assumed $f = 0.84$ for CS2 (Armitage and Ridout, 2015). $\eta_s$ can be parameterized as a function of the snow density, i.e., $\eta_s=(1+0.51\rho_s)^{1.5}$ (Ulaby et al., 1986). Here we present how the equations were solved.

65 First, the traditional method for the ice thickness retrieval with the snow depth as input can be written in the following equation by substituting $F_i$ with $F_r$ using Eq. (AR4).

$$H_i = \frac{\rho_w}{\rho_w - \rho_i} F_r + \frac{(f\eta_s - 1)\rho_w + \rho_s}{\rho_w - \rho_i} h_s \tag{AR5}$$

Then, substituting $h_s$ with $\alpha H_i$ and rearranging the equation yield the equation for $H_i$ as a function of radar freeboard and $\alpha$, without snow depth information.

$$\quad H_i = \frac{\rho_w}{\rho_w - \rho_i - \alpha\{(f\eta_s - 1)\rho_w + \rho_s\}} F_r \qquad (AR6)$$

Note that Eq. (AR6) becomes equivalent to the equation for the total freeboard if $f = 0$ (no wave penetration into snow layer).

This new setup requires the data processing chain to be modified as well. Here we describe what changes were made (in Sect. 3.3 and 3.4 of the revised manuscript). First, CS2-like radar freeboard was derived from OIB total freeboard ($F_t^{OIB}$) and snow depth ($h_s^{OIB}$). From Eq. (AR4) and the relationship $F_i = F_t - h_s$, the radar freeboard can be expressed as follows.

$$F_r^{OIB} = F_t^{OIB} - h_s^{OIB} - (f\eta_s - 1)h_s^{OIB} \qquad (AR7)$$

Because the main objective of using OIB data is to evaluate the relative performance of the simultaneous retrieval method when the method is applied to CS2 data, the radar penetration factor ($f$) for OIB data processing was also set to be 0.84 (Artimage and Ridout, 2015). In the data processing chain, $h_s^{OIB}$ was removed if it is smaller than the given uncertainty level of the dataset (~5.7 cm) or it is larger than the total freeboard $F_t^{OIB}$.

The CS2 radar freeboard ($F_r^{CS2}$) was obtained from CS2 ice freeboard dataset. The CS2 ice freeboard data ($F_i^{CS2}$) distributed by NSIDC (Kurtz et al. 2017) assume that radar scattering horizon locates at snow–ice interface and applies a wave propagation speed correction. However, the correction was made using the MW99 snow depth ($h_s^{MW99}$) climatology with an erroneous correction form of $h_c = (1 - \eta_s^{-1}) h_s$, instead of the correct correction form of $h_c = (\eta_s - 1) h_s$ (Mallet et al. 2020). Thus, at this point, it is straightforward to derive the CS2 radar freeboard by removing the correction term as in the following equation.

$$\quad F_r^{CS2} = F_i^{CS2} - (1 - \eta_s^{-1})h_s^{MW99} \qquad (AR8)$$

Finally, analyses in the first version of the manuscript were conducted again using the radar freeboard rather than using the ice freeboard. This time SIC criteria for $\alpha$ calculation was set to be 95% (original: 98%) for a wider coverage. Figs. 6–9 in the revised manuscript are the reprocessed results. Despite of these changes, we find little changes in the conclusions we made in the first version of the manuscript. In addition, for more comprehensive information, snow depth comparison results are provided in Fig. 9.

The changes for method and data can be found in Sect. 2.3 and in Sect. 3.3–3.4, respectively. The corresponding results can be found in Sect. 4.2 and 4.3 and Figs. 6–9.

**1.3 Uncertainty Analysis**

*The authors state in their Discussion and Conclusions section:*

> *Overall, the developed α-based method yields ice thickness and snow depth, without relying on a priori 'uncertain' snow depth information, which results in uncertainty in the ice thickness retrieval.*

105 *They are of course correct to identify that uncertainty in snow depth leads to uncertainty in the ice thickness retrieval. To avoid having to quantify snow depth, they instead rely on a parameter equal to h/H, which they empirically derive from the temperature of the air-snow and ice-snow interfaces.*

*Clearly there is significant uncertainty in the value of α, and the authors should try to quantify how this propagates through into uncertainty in ice thickness. It's possible that their α parameter is more uncertain than*
110 *other published data for h, and if so this method will deliver lower quality estimates of H than the traditional method.*

*It seems (looking at Fig. 1 of this review) that a given error in α would have a more serious impact on H than the same error in h, because the gradients of the lines are much more similar in on the left panel of Fig. 1 than on the right. This issue scales with the alpha parameter (i.e. as the freeboard goes down), and at high α very*
115 *small uncertainties in alpha will lead to large uncertainties in H.*

*As alpha becomes so large that the freeboard tends to zero (not that uncommon in the Atlantic sector of the Arctic), the method seems to lose its usefulness, whereas the traditional method continues to function. That is to say in the case of near-zero freeboard, the traditional method still provides an estimate of H, but that proposed by the authors does not (see Eq. 7 as $h_f \rightarrow 0$).*

120 *This is addressed in L161/162, where a critical value is given for alpha, and it is explained that for alpha above this value data are not produced. How often does this occur? And what is the effect on H of small errors in alpha just below this critical threshold?*

To identify the uncertainty of simultaneous method, snow depth error ($\Delta h_s$) equivalent to $\alpha$ error ($\Delta\alpha$) was calculated for chosen three typical sea ice conditions (thicker / moderate / thinner). The
125 simultaneous method showed a small sensitivity to $\Delta\alpha = 0.03$, which is RMSE value of the regression equation when total freeboard was used. On the other hand, sensitivity was greater when radar freeboard was used, especially for thinner ice where $\alpha$ is close to $\alpha_{crit}$. In case of ice thickness error ($\Delta H_i$), gap of sensitivity between total freeboard and radar freeboard methods was reduced because $\Delta H_i$ is more sensitive to $\Delta h_s$ when the total freeboard is used. This characteristic of sensitivity is
130 consistent with results from OIB analysis.

Majority of data used in this study belong to moderate or thicker ice and retrieved $\alpha$ rarely exceeds $\alpha_{crit}$. Therefore, there would not be many cases having great uncertainty that might be expected from thinner ice condition. As a matter of fact, it seems that retrieved $\alpha$ shows reasonable values upon presumed thermodynamic condition. Areas where thermodynamic condition is not met are located at
135 around the marginal ice zones and in the east of Greenland. Details can be found in Appendix B in the revised manuscript. In addition, we clarify that "a priori 'uncertain' snow depth in formation' is MW99 snow depth climatology.

**2. Minor Points**

140 ### 2.1. General

*• L21: The authors should consider directing the reader to Laxon et al. (2003) when illustrating that thickness has been estimated for nearly two decades.*

Laxon et al. (2003) is now included in the revised manuscript.

• *When discussing studies indicating the height difference between the scattering horizon and the snow-ice interface, the authors should consider directing the reader to Nandan et al. (2017) and Willatt et al. (2010, 2011).*

Nandan et al. (2017) and Willatt et al. (2011) are now referred in the revised manuscript. Because characteristic of sea ice is different between Arctic and Antarctic, study on Antarctic sea ice by Willatt et al. (2010) is not included.

• *L62 & 64: Define RTM before using the acronym*

The acronym 'RTM' stands for 'Radiative Transfer Model'. It is now defined in the revised manuscript.

• *The font sizes of some annotations to Figure 3 should be increased so as to be legible and comparable to the (a), (b), (c) lettering.*

Annotations are now increased to be legible (see Fig. 3 in the revised manuscript).

• *In Fig. 2, the box that reads 'Find temperature discontinuity point'. It is my understanding that the temperature is continuous (but not a smooth function), and therefore it has no discontinuities (but its gradient does). Should this box then read 'Find temperature gradient discontinuity point'?*

Yes, what discontinuous is temperature gradient, not temperature. The text now reads 'Find temperature gradient discontinuity point' (see Fig. 2 in the revised manuscript).

• *I think the notation of H and h in combination with $h_f$, $h_{rf}$ and $h_{tf}$ is confusing to the casual reader. For instance, the fact that h and $h_f$ look so similar but are in fact unrelated confused me initially. Even changing $H \rightarrow H_{ice}$ and $h \rightarrow h_{snow}$ would clarify this.*

Following your comment, we have changed our notations throughout the revised manuscript (See Fig. 1 in the revised manuscript).

**2.2. Validation of H against OIB Data**

*The authors are able to create two products from freeboard data obtained by OIB and CryoSat-2, one for snow depth (h), and one for ice thickness (H). They then rightly try to assess the quality of these data products against other datasets, namely the OIB snow depth and ice thickness data. There are at least five algorithms published to process the raw OIB radar returns into along-track snow depth data, and they produce a spread in the mean snow depth (Kwok et al., 2017). 'Validation' of a model or data implies comparison to true or certain values, and it is unclear which OIB snow depth product (if any) represents the truth. This limits the strength of the validation exercise. Nonetheless, I understand that OIB snow depth values have historically been taken as the truth in published work so this is a perhaps not a big issue. It might also be argued that the spread of different*

180    *OIB data is sufficiently small relative to other methods of snow depth estimation to allow OIB to approximate*
*the truth for validation purposes.*

*I feel that there is however a more significant issue with the authors' claims to have 'validated' their ice thickness*
*data against OIB ice thickness data ($H_{OIB}$). OIB aircraft instruments do not measure thickness ($H_{true}$) directly,*
*but instead estimate it based on freeboard, snow depth, snow density and ice density values. As such, OIB*

185    *thickness data (while likely to be the most accurate data on $H_{true}$ outside of in-situ measurement), undoubtedly*
*suffer from biases involving snow depth, snow density and ice density, and therefore should not be mistaken for*
*$H_{true}$.*

*The technique for determining $H_{OIB}$ is very similar to that presented in this manuscript: the authors use identical*
*freeboard, snow density and ice density values to estimate thickness with the hydrostatic equilibrium assumption.*

190    *Given these similarities, comparing the thickness estimates in this paper with OIB thickness estimates doesn't*
*really qualify as independent validation.*

*It seems more like the exercise of comparing H estimates is in fact comparing the novel snow depth estimates*
*with OIB snow depths (Fig 7 top row; a valuable analysis), and then investigating how that singular difference*
*propagates into sea ice thickness estimates. I suspect that the strong agreement between the two datasets*

195    *presented in the middle row of Fig. 7. is largely a result of the identical radar freeboards and geophysical*
*parameters used in each processing chain.*

*After all, much of sea ice thickness is determined by radar freeboard information, independent of snow data.*
*The fact that the 'simultaneous' method matches $H_{OIB}$ data more closely than MW99 is therefore evidence that*
*the snow depth product produced by the 'simultaneous' method is closer to OIB snow depths than MW99*

200    *(because everything else is equal).*

*I think it is perfectly reasonable (and in fact expected) to compare H estimates from the new method with $H_{OIB}$.*
*However, I think this should be presented as a 'comparison with' or 'evaluation against' OIB data, rather that*
*implying that the new data are being validated against some true value. It is also an understandable bit of*
*reasoning to say that values which are closer to $H_{OIB}$ are likely to be closer to $H_{true}$, but if this assumption is*

205    *made it should be stated explicitly.*

    We agree upon your notion that snow depth and ice thickness comparisons are the same problem.
To address your comment, we first clarified how $H_i^{OIB}$ is calculated in Sect 3.3. Then, we changed the
subtitle of Sect 4.2 from '*Validation* against OIB estimates' to '*Evaluation* against OIB estimates'.
Finally, *validation* on ice thickness contents were modified in the direction to address that the estimated

210    snow depth showing a more consistency with $h_s^{OIB}$ implies improved ice thickness. Accordingly, the
snow depth comparisons between $h_s(\alpha^{sat}, F_r^{OIB})$ vs. $h_s^{MW99}$ and $h_s(\alpha^{sat}, F_r^{CS2})$ vs. $h_s^{MW99}$ are included
in Fig. 9 in the revised manuscript.

**2.3. Limitations of Other Data**

215    *L66 - 69: Other approaches worth mentioning are snow depth retrieval using dual-frequency altimetry*
*(Guerreiro et al., 2016; Lawrence et al., 2018, Kwok and Markus, 2018), snow on sea ice model accumulating*
*snowfall from reanalysis (Petty et al., 2018), multilinear regression (Kilic et al., 2019), and the neural network*
*approach (Braakmann-Folgmann and Donlon, 2019). However, these methods do not satisfactorily meet the*
*criteria required for freeboard to ice conversion over the entire Arctic Ocean basin scale or multi-year time*

 *scale.*

*The approach of Guerreiro et al. (2016) and Lawrence et al. (2018) are limited latitudinally by the AltiKa orbital inclination and Lawrence et al. (2018) additionally through calibration with OIB which only operates in Spring. As the authors identify, they are limited in spatial or temporal extent. While there are limitations to the data products of Petty et al. (2018), Kilic et al. (2019) and Braakmann-Folgmann and Donlon (2019), it's not obvious that these can be characterized by failure to cover the entire basin on a multiyear timescale. As such, the statement on L69 that they do not satisfactorily meet these criteria should be clarified.*

The purpose of this paragraph was to introduce recent researches to readers. Therefore, the last sentence describing limitations of other data, which is not necessary, was removed. In case of Petty et al. (2018), we decided to not include it in the text to keep manuscript's focus on remote sensing, as characteristic of their product seems to be closer to model than remote sensing.

**2.4. Rainbow Color Schemes**

*Where possible, authors should avoid presenting continuous data with 'rainbow' color schemes as in Figures 6 & 8. This is because (among other reasons) the scheme tends to imply sharp transitions in the data where they do not exist (Borland and Taylor, 2007). Alternatives for geoscientists are given by Light and Bartlein (2004), Stauffer et al. (2015) and Thyng et al. (2016).*

Thanks for valuable comment. We changed our color scheme to generate figures from 'jet (rainbow)' from 'viridis', which is perceptually uniform colormap, available in matplotlib/python. Figs. 6, 8, and A1 in the revised manuscript are new plots with the new color scheme.

**Response to Isobel R. Lawrence**

**1. Major comments**

*In order to use this methodology with satellite data from CryoSat-2, ice freeboard (the elevation of the snow/ice interface above the ocean surface) is required. However, it is impossible to retrieve ice freeboard from CryoSat-2 without a-priori knowledge of the snow layer. Since the radar pulse slows down as it travels through the snow, snow depth is required in order to correct for the slower speed of propagation and estimate sea ice freeboard.*

*To compound the issue, the equation to convert radar freeboard into ice freeboard is incorrectly reported in a number of studies, including that of Kurtz et al. (2014, eq. 16) which describes the CS2 ice freeboard dataset you use in your analysis.*

*Please see Mallett et al. (2020) for the correct derivation of the equation and details of its misreporting in the literature. The correct equation for sea ice thickness from radar altimetry (assuming full snow penetration) is:*

$$H = \left( f_r + h\left(\frac{c}{c_s} - 1\right) \right)\left(\frac{\rho_w}{\rho_w - \rho_i}\right) + h\left(\frac{\rho_s}{\rho_w - \rho_i}\right),$$

*where $f_r$ is radar freeboard, as estimated from radar altimeters like CryoSat-2. If this equation cannot be solved by the proposed methodology (I do not believe it can be), then the paper should be restructured to focus on the*

*laser case. The methodology remains valid for use with snow freeboards, and these are available from ICESat and now ICESat-2, so perhaps section 4.3 could be changed to an application to ICESat data. I appreciate that this will require a substantial amount of work, which is why I consider this revision major. However I find this methodology novel and valuable and the results in section 4.2 are encouraging; I would like to reiterate therefore that I think the paper deserves publication subject to this alteration and the following minor revisions:*

Thanks for the comment on the fact that the radar algorithm depends on the snow depth, even before retrieving the snow depth. As a matter of fact, another reviewer raised the similar concern, and thus following responses are nearly same.

Recognizing the problems related to the radar altimetry, we modified equations for the model system to handle the radar freeboard as well. The modified model system is delineated in Fig. 1 (with slightly changed notations).

Now, the system includes correction terms regarding the wave propagation speed change in the snow layer ($F_c$), and the displacement of the scattering horizon from the ice surface ($F_p$) following Kwok and Cunningham (2015) and Armitage and Ridout (2015).

$$F_i = F_r + \left(F_c - F_p\right) \tag{AR1}$$

$$F_c = (\eta_s - 1)\,f h_s \tag{AR2}$$

$$F_p = (1 - f)h_s \tag{AR3}$$

Here, $\eta_s$ denotes the refractive index of snow layer ($\eta_s = c/c_s$) and $f$ denotes the radar penetration factor, which is the depth of the radar scattering horizon relative to the snow depth (e.g. $f = 1$ if the radar scattering horizon is at snow–ice interface and $f = 0$ if the radar scattering horizon is at air-snow interface), respectively. Combining three equations yields the following relationship.

$$F_i = F_r + (f\eta_s - 1)h_s \tag{AR4}$$

Luckily, we were able to include this issue in the retrieval, by modifying Eqs. (5) and (7) of the first version of manuscript (see below for Eqs. (5) and (7) written with new notation), using Eq. (AR4).

Eq. (5): $H_i = \left(\dfrac{\rho_w}{\rho_w - \rho_i}\right)F_i + \left(\dfrac{\rho_s}{\rho_w - \rho_i}\right)h_s$

Eq. (7): $H_i = \dfrac{\rho_w}{\rho_w - \rho_i - \alpha\rho_s}F_i$

It is because Eq. (AR4) does not include additional unknowns, for given parameterization and assumption on the radar penetration. We assumed $f = 0.84$ for CS2 (Armitage and Ridout, 2015). $\eta_s$ can be parameterized as a function of the snow density, i.e., $\eta_s = (1 + 0.51\rho_s)^{1.5}$ (Ulaby et al., 1986). Here we present how the equations were solved.

First, the traditional method for the ice thickness retrieval with the snow depth as input can be written in the following equation by substituting $F_i$ with $F_r$ using Eq. (AR4).

$$H_i = \frac{\rho_w}{\rho_w - \rho_i}F_r + \frac{(f\eta_s - 1)\rho_w + \rho_s}{\rho_w - \rho_i}h_s \tag{AR5}$$

Please note that this equation is equivalent to the equation assuming full snow penetration which

you presented. Then, substituting $h_s$ with $\alpha H_i$ and rearranging the equation yield the equation for $H_i$ as a function of radar freeboard and $\alpha$, without snow depth information.

$$H_i = \frac{\rho_w}{\rho_w - \rho_i - \alpha\{(f\eta_s - 1)\rho_w + \rho_s\}} F_r \tag{AR6}$$

Also note that Eq. (AR6) becomes equivalent to the equation for the total freeboard if $f = 0$ (no wave penetration into snow layer).

This new setup requires the data processing chain to be modified as well. Here we describe what changes were made (in Sect. 3.3 and 3.4 of the revised manuscript). First, CS2-like radar freeboard was derived from OIB total freeboard ($F_t^{OIB}$) and snow depth ($h_s^{OIB}$). From Eq. (AR4) and the relationship $F_i = F_t - h_s$, the radar freeboard can be expressed as follows.

$$F_r^{OIB} = F_t^{OIB} - h_s^{OIB} - (f\eta_s - 1)h_s^{OIB} \tag{AR7}$$

Because the main objective of using OIB data is to evaluate the relative performance of the simultaneous retrieval method when the method is applied to CS2 data, the radar penetration factor ($f$) for OIB data processing was also set to be 0.84 (Artimage and Ridout, 2015). In the data processing chain, $h_s^{OIB}$ was removed if it is smaller than the given uncertainty level of the dataset (~5.7 cm) or it is larger than the total freeboard $F_t^{OIB}$.

The CS2 radar freeboard ($F_r^{CS2}$) was obtained from CS2 ice freeboard dataset. The CS2 ice freeboard data ($F_i^{CS2}$) distributed by NSIDC (Kurtz et al. 2017) assume that radar scattering horizon locates at snow–ice interface and applies a wave propagation speed correction. However, the correction was made using the MW99 snow depth ($h_s^{MW99}$) climatology with an erroneous correction form of $h_c = (1 - \eta_s^{-1}) h_s$, instead of the correct correction form of $h_c = (\eta_s - 1) h_s$ (Mallet et al. 2020). Thus, at this point, it is straightforward to derive the CS2 radar freeboard by removing the correction term as in the following equation.

$$F_r^{CS2} = F_i^{CS2} - (1 - \eta_s^{-1})h_s^{MW99} \tag{AR8}$$

Finally, analyses in the first version of the manuscript were conducted again using the radar freeboard rather than using the ice freeboard. This time SIC criteria for $\alpha$ calculation was set to be 95% (original: 98%) for a wider coverage. Figs. 6–9 in the revised manuscript are the reprocessed results. Despite of these changes, we find little changes in the conclusions we made in the first version of the manuscript. In addition, for more comprehensive information, snow depth comparison results are provided in Fig. 9.

The changes for the definition of radar freeboard can be found in introduction (the second paragraph) and Fig. 1 in the revised manuscript. The changes for method and data can be found in Sect. 2.3 and in Sect. 3.3–3.4, respectively. The corresponding results can be found in Sect. 4.2 and 4.3 and Figs. 6–9.

 **2. Minor comments**

• *I think you need to include an uncertainty budget for your sea ice thickness and snow depth estimates.*

Sensitivity test for our method was conducted and the result is included as Appendix B in the revised manuscript. In Appendix B, snow depth error caused by $\alpha$ error is presented for different cases of $\alpha$ and freeboard.

330

• *L30: "However, the radar scattering horizon is often treated as the snow–ice interface". Include Hendricks et al. (2016), Guerreiro et al. (2017), Tilling et al. (2018) as refs here since AWI, LEGOS and CPOM CryoSat-2 ice thickness products all make the same assumption.*

Hendricks et al. (2016), Guerreiro et al. (2017) and Tilling et al. (2018) are now referred in the revised manuscript.

• *L72: ..."for given densities and freeboard" – (and assuming no snow penetration for laser and full snow penetration for radar)*

We rewrote the sentence as follow:

"… for given densities, freeboard, and assumptions on radar penetration of the snow layer".

• *L138: Could you say how many are discarded based on this criterion, and out of how many total.*

By examining the outputs from the program, we found no outputs discarded by this criterion, therefore, we removed this sentence from the revised manuscript.

Here we provide total number of buoy data obtained from different averaging periods for your information (Table AR1, will not be included in the text).

• *L159: Can you provide a reference for the OIB data processing document where the densities are given?*

We now referred Kurtz et al. (2013) in the revised manuscript.

350

• *L160: I understand that you keep ice density constant in order to compare with OIB data, but later when comparing with satellite-derived ice thickness should you not then use the densities used in those products for a fair comparison?*

As you mentioned, '*CS2 H*' in Fig 9 of the manuscript should not be same as the CS2 product available from NSIDC. Instead, CS2 ice thickness was reproduced with the same densities and MW99 snow depth for the fair comparison. New figure is given in Fig. 9 in the revised manuscript. We also clarified this data processing in the comparison section.

• *L198: "It was reported…" – By who?*

It was reported by Lee et al. (2018). We clarified this in the revised manuscript.

360

• *L200: Where does $T_{si}$ for March come from if the Lee et al. (2018) dataset is only December-February?*

We produced $T_{si}$ for March by applying the same algorithm. We clarified this in the revised manuscript.

365  • *L201: "…if data frequency is over 20". Do you mean if 20 days out of the month contain data? Or are you referring to a number of points per grid cell?*

Monthly mean temperature was calculated by gird cell by grid cell and the average was done only for the grid cells where there are more than 20 data available during a month. We clarified this in the revised manuscript.

370

• *L205: Please could you provide the details and a reference for which OIB dataset you used and where it is available from? i.e. L2, L4, Quicklook?*

We utilized L4 dataset for 2011-2013 period, and Quick look dataset for 2014-2015 period. Details on OIB data are now provided in the revised manuscript. Reference and accessibility information were
375  already included in 'Data availability' section.

• *Figure 4: Why do you choose to show us the 7-day averaged plot in Fig.4 when Figure 3 was showing 15-day averaged temperature profiles?*

We intended to show results from various averaging period to readers. The results for different
380  averaging period can be found in Figs. 5 and 6 in the text. For your information, same figures for different averaging periods are presented in Fig. AR1 (will not be included in the text).

• *L235: At the end of this sentence you could refer the reader to the appendix.*

Appendix A is now referred at the end of the sentence.

385

• *L244: bias is not near-zero in Fig 4b, it is zero.*

Yes, it is zero. The comment is now applied in the revised manuscript.

• *L269: Did you calculate a different alpha for each year, and apply the different alpha to each year of OIB*

390    *data? Or did you just average all the years together? Please clarify this in the text.*

We calculated and applied a different $\alpha$ for each year. It is now clarified in the revised manuscript.

*• L295: Do you get the MW99 for input into Eqs. (4) and (5) from the CS2 data? If so is it monthly grid-averaged? How do you assign each OIB point a snow depth?*

395    Yes, MW99 was obtained from the CS2 dataset. OIB data were reformatted in a 25 km polar stereographic grid by method described in Sect. 3.3 in the manuscript. As OIB data is reformatted into the grid format, it is straightforward to assign the OIB snow depth to the MW99.

One possible concern is that the monthly mean of daily MW99 might differ significantly from the daily MW99, because the MW99 depends on the sea ice type. However, it may not be a critical issue

400    when following points are considered: 1) W99 is already a monthly climatology, 2) ice type distribution would not be changed significantly by the sea ice drift in March because the Arctic Ocean is nearly filled with sea ice and thus the sea ice mobility is reduced.

*• L301: "Therefore, if there are decreasing trends in both ice thickness and snow depth, the decreasing trend of*
405    *ice thickness estimated from the constant snow depth will be diminished in radar, while being amplified in lidar"*
*– This sentence seems overcomplicated. To me, all that the bottom two plots of Figure 7 demonstrate is that*
*MW99 snow depth is larger than OIB snow depth. For the laser case, this means that using W99 causes ice*
*thickness to be underestimated compared to H(OIB), and for the radar case using W99 results in ice thicknesses*
*that are too thick compared to OIB. Perhaps you could plot MW99 against h(OIB) to clarify this? The retrieval*
410    *of sea ice thickness from ICESat has not traditionally used the Warren climatology- see Kwok and Cunningham*
*(2008) and Petty et al. (2020). Therefore I don't think it's justified to call this 'ICESat-like thickness' since you*
*are not using the same snow depth product that they do.*

Yes, what you mentioned is the appropriate interpretation of Fig. 7; MW99 is larger than OIB snow depth. This can be verified by Fig. S1 (included as a supplementary figure). However, we attempted to
415    address a possible unintended result of 'diverging direction in errors in ice thickness retrieval, when the same snow depth error is applied to two different satellite altimetry measurements'. We modified this paragraph to deliver such message.

Regarding the naming issue, we are not referring existing products when we call ICESat-like and CS2-like thickness. Those are explicitly defined in the revised manuscript text. Besides, there is an ice
420    thickness dataset from ICESat total freeboard distributed by NSIDC (Yi and Zwally, 2009; doi: 10.5067/SXJVJ3A2XIZT) which uses MW99. However, we think that the expression "current practices of retrieving sea ice thickness" might confuse readers. Therefore, we replaced such expression with "MW99 method" for clarity.

425

**3. Typos / Grammar**

*• L128: "Therefore, the interface searching algorithm…" -> "Therefore, an interface searching algorithm…"*

*• L165: "Sea ice thicknesses converted from MW99 using Eqs. (4) and (5) are also compared to examine how simultaneous retrievals…" -> "Sea ice thicknesses are also calculated from Eqs. (4) and (5), using MW99 as snow depth, to examine how simultaneous retrievals…"*

*• L194: "This reformatted AASTI-v2 data are called…" -> "This reformatted AASTI-v2 dataset is called…"*

*• L293: "Examining how the current practices of retrieving the sea ice thickness through ICESat and CS2 measurements are compared with the simultaneous retrievals is of interest" -> "We now examine how the current practises of retrieving sea ice thickness from ICESat and CS2 measurements compare with our method."*

*• L294: "In doing so, OIB-measured…" -> "To do so, OIB-measured…"*

*• L297: "Apparently, ICESat-like thickness tends…." -> "According to our analysis, ICESat-like thickness tends…."*

*• L416: "…which are hard to be quantified explicitly." -> "…which are hard to quantify explicitly."*

All comments were applied in the manuscript and some other grammatical errors were also corrected in the revised manuscript.

**Table AR1.** Number of outputs obtained from the interface searching algorithm

| Averaging period | # of obtained outputs |
| --- | --- |
| 1 day | 542 |
| 7 days | 97 |
| 15 days | 59 |
| 30 days | 36 |

500

[revised manuscript text omitted]
 betweento those from the MW99 method and the α method using. (a) Snow depth from OIB iceradar freeboard and, (b) snow depth from CS2 radar freeboard on March 2014. CS2-like H (OIB) denotes the, (c) ice thickness estimated from the MW99 snow depth and OIB iceradar freeboard-, and (d) ice thickness from CS2 radar freeboard.

[Figure]

[Figure]

765 **Figure A1.** Distribution of physical variables on scatterplots of the temperature difference ratio of snow and ice layer ($\Delta T_{snow}/\Delta T_{ice}$) and the snow–ice thickness ratio ($\alpha$). Color denotes the value of physical variables: (a) ice thickness ($H$), (b) snow depth ($h$), (c) air–snow interface temperature ($T_{as}$), (d) snow–ice interface temperature ($T_{si}$), (e) temperature difference within snow layer ($|\Delta T_{snow}|$), and (f) temperature difference within ice layer ($|\Delta T_{ice}|$).

[Figure]

770

**Figure A2.** Histogram of estimated (left column) $k_{ice}$ and (right column) $k_{snow}$. The top and bottom row denote the first and the second part, respectively. The size of the bins is 0.05 W K$^{-1}$ m$^{-1}$.

[Figure]

775 **Figure B1.** Locations of physical states for typical types (A, B, C) on the freeboard-thickness ratio space. Blue dots are from (left) OIB data and (right) retrieved thickness ratio and CS2 radar freeboard.

**Response to Reviewer 2**

**General comment**

*The authors have done a thorough job responding to and addressing my comments. In particular, the application of the methodology to radar freeboard from CryoSat-2 has been demonstrated (I apologise for assuming it would not be possible!). The only thing outstanding is an uncertainty estimate for the CS2-derived snow depth and ice thickness. The sensitivity analysis that has been included in the appendix explores the impact Δα has on snow depth and sea ice thickness, but this alone is not sufficient to describe the uncertainty on these products since, for example, error on radar freeboard should also be accounted for. If these products are to be made available, in particular to the modelling community, they must come with an error budget.*

*The methodology presented in this manuscript is a novel and valuable addition to the sea ice community. I therefore recommend the paper for publication following minor revisions.*

Authors appreciate Dr. Isobel R. Lawrence for providing constructive comments. This time, we included uncertainty budget analysis by conducting error propagation analysis. Please find the following point-by-point responses to the referee's comments below. A marked-up version of the revised manuscript regarding the changes is attached at the end of this authors' response.

**Minor revisions**

**The accuracies of CS2 retrievals / Error budget**

*I find the final paragraph of section 4.3 (L359-369) extremely difficult to follow. Indeed, after studying Figure 9 for some ten minutes I am still at a loss as to what these plots actually tell us. If the aim is to perform a validation against OIB measurements, why not just show a scatter plot of $h_s^{OIB}$ vs $h_s(\alpha^{sat}, F_r^{CS2})$ and $H_i^{OIB}$ vs $H_i(\alpha^{sat}, F_r^{CS2})$? That would be a far simpler and more relevant plot which the reader will understand immediately.*

Sorry for the confusion. We should have provided the reason why we did it in an indirect way. We clarify the paragraph and provide the reason.

"To assess the accuracy of CS2 retrievals, reference snow depth and ice thickness collocated with CS2 freeboard in space and time are necessary. However, different from simultaneous retrievals from OIB freeboards in Sect. 4.2, evaluation with the required matching data may not be possible from the monthly composite of CS2 data used in this study. Here, instead of using monthly collocated match-up data, an indirect way is used to examine the accuracy of CS2 retrievals. We do so by examining whether the relationship between the simultaneous method and the MW99 method, based on retrievals from the OIB freeboard, can be reproduced by CS2-based retrievals. If similar results are obtained, respective accuracies can be deduced against those noted from the evaluation against OIB measurements."

*I think an uncertainty estimate for the satellite-derived products needs including in section 4.3. A simple propagation of errors could be performed on Equation 12 to estimate the error on Hi, and similarly error on $h_s$ could then be propagated from Equation 3. For this, the uncertainty on radar freeboard and $\alpha$ are required. In the manuscript $\Delta\alpha$ is estimated to be 0.036, equal to the RMSE between observed and regressed $\alpha$, where regressed $\alpha$ are derived from buoy-measured interface temperatures. Does the same $\Delta\alpha$ apply for $\alpha$ derived from satellite temperatures? Evidently errors in $T_{si}{}^{sat}$ and $T_{as}{}^{sat}$ will result in errors in $\alpha$. This should at least be discussed in section 4.3, even if it is not possible to incorporate errors on $T_{si}{}^{sat}$ and $T_{as}{}^{sat}$ into the final uncertainty budget (if for example the satellite temperature products do not come with an uncertainty). Errors on radar freeboard should be available with the CS2 product you are using. If not, see discussion in Tilling et al (2018) for their estimate of CS2 radar freeboard uncertainty.*

Your understanding is correct. The value of 0.036 is for buoy temperatures, not for satellite derived temperatures. Incorporating your comment, the difference in root mean square between $\alpha^{OIB}$ and $\alpha^{sat}$ is calculated and is used for $\Delta\alpha$. Appendix B is updated accordingly.

Then, we calculated the uncertainty budget for satellite derived $\alpha$ and CS2 freeboard measurements on ice thickness and snow depth by using Gaussian error propagation equation. The results are in Appendix C and referred at the end of Sect. 4.3.

In the case of ice thickness, freeboard-related uncertainty is greater than $\alpha$-related uncertainty. Total uncertainty of ice thickness estimate ranges from 0.8 m to 2.0 m and total uncertainty of snow depth estimate ranges from 4 cm to 40 cm. It is noted that $\alpha$-related uncertainty is greater than freeboard-related uncertainty for snow depth estimation. Both uncertainties in ice thickness and snow depth are greater for MYI region than for FYI region. It is thought that the improvement of accuracy in satellite derived temperatures can reduce the snow depth uncertainty while the improvement of freeboard accuracy can reduce the ice thickness uncertainty.

**Other minor comments**

*L15: "retrieved ice thickness was found to be better than the methods relying on the use of snow depth climatology as input, in terms of mean bias and RMSE." - This is not true, RMSE on ice thickness from radar freeboard is smaller using the MW99 method (0.344 vs 0.5 $\alpha$-method, figure 7)*

The sentence is corrected by removing RMSE. Now the new sentence reads as follow.

"retrieved ice thickness was found to be better than the methods relying on the use of snow depth climatology as input, in terms of mean bias."

*L68: "Other satellite remote sensing approaches include the snow depth retrieval using dual-frequency altimetry (Guerreiro et al., 2016; Lawrence et al., 2018, 70 Kwok and Markus, 2018),*

*multilinear regression (Kilic et al., 2019), and a neural network approach (Braakmann-Folgmann and Donlon, 2019)." – I think here you need to add something about the limitations of these methods. Otherwise it is unclear why a new snow product is necessary.*

The following sentence is now added regarding limitations of those studies.

"In spite of promising results, the dual frequency altimetry method is available only for regions where two altimeters overlap with each other, reducing the great deal of spatial coverage. On the other hands, the regression/neural network methods based on AMSR-2 TBs are prone to the overfitting problem, limiting their applications to other microwave sensors."

*L170: "A sensitivity test indicated that the influence of a 0.3°C difference in the freezing temperature on $\alpha$ was negligible". Could you give a percentage value or some quantification of it being negligible?*

Additional information is now provided at the end of the sentence.

"… (e.g. approximately 1.2% difference for typical interface temperatures of $T_{as}$ = -30 °C and $T_{si}$ = -20 °C)"

*L233: The Quicklook dataset URL you provide takes you to 'Bootstrap Sea ice concentrations". Please check the DOI. Also I suggest moving the url to the end of the sentence.*

Thanks for the comment. DOI is corrected and the url has been moved to the data availability section. It is now located at the end of the sentence.

*L235: "The OIB data are also reformatted into the 25 km grid format for comparison. If the location of one OIB individual data point falls within a certain 25 km grid area, then the point data is binned in a corresponding grid. After completing the grid assignment, grid value is determined by calculating a simple arithmetic mean of all data within that grid area." – Do you just mean "the OIB data are averaged on the same 25km grid"?*

The sentence is now clarified, as follows:

"The OIB data are also reformatted into the 25 km grid format by averaging pixel-level OIB observations on the 25 km grid."

*L251: I find lines 245 to 250 slightly confusing. I suggest you move the equation for $\eta_s$ to after equation 14. i.e:*

*"[Eq 14],*

*where $\eta_s$ = [...] and $\rho_s$ is taken from the Warren climatology, after Kurtz (2017)"*

Following the comment, the equation for $\eta_s$ is located after Eq. (14):

[Eq. (14)]

"Here, $\eta_s$ was parameterized as a function of the snow density, i.e. $\eta_s = (1 + 1.7\rho_s + 0.7\rho_s^2)^{0.5}$ (Tiuri et al., 1984), and $\rho_s$ is taken from the W99 climatology, after Kurtz and Harbeck (2017)."

*Figure 3 caption: "The period number is equivalent to the number of time-averaging bin." – I do not understand what the period number is.*

If time averaging period is 15 days, there are 10 time-averaging bins because 151 days (November 1 to March 30) are divided into 15-day increments. We sequentially numbered each time averaging bin which has a 15-day length. That number is the period number in Fig. 3 caption. The following figure for the averaging period of 15 days would help understanding.

[Figure]

For clarification, the description is changed as follow.

"The period number indicates the sequential 15-day period from November 1 (e.g. 'period: 2' denotes a time-averaging period of November 16th to November 30th)."

**Typos / Grammar**

*L19: "…buoyancy equation and radar penetration…" -> "…buoyancy equation or the radar penetration…"*

*L26: "…the height from the sea surface in cracks and leads to the snow surface." -> "…the height from the sea surface in leads, to the snow surface."*

*L51: "variation of snow–ice system" -> "variation of the snow–ice system"*

*L58: "TB's" -> "TBs"*

*L156: Remove "respectively"*

*L181: "…by multiplying the obtained sea ice thickness and $\alpha$." -> "…by multiplying the obtained sea ice thickness and $\alpha$ (Eq. (3))."*

*L194 "…as parts…" -> "…as part…"*

*L201: "depending" -> "dependent"*

Sentences are corrected following your suggestion.

*L248: "In this dataset, $\eta_s$ was parameterized as a function of the snow density" -> "$\eta_s$ was parameterized as a function of the snow density" – The 'in this dataset' suggests to me that you mean your dataset!*

Sorry for the confusion. 'in this dataset' is now removed from the sentence.

*L253: "...values are used for comparison." -> "...values are used for comparison with results from our simultaneous method."*

*L307: "scatterplots of comparing retrievals" -> "scatterplots comparing retrievals"*

*L351: "shows $\alpha$ values that is generally larger than that over" -> "shows $\alpha$ values that are generally larger than those over"*

*L353: "$H_i$ shows a similar geographical distribution as shown in the freeboard (the first row)" -> "$H_i$ shows a similar geographical distribution to radar freeboard (the first row)"*

*L356: "and results are given at the bottom" -> "and results are shown in the bottom row"*

*L356: "The obtained snow distribution indicates that thicker snow areas are generally coincident with thicker MYI areas. Likewise, the thinner snow area coincides with the thinner FYI area" -> "The obtained snow distribution indicates that thicker (thinner) snow areas are generally coincident with thicker MYI (thinner FYI) areas."*

Sentences are corrected following your suggestion.

*L385: "As a matter of fact, the ice thickness results were more accurate than they were from the current retrieval methods relying on the input of snow depth (this time MW99 snow climatology), in terms of mean bias and RMSE." – This sentence is not accurate. RMSE on ice thickness from radar freeboard is smaller using the MW99 method (0.344 vs 0.5 $\alpha$-method, figure 7)*

Sentence is corrected (removed RMSE).

*L406: "The results that radar freeboard and the total freeboard yielded had nearly the same outputs when the $\alpha$-approach was used" – This sentence does not make sense to me.*

This sentence is removed.

*L455: "hard wind slap" -> "hard wind slab"*

Sentence is corrected.

*L471: "Because hs is a combination of freeboard and $\alpha$" - Do you mean "Because Hi is a combination of freeboard and $\alpha$"?*

Both ice thickness and snow depth are combination of freeboard and $\alpha$. Sentence is corrected as follow.

"Because $H_i$ and $h_s$ are the combination of freeboard and $\alpha$"

*L476: "With $\Delta\alpha= \pm0.03$, which is an RMSE range in the $\alpha$-prediction equation" – From figure 4b, the RMSE = 0.04, not 0.03.*

$\pm0.03$ was RMSE range in the $\alpha$-prediction equation for the 30-day averaging period. However, regarding your comment, root mean square difference value between $\alpha^{OIB}$ and $\alpha^{sat}$ is used for $\Delta\alpha$. This part has changed as follow:

"With $\Delta\alpha = \pm0.05$, which is the root mean square difference (RMSD) value between $\alpha^{OIB}$ and $\alpha^{sat}$"

*L483: "a much small number" -> "a much smaller number"*

[revised manuscript text omitted]